# Topological identification and interpretation for single-cell gene regulation elucidation across multiple platforms using scMGCA

Zhuohan Yu[1], Yanchi Su[1], Yifu Lu[1], Yuning Yang[2], Fuzhou Wang [3], Shixiong Zhang[3], Yi Chang[1], Ka-Chun Wong [3] ✉ & Xiangtao Li [1] ✉

Single-cell RNA sequencing provides high-throughput gene expression information to explore cellular heterogeneity at the individual cell level. A major challenge in characterizing high-throughput gene expression data arises from challenges related to dimensionality, and the prevalence of dropout events. To address these concerns, we develop a deep graph learning method, scMGCA, for single-cell data analysis. scMGCA is based on a graph-embedding auto-encoder that simultaneously learns cell-cell topology representation and cluster assignments. We show that scMGCA is accurate and effective for cell segregation and batch effect correction, outperforming other state-of-the-art models across multiple platforms. In addition, we perform genomic interpretation on the key compressed transcriptomic space of the graph-embedding autoencoder to demonstrate the underlying gene regulation mechanism. We demonstrate that in a pancreatic ductal adenocarcinoma dataset, scMGCA successfully provides annotations on the specific cell types and reveals differential gene expression levels across multiple tumor-associated and cell signalling pathways.

Single-cell RNA sequencing (scRNA-Seq)[1] allows the investigation of transcriptomic landscapes; it is an increasingly popular platform for characterizing cellular heterogeneity[2], discovering complex tissues and diseases[3,4], and inferring cell trajectories[5] at the single-cell level. Recently, many computational models have been developed to distinguish and annotate cell types, enabling efficient downstream analysis[6]. However, these computational models often suffer from various challenges due to the high dimensionality of data and high probability of dropout events from the low capture rate and insufficient sequencing depth[7–10]. There is an urgent need to develop effective computational models that capture the relationships between cells and identify the high-probability dropout events in scRNA-seq data.

As an unsupervised learning method, cluster analysis has become a key step in defining cell types based on transcriptome data as well as the basis for downstream analyses. Accordingly, clustering methods

have been well-developed to address the challenges over the past years; for instance, five very popular toolkits including SC3[11], Seurat[12], SCANPY[13], SINCERA[14] and SingleR[15] have been developed for the downstream analysis of scRNA-seq data. SC3 is a consensus clustering algorithm using genetic filtering and PCA and Laplacian transformation[11]. Seurat integrates scRNA-seq data with *insitu* RNA patterns to infer cell locations and clusters[12] while SCANPY is a scalable toolkit for analyzing single-cell gene expression data built jointly with anndata[13], both using shared nearest neighbor (SNN) modular optimization and Leiden for clustering. SINCERA converts the data to a z-score before clustering and then cell types are identified in the hierarchical structure[14]. In general, these algorithms provide fast analysis and presentation to users' lacking prior scRNA-seq knowledge.

While these toolkits provide valuable information, it is apparent that current dimensionality reduction methods often either suffer from multiple simultaneous techniques and biological variability or

[1]School of Artificial Intelligence, Jilin University, Jilin, China. [2]Donnelly Centre for Cellular and Biomolecular Research, University of Toronto, Toronto, ON, Canada. [3]Department of Computer Science, City University of Hong Kong, Hong Kong SAR, China. ✉e-mail: kc.w@cityu.edu.hk; lixt314@jlu.edu.cn

ignore the intrinsic data distribution, resulting in the issue of over-crowding in the latent space and thus inaccurate cell clustering. To address these limitations, deep embedding clustering approaches have been successfully developed to model the high-dimensional and sparse scRNA-seq data; for example, deep count autoencoder network (DCA) uses an autoencoder based on negative binomial loss, with or without zero expansion to denoise scRNA-seq count data[16]. On this basis, scDeepCluster learns the feature representation and clustering simultaneously in the autoencoder from the zero-inflated negative binomial distribution model[17]. A year later, scDeepCluster was extended to scDCC, with a prior exploitation of the domain knowledge in a semi-supervised manner. In addition, other deep embedding models have been proposed to enhance the clustering results from diverse perspectives such as, DESC[18], scVI[19], scCAEs[20], scDHA[21] and SCA[22]. However, these deep embedding clustering methods typically ignore cell topological information and the heterogeneity among cell populations. Recently, emerging graph neural networks (GNNs) have been shown to naturally model heterogeneous cell-cell relationships and the complex gene expression patterns from compressed latent spaces[23,24].

Here, motivated by these discoveries, we propose a graph convolutional autoencoder framework, called scMGCA, to analyze scRNA-seq datasets. scMGCA adopts the graph convolutional network (GCN) as an encoder to extract the key structural information of the scRNA-seq gene expression count matrix and cell graph; it is paired with a multinomial-based decoder to capture the global probability structure of the data. Then, self-optimized embedded clustering is applied to cluster the low-dimensional representation by adopting the Kullback-Leibler (KL) divergence. In addition, three training losses including the clustering loss, multinomial-based loss and cell graph reconstruction loss, are simultaneously optimized to discover the cell clustering label assignment and to protect the cell-cell topological representation. Moreover, to formulate the cell-cell relationships, we propose the cell-PPMI graph with a positive pointwise mutual information (PPMI) matrix and random surfing to aggregate adjacent cells in the co-occurrence probability PPMI matrix. We evaluate the clustering performance of scMGCA by comparing it with the state-of-the-art clustering methods on several real scRNA-seq datasets, and reveal that scMGCA is superior to these scRNA-seq clustering methods for cell segregation and visualization. In particular, scMGCA is also able to correct for the batch effect of data coming from different scRNA-seq protocols. We then analyze the functional genomic interpretation of the principal compressed transcription space in the graph-embedding autoencoder to provide robust understanding of the functional significances of scMGCA. To further demonstrate the potential of scMGCA in tumor disease research, we apply it to the pancreatic ductal adenocarcinoma (PDAC) dataset and identify the cell types which elucidated PDAC-related regulatory mechanisms and PDAC cellular communication.

## Results

scMGCA can learn the low-dimensional representation from high-dimensinal and sparse scRNA-seq data by providing a global view of the entire cell graph and count matrix (Fig. 1) to aid in the downstream analysis of scRNA-seq data. It has five main steps to learn the graph-embedding representation. (1) A cell-PPMI matrix is generated on the normalized single-cell gene expression matrix to capture the cell-cell topology. (2) We established the framework of graph convolution autoencoder. GCN is used to integrate the expression matrix and cell graph (cell-PPMI) to extract the main gene information and cell topology, and to preserve it in a latent embedded representation. (3) High probability dropout events of the single-cell data are simulated by multinomial distribution, and a multinomial-based decoder is developed to characterize multimodal distribution. (4) A self-optimized clustering task based on Kullback-Leibler (KL) divergence is performed on the embedded representation, which is

simultaneously trained with the loss functions in the graph convolution autoencoder. (5) The latent embedded representation learned by scMGCA enables clustering, dimensionality reduction, and visualization of the single-cell data. In addition, it is possible to interpret the biological significance of genes and discover disease regulatory mechanisms.

We first normalized the scRNA-seq gene expression matrix and selected the highly variable genes, and then built a cell graph that represents the cell-cell relationships of the normalized data. For the construction of cell graph, we used a random surfing algorithm based on KNN graph to aggregate neighboring cells. Intuitively, the random surfing algorithm captures the transition probability between different vertices from the unweighted KNN graph structure information. After that, the positive pointwise mutual information (PPMI) matrix was calculated to describe similar cells by assigning a co-occurrence attention mechanism, which further enhances the cell graph. Finally, the cell graph, an undirected weighted graph, can be fed into the model.

Under the framework of graph convolution autoencoder, scMGCA uses two layers of GCN to effectively extract the gene expression information and cell-cell topological structure from the scRNA-seq data matrix and cell graph, which is then preserved in a low-dimensional embedded representation. To capture the global probabilistic structure of the data, a multinomial-based decoder was developed to characterize scRNA-seq data distribution and simulate dropout events using the multinomial distribution. It should be noted that the zero-inflated distribution was not used here as recent studies have shown that single-cell count matrices largely follow a multinomial distribution[25]. In addition, we adopted an inner product decoder to reconstruct the content and structural information of the cell graph from the low-dimensional representation.

Finally, we perform the Kullback-Leibler (KL) divergence on the latent representations to self-optimize the clustering task by enhancing the auxiliary target distribution. Then, by simultaneously revising clustering loss, multinomial-based loss and cell-graph reconstruction loss, scMGCA can jointly evaluate the clustering label assignment and feature learning of topological structures by training the entire graph convolution autoencoder, which promptly compensates for model deviations from the correct clustering distribution. After that, we can directly obtain the predicted cluster labels from the final optimized result. In particular, we selected the best strategy from K-means and spectral clustering to initialize the clustering centers, resulting in the robust clustering performance of the overall model. In addition to its significant clustering performance, the low-dimensional latent representation learned by scMGCA also provides good reduction and visualization effects of the original data. In terms of application, we transferred the standard deviation rank of the weight matrix into the network to screen for genes with biological interpretability in the latent embedded representation, providing new marker genes for the single-cell datasets. Further, scMGCA was able to identify cell types in cancer datasets and explore disease regulatory mechanisms.

### scMGCA provides better performance than other single-cell clustering methods across multiple platforms

We compared scMGCA with 12 other single-cell clustering methods on 20 real scRNA-seq datasets across multiple platforms to evaluate the clustering performance of scMGCA. These benchmark methods are divided into three main categories, namely deep embedded clustering methods (scziDesk[26], scDeepCluster[17], DCA[16], DEC[27], DESC[18], scVI[19], scCAEs[20]), deep graph embedded clustering methods (scGNN[23], scGAE[28], GraphSCC[24]) and basic single-cell clustering methods (Seurat[12], SHARP[29]).

The ground truth cell type labels for these datasets are available, and we used normalized mutual information (NMI), adjusted rand index (ARI) and average silhouette width (ASW) as the evaluation

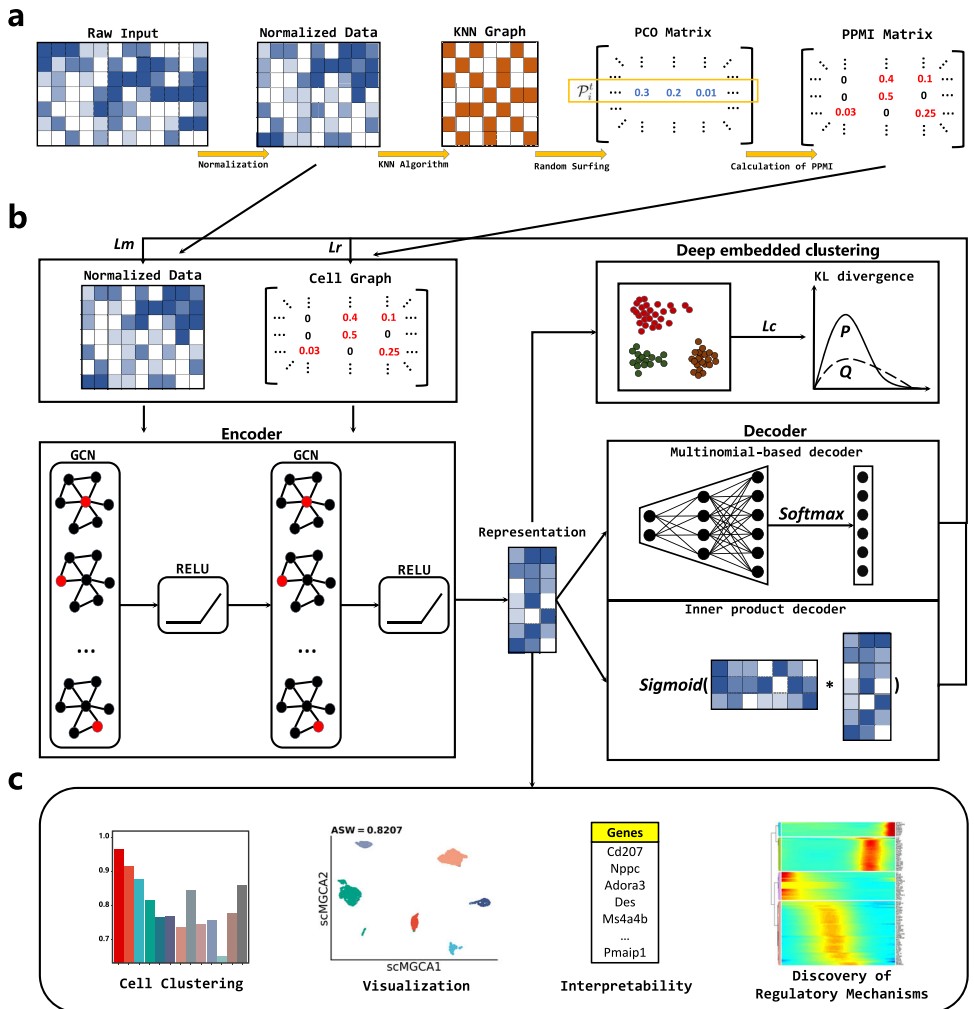

**Fig. 1 | The network architecture of the scMGCA algorithm. a** The construction process of the cell-PPMI graph. **b** The workflow of scMGCA. **c** Functional genomics application capabilities of latent embedded representations trained by scMGCA.

metrics of the clustering performance (Fig. 2a and Supplementary Figs. 1–2). As shown in those figures, for the 20 datasets, scMGCA has the highest NMI, ARI and ASW values for 15 of them, respectively. Overall, across the 20 datasets, scMGCA has the highest mean NMI, ARI and ASW values, reaching 0.8304, 0.8278, and 0.5827, respectively. After scMGCA, scziDesk has superior NMI and ARI values, while scVI has superior ASW values. In addition, we also compared the overall clustering performance (by NMI, ARI and ASW) of the methods across various data platforms, including plate-based platform, flow-cell-based platform, Smart-Seq2, SMARTer, 10X Genomics, and Drop-seq. The experimental results indicated that scMGCA outperformed the other 12 clustering algorithms on across multiple platforms, and demonstrating the effectiveness and precision of scMGCA in clustering across multiple platforms. In particular, we optimize the compared deep learning methods and take their best case as the final result (Supplementary Tables 1–18).

In addition, we visualized the low-dimensional embedded representations of scMGCA and 10 of the other embedded clustering methods in two dimensions using UMAP[30] (Fig. 2b). Overall, scMGCA and scziDesk outperformed the other methods for cell clustering on the 'Qx Limb Muscle' dataset, confirming the results with NMI. However, scziDesk had a few cell clustering results that are confounded in the skeletal muscle cell and macrophage clusters, while scMGCA successfully clustered almost all of the cells on the 'Qx Limb Muscle' dataset. This may indicate that the latent embedded representation of scMGCA effectively preserves the key information and data distribution learned from the

scRNA-seq gene expression matrix and cell graph. We performed UMAP visualization comparisons on all datasets (Supplementary Figs. 3–7). In addition, to investigate whether scMGCA can detect rare cell types and small clusters that cannot be detected by other methods, we compared scMGCA with other deep learning methods including scziDesk, scDeepCluster, DCA, DEC, scGNN, scGAE, and GraphSCC on four datasets ('QS heart', 'Muraro', 'Qx limb Muscle', and 'Adam') that contain rare cell types and small clusters for in-depth examination (Supplementary Note 3 and Supplementary Fig. 8).

We also compared the running time of scMGCA with other clustering algorithms. For the test dataset, we adopted 20 small-scale datasets with 300–25,000 cells, and a large-scale dataset called 'Tabula Muris' with 70,118 cells from 20 organs and tissues. The experimental results are depicted in Fig. 2c. It is worth noting that scMGCA divides the data into multiple batches for learning when clustering datasets with >25,000 cells. Figure 2c shows that the running time of the traditional methods, including Seurat and SHARP, is always faster than all deep learning methods. Compared to other GNN-based methods, our proposed model can provide shorter running time. We also found that scGAE cannot be run on the datasets with >12k cells. We observe that the running time curve of scMGCA drops and is within reasonable ranges (i.e. all less 1.5 h) as a result of the use of batch learning for the 'Tabula Muris'. To explore the specific changes in the running time of scMGCA with the number of cells and genes, we tested the respective running time of scMGCA for different numbers of cells and different numbers of genes on 'Tabula Muris' (Supplementary Fig. 9). It can be observed that

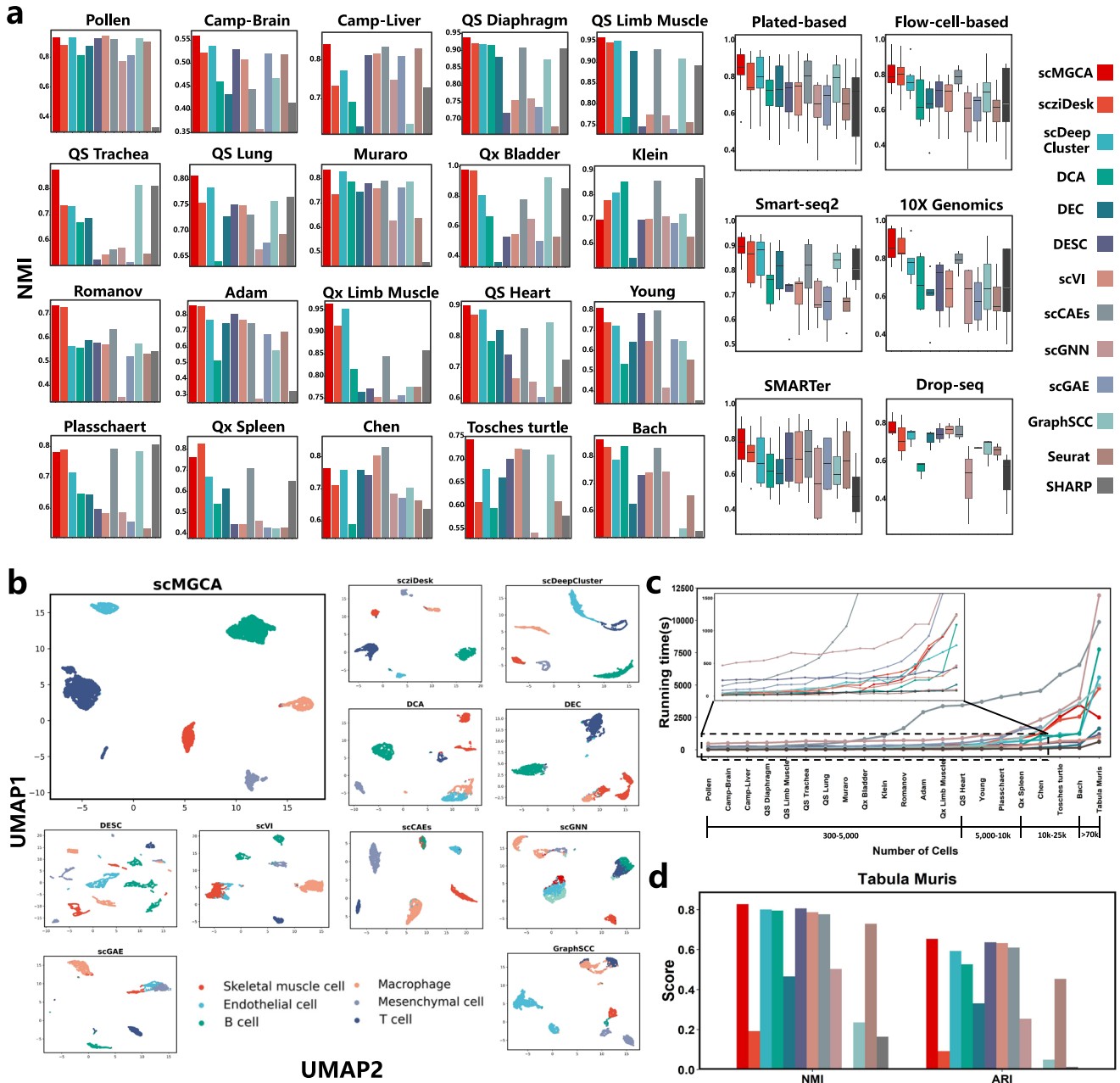

**Fig. 2 | Cell clustering performance of scMGCA compared to other state-of-the-art clustering methods. a** Comparisons of NMI values between scMGCA and 12 single-cell clustering methods on 20 real scRNA-seq datasets across multiple platforms (left, $n = 1$ in each group) and on different platforms (right, $n = 10, 10, 5, 5, 4, 3$ in each group; center line, median; box limits, upper and lower quartiles; whiskers, $1.5 \times$ interquartile range). **b** Comparison of clustering results with 2D visualization by UMAP on the 'Qx Limb Muscle' dataset. **c** Comparison of the running time between scMGCA and other state-of-the-art clustering methods. **d** The clustering performance of all 13 methods on the 'Tabula Muris' dataset. The blank bar chart in figures generated by scGAE are due to the fact that scGAE cannot run them even with large memory. Source data are provided as a Source Data file.

scMGCA demonstrates a monotonically linear increase in running time along with the number of cells and genes, demonstrating its potential as the computing power increases in the future. Further, scMGCA still has good clustering performance on 'Tabula Muris' (Fig. 2d).

Finally, to further demonstrate the scalability of scMGCA, we applied scMGCA to the dataset composed of 1,306,127 mouse brain cells (Supplementary Note 5 and Supplementary Fig. 10). Most comparison algorithms failed or produced no results within 24 h for such a big dataset. Ultimately, only scMGCA, DEC, DESC, and traditional methods were able to run successfully. Therefore, we compared their running time and memory usage at different numbers of cells (Supplementary Fig. 11). From the experimental results, we observe that,

excluding the extra time and memory consumption from the cell graph computation, there is not much difference between scMGCA and the deep learning methods. Moreover, it is worth noting that, among several graph neural networks compared in this paper, scMGCA is the only one that can run successfully; it can also be regarded as an advancement in deep graph learning method development on large-scale scRNA-seq datasets.

## scMGCA effectively performs dimensionality reduction correction and improves visualization of scRNA-seq data

Dimensionality reduction and visualization of high-throughput data have always been the main goals of scRNA-seq data clustering to

facilitate downstream analysis and cell type discovery. Here, we compared scMGCA with three other dimensionality reduction methods, namely PCA[31], t-SNE[32], and UMAP[30]. We adopted the average silhouette width (ASW) as the evaluation metric that measures true cell labels and the reduced matrix. ASW reflects the degree of aggregation of the same cell type and the extent of separation between different cell types, and a value close to 1 indicates good performance. Since t-SNE and UMAP have many parameters which can greatly affect the 2D embedding, we compared scMGCA with t-SNE and UMAP in wider parameter space; and the experimental results demonstrated that scMGCA is superior to t-SNE and UMAP (Supplementary Note 6 and Supplementary Tables 19–21). We then compared the best case of t-SNE and UMAP with PCA and scMGCA (Fig. 3a). Figure 3a depicts that scMGCA has the highest ASW on 'QS Trachea' and 'QS Heart' datasets, and the performance of dimensionality reduction correction and visualization significantly outperforms the other compared methods. Furthermore, the overall average ASW obtained on the 20 datasets is much higher than that of the other methods in Fig. 3b. The ASW values and visualization performance of all datasets are shown in Supplementary Figs. 12–15.

We also carried out a comprehensive comparison between scMGCA and two state-of-the-art single-cell transcriptome analysis methods (SCANPY and Seurat), measuring their clustering performance, dimensionality reduction, and visual comparison (Fig. 3c). scMGCA performs dimensionality reduction correction and clustering based on critical gene expression information and topology captured by a graph neural network. SCANPY and Seurat utilize PCA to reduce the dimensionality of the gene expression matrix, and then cluster in a low-dimensional space. The functions and parameter settings used in SCANPY and Seurat are shown in Supplementary Note 8. We used the UMAP embedded scatter plot to grade the three methods and compared the predicted labels with the true labels on three datasets, 'QS Limb Muscle', 'QS Diapharm' and 'Qx Bladder'. In Fig. 3c, we certainly observe that scMGCA performs better than SCANPY and Seurat in terms of effectiveness and clustering performance when dimensionality reduction correction, and it separates most of the cells correctly. Specifically, on the 'QS Diapharm' dataset, scMGCA identifies better the compacted mesenchymal and skeletal muscle cell clusters compared to the other methods. On the 'Qx Bladder' dataset, scMGCA is able to distinguish urothelial cells from mixed bladder cells. For comprehensive analysis, we chose five evaluation metrics (Davies-Bouldin, ASW, NMI, ARI and V-Measure) to assess the three models (Fig. 3d). We observed that scMGCA surpasses SCANPY and Seurat for all metrics on the three datasets, demonstrating scMGCA's strong clustering performance. Comparisons on all the datasets are presented in Supplementary Figs. 16–18.

### scMGCA can remove the batch effect in human pancreatic data from different scRNA-seq protocols

Batch effect correction and clustering of data generated by multiple different scRNA-seq protocols is challenging due to the strong batch differences among different scRNA-seq protocols. To investigate the batch effect of data generated by different scRNA-seq protocols, we combined four publicly available human pancreas datasets generated from CEL-seq[33], CEL-seq2[34], Fluidigm C1[35], and Smart-seq2[36]. For benchmark comparison, we selected five state-of-the-art batch effect correction methods including DECS[18], Harmony[37], MNN[38], scVI[19], and Scanorama[39].

In Fig. 4a, we observe that scMGCA can effectively merge datasets from different scRNA-seq protocols and remove the batch effect. Compared to other batch effect correction methods, both DESC and Harmony can substantially complete correction of the batch effect, but they do not mix all cells uniformly. scVI could only mix the datasets from CEL-seq and CEL-seq2, with no correction of Fluidigm C1 and Smart-seq2. Scanorama did not remove the batch effect from Smart-

seq2, while MNN completely separated the data generated by the four scRNA-seq protocols. Furthermore, Fig. 4a also reveals that scMGCA can effectively aggregate cells of the same type and separate cells of different types. DESC also aggregated most cells, with the exception of beta cells which were mixed with alpha cells, acinar cells, and ductal cells. Harmony mixed alpha cells are mixed with acinar cells, delta cells, and beta cells. The three methods (MNN, scVI, and Scanorama), failed in effectively seperating the different cell types, resulting in downstream confusion. These findings are also confirmed by the comparison of the clustering performance as depicted in Fig. 4b. Overall, scMGCA and DESC are superior to other batch effect correction methods, whereas MNN, scVI, and Scanorama perform poorly across those three metrics: NMI, ARI, and ASW.

To explore the process by which scMGCA removes batch effects, we visualize the latent embedding representation of scMGCA at different epochs. As shown in Fig. 4c, scMGCA is capable of aggregating the cells and gradually mixing the datasets generated by the various scRNA-seq protocols. This demonstrates that scMGCA not only accurately clusters cells but also simultaneously corrects for batch effect in multiple datasets with strong batch differences.

### Evaluation of hyperparameter selection and ablation study

We evaluated the impact of different numbers of cell clusters, transfer steps, numbers of selected genes and diverse cluster center initialization methods on scMGCA, as well as the effectiveness of each component in scMGCA of the 20 real scRNA-seq datasets.

For the cluster number effect, we compared the clustering performance of scMGCA after perturbing the cluster number, that is, we set the number of experimental clusters to $\{K-2, K-1, K, K+1, K+2\}$, where $K$ is the true cluster number. The heatmap of Fig. 5a indicates that, for most datasets, a better NMI value is achieved for the true number of clusters. The datasets with a low number of clusters, such as 'Qx Bladder' ($K=4$), 'QS Trachea' ($K=4$) and 'QS Diapragm' ($K=5$), are more affected by disturbance than others. The reason may be that they contain less cell populations and unstable structures. On the whole, scMGCA is insensitive to the number of clusters. The heatmap of the ARI measures is depicted in Supplementary Fig. 19.

Figure 5b shows the clustering performance of scMGCA with different numbers of transfer steps ($s$) and different numbers of selected genes ($d$). The results demonstrate that scMGCA has the best overall clustering performance when $s=2$ and $d=500$ compared to other conditions. The initialization of the clustering centers is very critical to the clustering process and will influence the final clustering performance. As scMGCA's clustering center initialization method is based on the optimal strategy in kmeans and spectral clustering, we conducted correlation analysis in the context of purely kmeans, spectral or random initialization methods, respectively (Fig. 5c). We observed that kmeans and spectral clustering models are indeed more relevant to scMGCA than random initialization methods. The kmeans has the highest correlation, which also implies that scMGCA chooses more clustering centers initialized by kmeans on the 20 scRNA-seq datasets. Further, we elaborated on the conditions and reasons under which the two initialization methods, kmeans and spectral clustering, can be effective, respectively (Supplementary Note 10 and Supplementary Table 22). We also analyzed other parameters in detail, including the weights of loss function, the parameters of cell-PPMI graph, the number of network layers, and different network frameworks (Supplementary Note 11), so as to give users an effective guide.

We performed an ablation study on scMGCA to investigate if the cell-PPMI graph augments the traditional KNN graph, and whether the multinomial-based decoder is more effective than ZINB in modeling single-cell data distribution. Specifically, we compared the clustering performance of scMGCA, scMGCA with KNN graph and scMGCA with ZINB decoder, using NMI (Fig. 5d) and ARI measures (Supplementary Fig. 25). The experimental results showed that scMGCA had the best

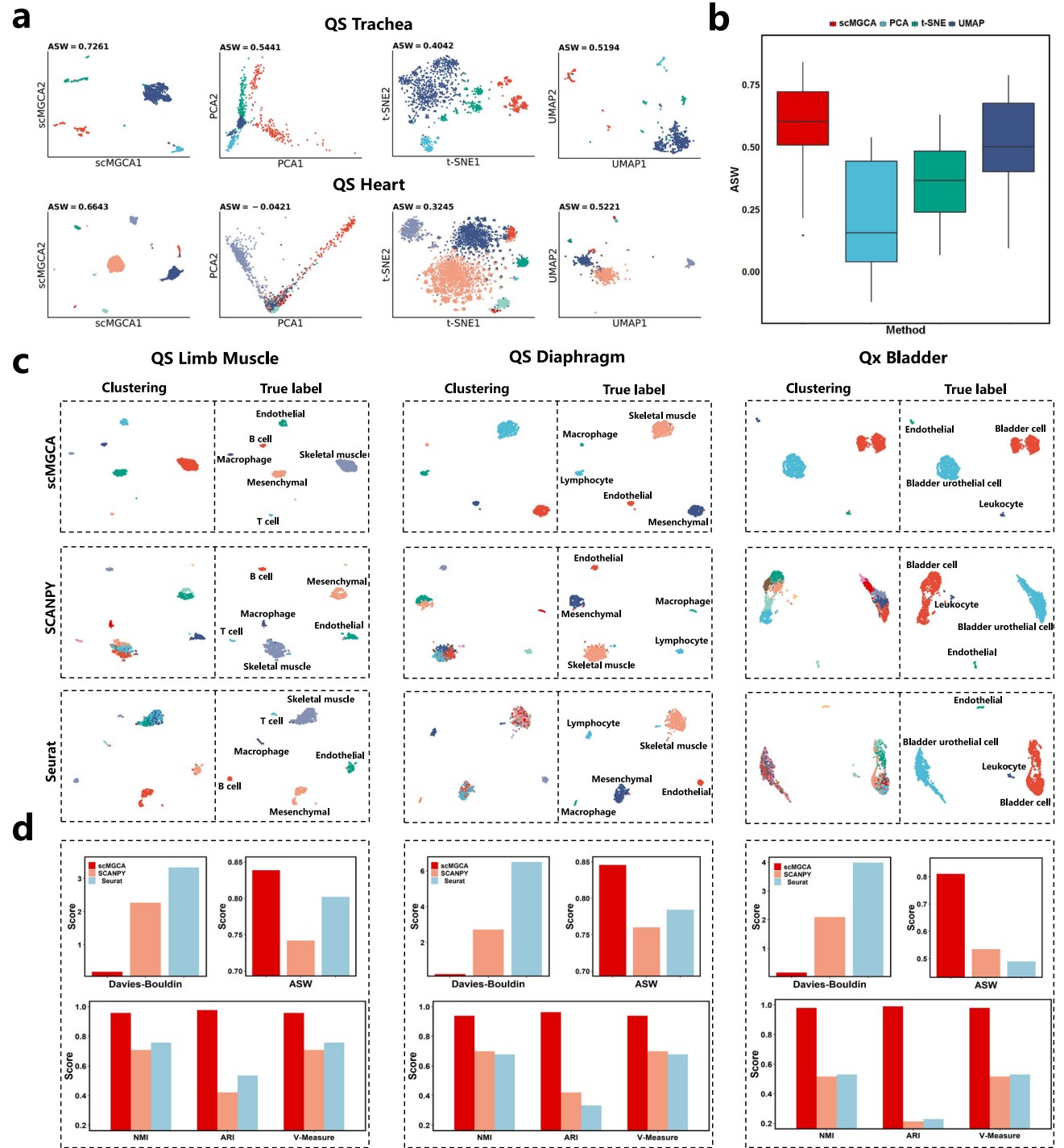

**Fig. 3 | Dimensionality reduction performance and visualization capacity of scMGCA.** **a** Visualization of scMGCA and three dimensionality reduction methods (PCA, UMAP and t-SNE) on 'QS Trachea' and 'QS Heart'. **b** Comparison of the ASW across 20 datasets (*n* = 20; center line, median; box limits, upper and lower quartiles; whiskers, 1.5 × interquartile range). **c** Comparison of scMGCA and two single-cell analysis packages (SCANPY and Seurat) on real clusters via UMAP, **d** assessed by a variety of evaluation metrics on 'QS Limb Muscle', 'QS Diaphram' and 'Qx Bladder'. Source data are provided as a Source Data file.

clustering performance on most datasets; and that removing the cell-PPMI graph and multinomial-based decoder reduced the performance of scMGCA for cell clustering. In particular, we further verified that the cell-PPMI graph is more suitable for single-cell data than the KNN graph (Supplementary Note 12). In addition, we also compared the performance of scMGCA with other clustering algorithms for dropout correction (Supplementary Note 13 and Supplementary Figs. 27–28). From the experimental results, it was observed that both NMI and ARI values of scMGCA decrease slightly on most datasets and it

outperforms all the other comparison methods. These experimental findings confirm that scMGCA is sufficiently stable and robust to provide promising performance in dropout event correction.

**Functional genomics interpretability of the latent embedded characterization of scMGCA**

We conducted a functional genomics interpretability analysis of the latent embedded representation of scMGCA on the 'Qx Limb Muscle' dataset, and validated that it preserves the key information of the

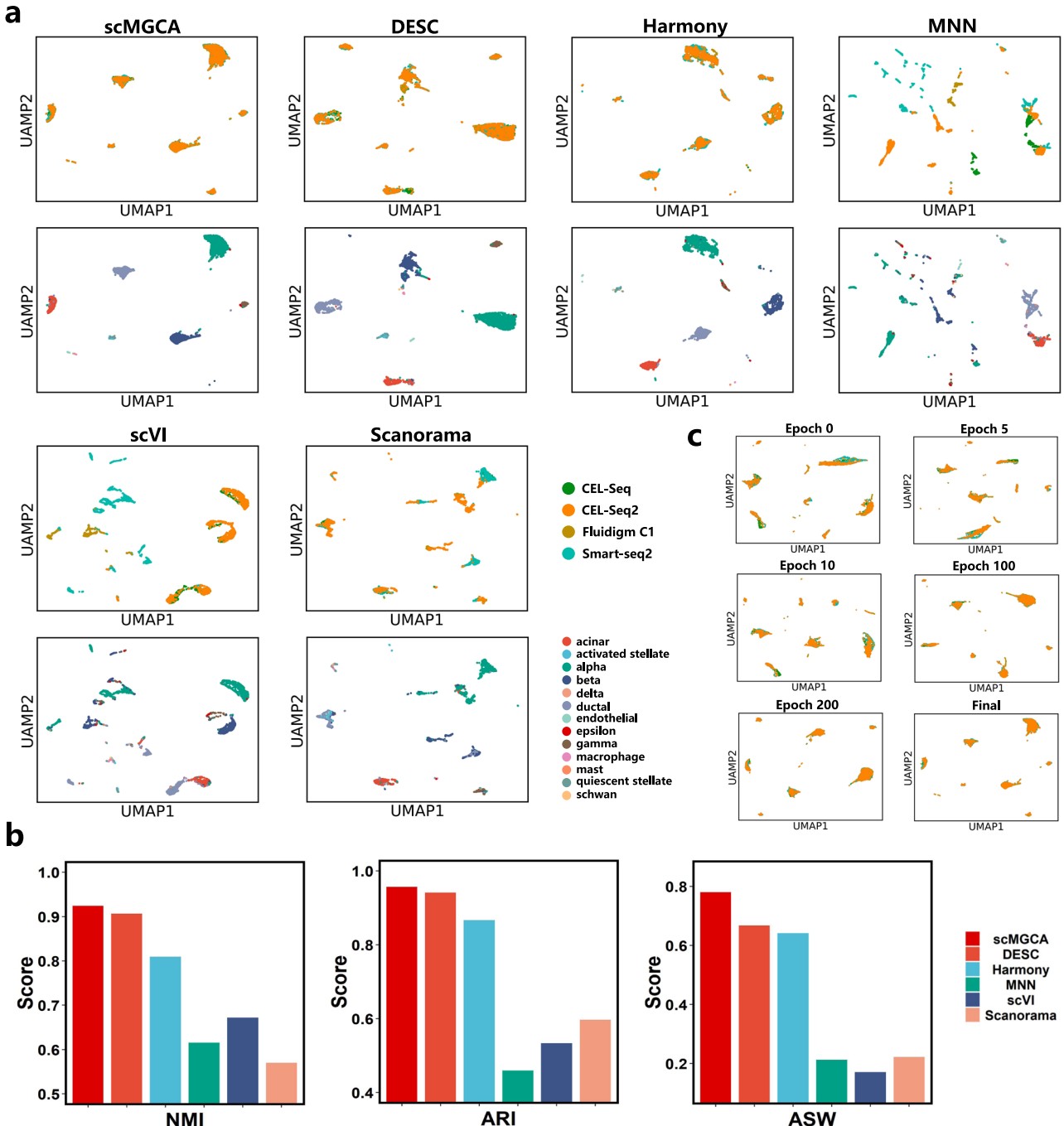

**Fig. 4 | Comparison of batch effect correction methods on the pancreatic islet data generated by different scRNA-seq protocols. a** Visualizing scMGCA with different methods for batch effect correction via UMAP. The cell batches are colored in the top row and cell types in the bottom row. **b** Comparison of clustering performance with evaluation metrics NMI, ARI and ASW. **c** UMAP plots showing scMGCA removes the batch effect gradually over iterations. Source data are provided as a Source Data file.

scRNA-seq gene expression data and thus has the capacity of functional genomics interpretation.

First, we used the t-SNE algorithm to project the latent embedded representation into a two-dimensional space at different stages of training and up to the end (Fig. 6a). The results show that, as the training progresses, cells on the latent representation gradually cluster together; and the cell clusters become more and more obvious, reflecting that the training and learning of the latent embedded representation is meaningful and effective. Moreover, to illustrate that the low-dimensional latent representation of scMGCA preserves the cell-cell topology of the original data, we calculated the paired Pearson

correlations between each pair of cells in the original scRNA-seq data and the data reduced by scMGCA, PCA, t-SNE, and UMAP (Fig. 6b). Obviously, the experimental results indicate that the latent representation of scMGCA on the 'Qx Limb Muscle' preserves the cell-cell topological information and the structure of the original data better than PCA, t-SNE, and UMAP (Fig. 6b).

To explore the functional significances of the latent embedded representation, we selected the top 200 highly expressed genes using the standard deviation and weight of the latent embedded representation (specific selection strategy in Methods), and then performed Gene Ontology (GO) enrichment analysis to detect enriched functional

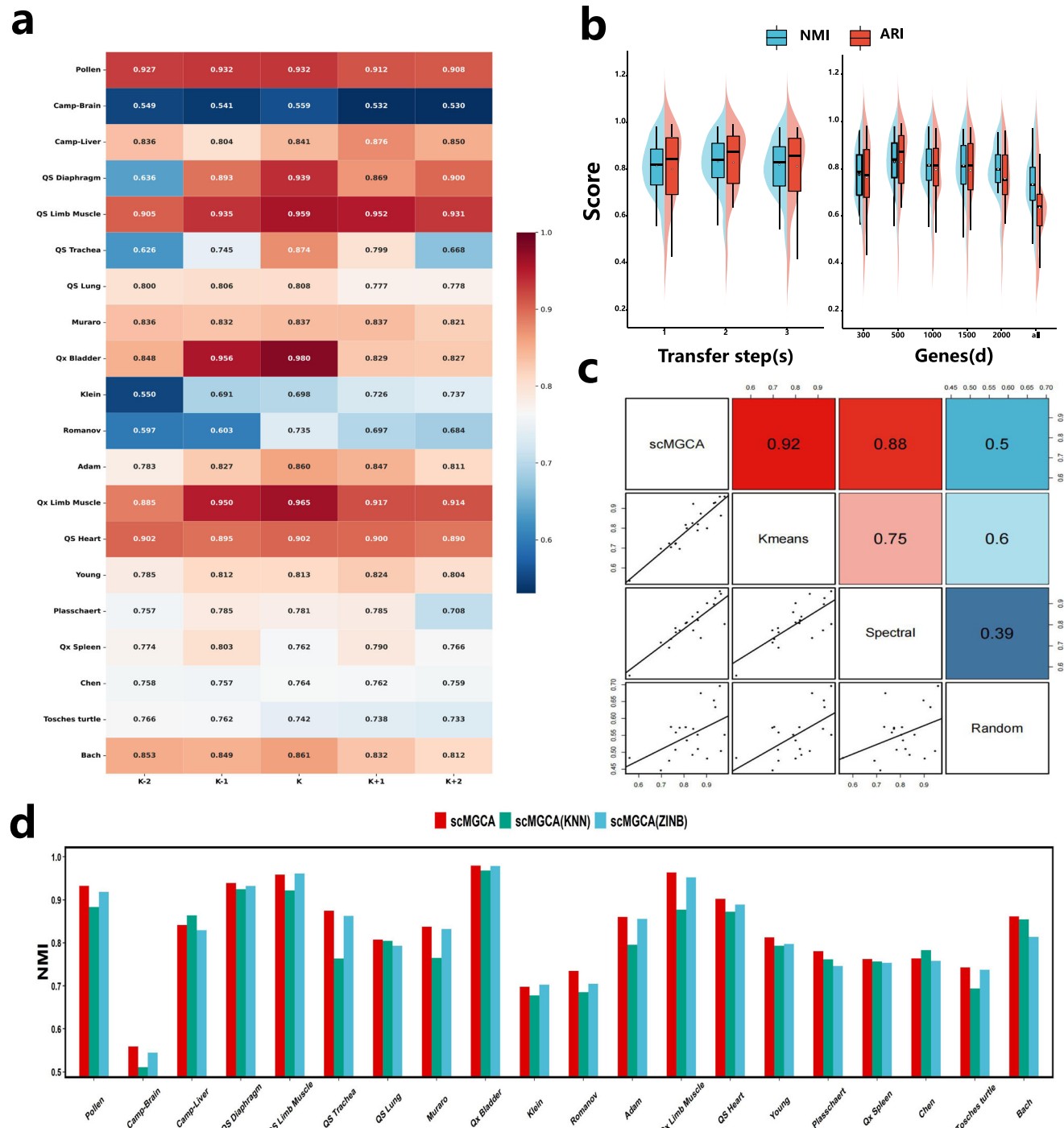

**Fig. 5 | Evaluation of hyperparameter selection and ablation analysis.** The impact of different numbers of **a** cell clusters, **b** transfer steps and selected genes ($n = 20$ in each group; center line, median; box limits, upper and lower quartiles; whiskers, 1.5 × interquartile range), **c** cluster center initialization methods on scMGCA. **d** The ablation study for the cell graph and multinomial-based distribution decoder. Source data are provided as a Source Data file.

attributes based on gene-associated GO terms (Fig. 6c). The GO terms are mainly enriched in the chemotaxis and migration of neutrophils in the biological processes (BPs). Neutrophils are the frontline cells of the innate immune system with an important role in inflammation and tissue wound healing[40,41]. We also performed KEGG pathway analysis to investigate the molecular pathways behind the selected genes (Fig. 6d). We observe that the pathway with the most of the selected genes is Cytokine-cytokine receptor interaction that corresponds to the molecular function (MF) predominantly enriched in cytokine receptor and ligand activity in the GO enrichment analysis results. Muscle cells produce cytokines into the circulation during exercise,

and the control of cytokine production depends on the interactions between various local and systemic factors[42]. Therefore, the results of the KEGG and GO analysis indicate that the genes selected in the latent embedded representation are essential for the binding of receptor ligands in the circulation. Furthermore, we employed STRING[43] to construct the PPI network for the selected genes, and then visualized the network structure showing an interaction score > 0.7 through Cytoscape[44] (Fig. 6e). In particular, Molecular Complex Detection (MCODE)[45] was adopted to identify the most important modules.

On this basis, to explore whether there are marker genes among the selected genes, we used the gene expression matrix of the 'Qx Limb

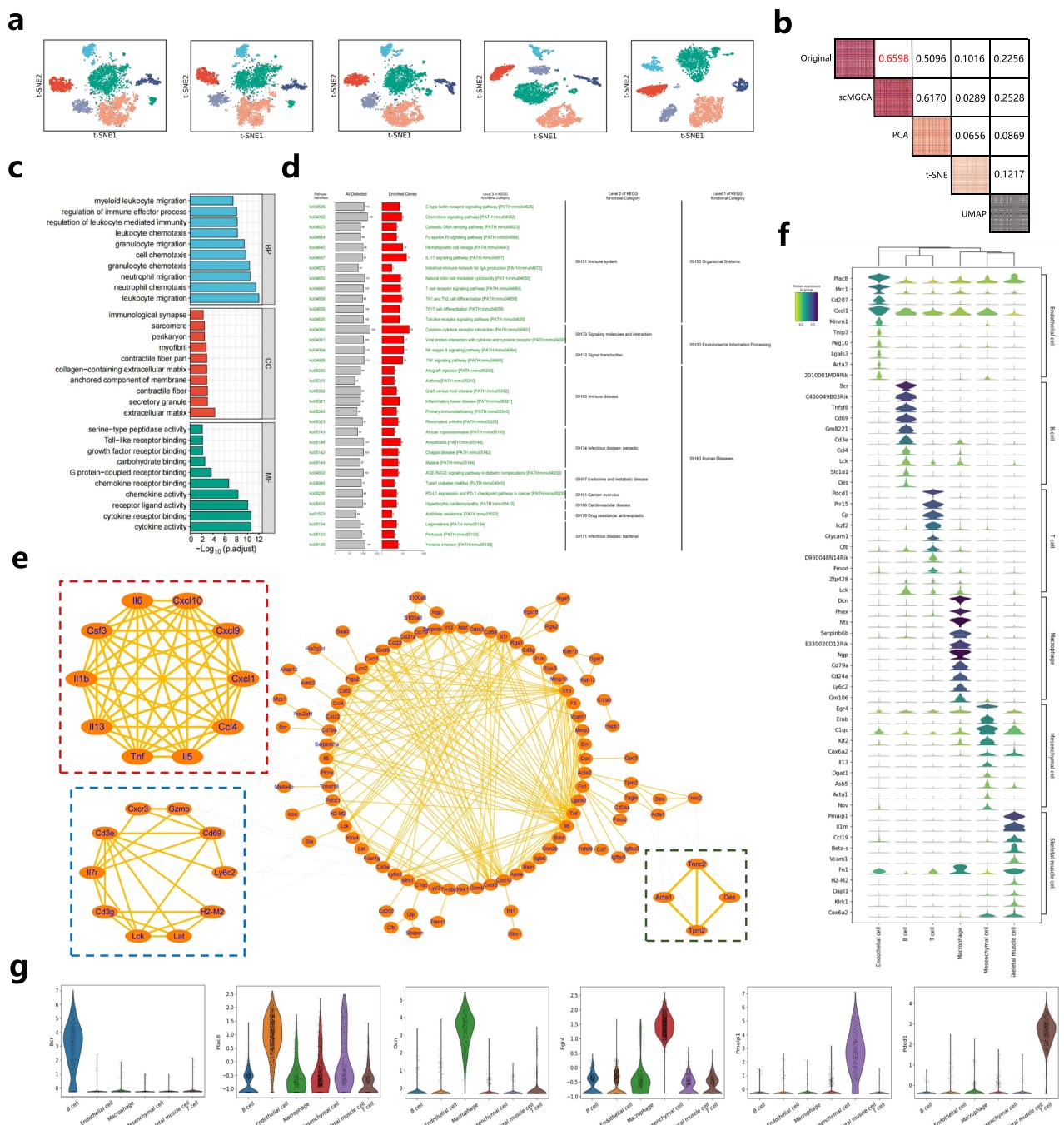

**Fig. 6 | Functional genomics analysis of the latent embedded representation of scMGCA on 'Qx Limb Muscle'. a** 2D-visualization of latent embedded representations of scMGCA at different training periods using t-SNE. **b** Cell topology preservation in the latent embedded representation of scMGCA compared with methods using PCA, t-SNE and UMAP. The upper triangular part shows the Pearson correlation coefficient of the corresponding two spaces, while the diagonal part represents the correlation heatmap of paired cells. **c** GO enrichment analysis (two-sided Wilcoxon test and adopt BH to adjust *p*-values for multiple comparisons), **d** KEGG pathway enrichment analysis, and **e** PPI network of selected genes. **f** Stacked violin plots of the top 10 genes expressed in each cluster. **g** Violin plots of top gene expression levels in each cell cluster. Source data are provided as a Source Data file.

Muscle' dataset and the predicted cluster label of scMGCA to analyze the expression levels of the selected genes in each cell cluster via SCANPY. Fig. 6f depicts the stacked violin plots of the top 10 genes expressed in each cluster, and we provide the detailed distribution of gene expression values of the top most genes of each cell cluster (Fig. 6g). We found that many selected genes serve as cell marker genes, such as *Bcr* for B cells[46] and *Pdcd1* for T cells[47], which further proves that the 200 genes selected in the latent embedded representation have meaningful functional interpretation. In addition, some genes that have not yet been discovered have the potential as new

cellular markers for biologists; for instance, *Egr4* has the highest gene expression level in mesenchymal cells, but not enough studies have been done to determine whether it is actually a cellular marker; therefore, *Egr4* may represent a marker gene for these cells. Similarly, *Pmaip1* could be a candidate marker gene for mouse skeletal muscle cells. In addition, we employed other clustering methods and used the same method to extract functional genes to compare with scMGCA (Supplementary Note 14 and Supplementary Figs. 29, 30). From the experimental results, we observed that scMGCA is able to selects genes that are highly expressed in each cell cluster, while the other clustering

algorithms appeared confused or not found. scMGCA identified differentially expressed genes closer to the genes according to the annotated labels. Furthermore, we also compared scMGCA with SCANPY and Seurat and found that scMGCA was able to detect genomic features which have not been detected by the standard methods (Supplementary Note 15 and Supplementary Fig. 31).

Finally, we applied the Monocle3[48] algorithm to the gene expression matrix consisting of these 200 interpretable genes to analyze the developmental trajectory of the time-series 'Klein'[9] and hESC dataset[49] in pseudo-time (Supplementary Note 16 and Supplementary Fig. 32). From Supplementary Fig. 32, it can be observed that the raw data does not lead to the trajectory path well and reveals the wrong stage of cell differentiation, whereas the cell trajectories inferred by the interpretable genes picked by scMGCA were closely related to the true stages of cell differentiatio accurately. The cell and time trajectories inferred from the interpretable genes selected by scMGCA accurately tracked the stages of cell differentiation, demonstrating that these genes can effectively infer the cell and time trajectories, and further elaborating the effectiveness for selecting interpretable genes.

### scMGCA can elucidate the underlying regulatory mechanisms of pancreatic ductal carcinoma

We applied scMGCA to a pancreatic ductal carcinoma (PDAC) dataset to illustrate its capability for single-cell analysis. The dataset was obtained from a primary PDAC tumor and a control pancreas from CRA001160, and contained a total of 57,530 cells and 18,008 genes[50]. The PDAC data were first preprocessed and clustered by scMGCA (Supplementary Note 17), and then cell types were annotated with marker genes in each cell cluster, as summarized in Supplementary Table 23. Finally, we identified 10 types of cells, including acinar cells, B cells, type 1 ductal cells, type 2 ductal cells, endocrine cells, endothelial cells, fibroblasts, macrophages, stellate cells, and T cells (Fig. 7a). Among them, the largest proportion of type 2 duct cells and the smallest proportion of endocrine cells were found. Notably, type 2 ductal cells were all detected in tumor cells as seen in Fig. 7b. We then visualized the expression levels of the top 10 differential genes in each cell cluster (Fig. 7c). We observe from the heatmap, that differential genes are clearly expressed in each cell cluster, demonstrating that scMGCA is effective in clustering and annotating the dataset. Specially, we also checked for transcription factors (TF) among these genes using RcisTarget[51], and obtained the DNA binding motif with the highest enrichment score for each cell cluster. Further, we interpreted the significant marker gene of each cell cluster in detail, including its expression distribution in all cells (Fig. 7d) and the difference in its expression levels between the cell clusters (Fig. 7e). We also performed pseudo-time analysis of the top 10 genes of each cell cluster and visualized them for 4 clusters (Fig. 7f). We see that the genes of clusters 3 and 4 are highly expressed at the start point and middle point of the pseudo-time course, respectively, while the genes of clusters 1 and 2 are highly expressed at the end point.

Then, we performed pathway enrichment analysis (Fig. 7g). Among all the cell types, the highest enrichment score was for pancreatic secretion; it is consistent with the fact that dysregulated pancreatic duct fluid secretion is the main process related to PDAC. The digestion and absorption of proteins and fats are also the processes significantly enriched in PDAC, as evidenced by other PDAC studies[52–54]. The pancreas contains different cells that produce enzymes important for digestion, including trypsin and chymotrypsin that digest protein, and lipase that breaks down fats. Therefore, the abnormality of the pancreas will indirectly or directly affect the digestion and absorption of proteins and fats. In addition, protein digestion and absorption have been reported to be associated with the development of pancreatic neuroendocrine tumors[55]. Other pathways, such as natural killer cell-mediated cytotoxicity and complement and coagulation cascades, have been confirmed in PDAC-related

research[56,57]. Systematically, we used CellChat[58] to infer cell-cell communication detected by scMGCA. A variety of signaling pathways were found by CellChat, of which the insulin signaling pathway had the highest level of signaling. Due to the plethora of metabolic derangements caused by hyperinsulinemia, researchers believe that insulin signaling plays a potentially decisive role in tumors, including PDAC[59]. From the hierarchical plot of cell interactions of the insulin signaling pathway in Fig. 7h, we observe that endothelial cells, fibroblasts, macrophages, and stellate cells in addition to duct cells, have a high percentage of cells that serve as sources of the insulin signaling pathway. Meanwhile, several studies have indeed demonstrated that the insulin signaling pathway active in these cells plays an important role in the progression of PDAC[60].

To verify whether the signaling pathways identified by the single-cell PDAC data are clinically relevant, we conducted a cohort study adopting TCGA's database public of invasive pancreatic ductal adenocarcinomas and its variants (PAAD) to validate their consistency. Specifically, since the type 2 ductal cells identified by scMGCA were all derived from tumor cells, it is reasonable to assume that type 2 ductal cells and tumors are correlated. On this basis, we performed GSVA enrichment on the type 2 ductal cells identified by scMGCA and 178 tumor samples from the TCGA data, respectively, and then compared them (Supplementary Fig. 33). It can be observed that the pathways enriched from PAAD tumor samples are mostly the same as those enriched from the type 2 ductal cells identified by scMGCA, and these pathways are essentially associated with PDAC. Among them, the co-enriched p53 (signaling) pathway is an important pathway in PDAC. As a sequence-specific transcription factor, the p53 protein is a major tumor suppressor[61]. Under oncogenic stress, p53 is activated to induce multiple programs, including cell cycle exit, apoptosis and replicative senescence, aiming to limit malignant cell proliferation[62]. In addition to this, it was mainly enriched for tumor-related functions such as glycolysis, protein secretion and pathways in cancer, further supporting the malignant state of the type 2 ductal cells. In conclusion, the experimental results indicate that the results of scMGCA analysis of PDAC can be validated by another independent cohort study and that scMGCA can elucidate the potential regulatory mechanisms of PDAC.

## Discussion

scRNA-seq enables the high-throughput measurements on gene expression patterns at the level of individual cells, accounting for cellular heterogeneity. However, the effective cell type annotation from high-dimensional and sparse sequencing data is still challenging by AI or even human. Therefore, it is worth exploring effective extraction and dissemination of cell-cell topology in the data. In this paper, we proposed scMGCA to address these problems. Specifically, we propose a cell-PPMI graph to aggregate neighboring cells through random surfing and a co-occurrence probability PPMI matrix, capturing critical cell-cell topology. Secondly, we employ GCNs under the framework of graph autoencoder to extract and integrate the gene expression information and cell topology, and to preserve the key information in low-dimensional latent embedded representations. Thirdly, we propose a multinomial-based decoder to decode the latent embedded representation and to train the network by optimizing a multinomial-based loss function. This step simulates the dropout events that lead to sparse scRNA-seq data through multinomial distribution, so that the overall model always follows the multimodal distribution for learning. Finally, we propose self-optimizing embedded clustering based on KL divergence on the low-dimensional representations, and simultaneously optimize the clustering loss and the other loss functions in the network.

To demonstrate that scMGCA can accurately cluster cells, we compared it with 12 single-cell clustering methods. The experimental results show that scMGCA has the best clustering performance and outperforms the other 12 comparison methods for data across

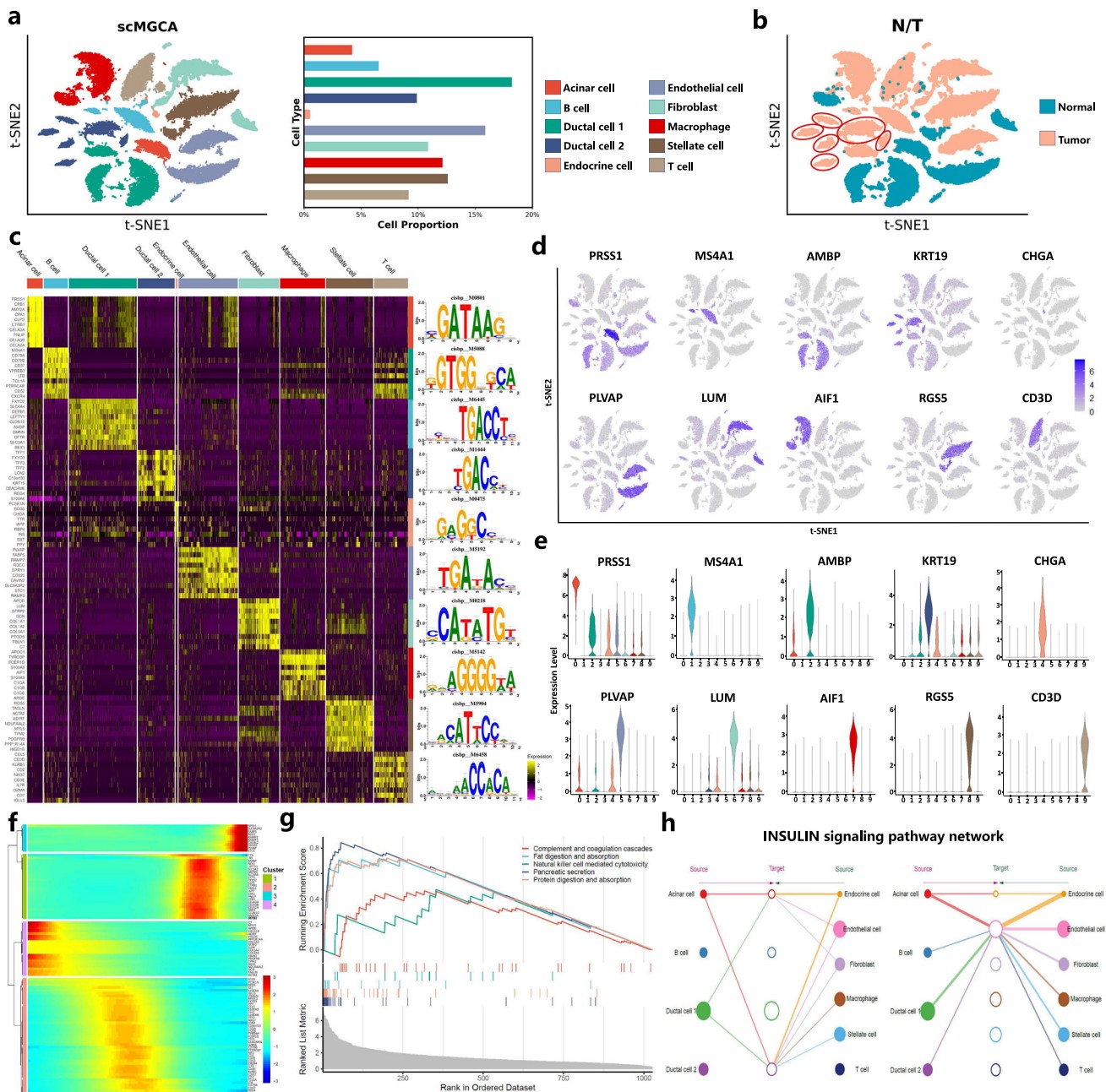

**Fig. 7 | PDAC dataset (CRA001160) analysis by scMGCA. a** Visualization of cell clusters identified by scMGCA via t-SNE (left) and the proportion of different cell types (right). **b** Visualization of the distribution of normal cells and cancer cells via t-SNE. **c** Heatmap of top 10 marker gene expression levels for each cell cluster, and the corresponding DNA binding motif for each cell cluster. **d** Visualization of expression distribution of the top marker gene of each cell cluster. **e** Violin plot of expression levels of top marker genes in each cell cluster. **f** Heatmap for pseudo-time analysis of marker genes. **g** GSEA on KEGG pathways for PDAC. **h** Hierarchical plot of communication between cell clusters in the Insulin signaling pathway. Source data are provided as a Source Data file.

multiple platforms, demonstrating the cross-platform capability of scMGCA in the wet-lab sense. In particular, some rare cell types and small clusters that cannot be picked up by other methods can be detected by scMGCA. For model scalability, scMGCA is able to successfully cluster 1.3 million datasets with reasonable time and memory consumption, which can be seen as an advancement in deep graph learning on large-scale datasets. scMGCA can also effectively reduce dimensionality and visualize blue the single-cell data. In experiments compared with three dimensionality reduction methods and two single-cell analysis packages, the ASW obtained for dimensionality reduction by scMGCA is the highest and the visualized results are close to the distribution of the real clusters. In addition, scMGCA is able to

correct for the batch effect generated by data from different scRNA-seq protocols, outperforming other state-of-the-art batch effect correction methods.

Meanwhile, scMCGA presents a new interpretable approach for functional genomics. We propose a ranking and screening method based on the standard deviation of the parameter matrix in the GCN encoder to select 200 highly expressed genes from the latent embedding representation in scMGCA. Through the functional analysis of these genes, we found that the latent embedding representation of scMGCA is biologically interpretable without any major sacrifices on its cell clustering performance. In addition, scMGCA is able to discover significantly expressed differential genes in each cell cluster, which

other deep learning methods cannot, and discover potential marker genes in some cell clusters that were not detected by standard methods. These experimental results not only verify that the underlying embedding representation of scMGCA is biologically interpretable, but also provide a new way for deep graph clustering to explore biological interpretability.

Moreover, scMGCA can also elucidate the underlying regulatory mechanisms of pancreatic ductal carcinoma (PDAC). We applied scMGCA to PDAC data for cell clustering and defined cell types by marker genes in the cell clusters. Interestingly, all type 2 ductal cells identified by scMGCA were derived from tumor cells. Furthermore, we found that type 2 ductal cells were enriched for several pathways essentially associated with PDAC, and most of them were consistent with the pathways enriched in tumor samples from clinical data. This demonstrates the ability of scMGCA to elucidate the underlying regulatory mechanisms of complex diseases and deliver biologically meaningful results.

In summary, scMGCA discards the traditional way of constructing cell graphs and jointly analyzes single-cell data through graph-embedding representation and multinomial distribution simulation. As a new graph-learning method for single-cell biology, can simultaneously perform cell clustering, dimensionality reduction, batch effect correction, million-level data analysis, interpretability analysis and single-cell data analysis of joint clinical data, which is rare in the current field of graph neural networks. In future work, we will continue to improve and expand scMGCA.

## Methods

### Data processing and normalization

We take the scRNA-seq gene expression count matrix $\mathbf{X} \in \mathbf{R}^{n \times g}$ as input, where $n$ and $g$ denote the number of cells and genes, respectively. The first step is to filter out genes that are not expressed in cells, reducing the impact of the high dropout rates in scRNA-seq data. Then, to facilitate training and optimization of the network, we convert the discrete data into continuous data and ensure the stability of the size factor, the normalization can be defined as follows:

$$\mathcal{H}(X_{ij}) = \ln\left(\text{median}(\mathbf{X})\frac{X_{ij}}{\sum_k X_{ik}}\right) \quad (1)$$

where median($\mathbf{X}$) denotes the median of the total expression values of all cells. According to Eq. (1), the discrete data is smoothed and rescaled by natural log transformation. Finally, we adopt the SCANPY package[13] to calculate the normalized dispersion values of each gene to rank them, and the top $d$ highly variable genes with high-level information are used in the study, that is, the final normalized matrix $\widehat{\mathbf{X}} \in \mathbf{R}^{n \times d}$.

### Cell graph

In this study, we develop a cell graph to represent the relationship between cells, inspired by a co-occurrence probability PPMI matrix[63]. The cell graph is first constructed from a KNN graph, where each node represents a cell. Through computing the Euclidean distance between nodes, and setting the number of neighbors $k$, each node is connected to the node within $k$ shortest distance including itself, and creates an edge between them, and the cell graph $\mathbf{K}$ can be defined below:

$$K_{ij} = \begin{cases} 1, & j \in \mathcal{N}_k(i) \\ 0, & j \notin \mathcal{N}_k(i) \end{cases} \quad (2)$$

where $\mathcal{N}_k(i)$ represents the neighborhood containing the nearest $k$ nodes with the $i$-th node as the center. It can be clearly seen that the KNN graph determines the domain of each node and its neighbor nodes through a local search method, which is a more effective way to model biological networks[64,65].

After that, we apply the random surfing model[66] to the KNN graph to further explore the potentially complex non-linear relationships between nodes in the domain. We introduce a transition matrix $\mathcal{P}^s$, where $\mathcal{P}^s_{ij}$ represents the probability of the $i$-th node reaching the $j$-th node after $s$ steps of transitions, and $\mathcal{P}^0$ is the initial transition matrix equal to the identity matrix ($\mathcal{P}^0 = \mathcal{I}$). The probability of returning to the initial vertex and restarting the process is set to $(1 - \alpha)$, the transition matrix iteration process can be defined as follows:

$$\mathcal{P}^s = \alpha \cdot \mathcal{P}^{s-1}\mathbf{K} + (1 - \alpha)\mathcal{P}^0 \quad (3)$$

Empirically, we set the transfer step $s$ to two steps ($s \le 3$), where it has been proved that the third-order neighborhood covers almost the entire biological network, and the neighborhood was not very informative for nodes with more than three hops[67,68]. To further explore Eq. (3), we can update the rule through the initial transition matrix $\mathcal{P}^0$:

$$\mathcal{P}^s = \alpha^s \cdot \mathcal{P}^0\mathbf{K}^s + (1 - \alpha)\sum_{t=1}^{s} \alpha^{s-t} \cdot \mathcal{P}^0\mathbf{K}^{s-t} \quad (4)$$

After $s$ steps, the probability matrix is $\mathbf{U} = \sum_{s=1}^{S} \mathcal{P}^s$. To augment the graph, we transform the $\mathbf{U}$ after a random surfing model into a positive pointwise mutual information (PPMI) matrix. The main method of augmentation is based on calculating the pointwise mutual information (PMI) of each node and measuring their correlation, which is defined below.:

$$PMI(x,y) = \log\left(\frac{p(x,y) \cdot |\mathbf{S}|}{p(x)p(y)}\right) \quad (5)$$

$$PPMI(x,y) = \max(PMI(x,y),0) \quad (6)$$

where $|\mathbf{S}| = \sum_x \sum_y p(x,y)$, $x$, and $y$ represent different nodes; $p(x)$ and $p(y)$ denote the probability of $x$ and $y$, respectively; $p(x,y)$ represents the co-occurrence probability of $x$ and $y$. According to probability theory, if $x$ and $y$ are more related, the ratio of $p(x,y)$ and $p(x)p(y)$ is larger. Therefore, the cells of the same type in the biological network have higher weights in the cell graph. Finally, we can approximately yield the PPMI matrix as the cell graph $\mathbf{A}$, which is defined below:

$$\mathbf{A} = \max\left(\log\left(\frac{\mathbf{U}\Theta}{\text{row}(\mathbf{U})\text{col}(\mathbf{U})}\right),0\right) \quad (7)$$

where $\Theta$ denotes the sum of all elements in $\mathbf{U}$. row($\mathbf{U}$) and col($\mathbf{U}$) are the column vector of the sum of each row of $\mathbf{U}$ and the row vector of the sum of each column of $\mathbf{U}$, respectively.

### Multinomial-based graph convolutional autoencoder

To better capture the cell graph structure and node relationships of scRNA-seq data, we propose a graph convolutional autoencoder based on multinomial distribution model. The pipeline of the model is to first preserve the main features and structure of the count matrix data and cell graph data into a low-dimensional space through graph convolution network (GCN), and then we use two parallel decoders to reconstruct the graph and extract the data information in the latent embedded representation; respectively, the inner product and multinomial-based decoder.

We take the pre-processed gene expression matrix $\widehat{\mathbf{X}}$ and the cell graph $\mathbf{A}$ as inputs. The encoder performs convolution operation through two-layers of GCN, and preserves the important information of $\widehat{\mathbf{X}}$ and $\mathbf{A}$ into the latent embedded representation $\mathbf{Z}$, which is defined

as follows:

$$Z = \sigma(\overline{A}\sigma(\overline{A}\widehat{X}W_1)W_2) \tag{8}$$

where $\sigma(\cdot) = \max(0,\cdot)$ denotes a RELU activation function; $\overline{A} = D^{-\frac{1}{2}}AD^{-\frac{1}{2}}$ is normalized cell graph, and $D = diag\{(I+A)1_N\}$ is the degree matrix; $W_l$ is the parameter matrix learned by $l$-th layer. The inner product decoder decodes the reconstructed cell graph information from $Z$, which is defined as follows:

$$\widehat{A} = \text{sigmoid}(Z^T Z) \tag{9}$$

where $\widehat{A}$ is the reconstructed cell graph $A$, $\text{sigmoid}(\cdot) = 1/(1+e^-)$ is sigmoid activation function. Therefore, we can define a reconstruction loss $\mathcal{L}_r$ to minimize it during training as follows:

$$\mathcal{L}_r = \frac{1}{n^2}\sum_{i=1}^{n}\sum_{j=1}^{n}||A_{ij} - \widehat{A}_{ij}||_2^2 \tag{10}$$

where $n$ represents the total number of nodes.

On the other hand, we employ a multinomial-based decoder to decode the data matrix information. We define $n_i = \sum_{j=1}^{d}\widehat{X}_{ij}$ to represent the total gene expression of the $i$-th cell and $m_{ij}$ denotes the unknown true relative abundance of the $j$-th gene expression in the total gene expression of the $i$-th cell, that is, $E(\widehat{X}_{ij}) = n_i m_{ij}$. Therefore, the vector $\widehat{X}_i = (\widehat{X}_{i1}, \widehat{X}_{i2}, ..., \widehat{X}_{id})$ follows a multinomial distribution, which can be defined as follows:

$$f(\widehat{X}_i|m) = \frac{n_i!}{\widehat{X}_{i1}!\widehat{X}_{i2}!\ldots\widehat{X}_{id}!}\prod_{j=1}^{d}m_{ij}^{\widehat{X}_{ij}} \tag{11}$$

To address dropout events, we propose using a binary random variable $U_{ij}$ for simulation, where $U_{ij} = 0$ means that the $j$-th gene is dropped in the $i$-th cell. Let $\pi_{ij} = P(U_{ij} = 1)$, we have $U_{ij} \sim Bernoulli(\pi_{ij})$. Theoretically, a high dropout rate implies low gene expression, and it can be inferred that $\pi_{ij}$ is positively correlated with $m_{ij}$. Assuming that $V_{ij}$ is the expected relative expression level of the $j$-th gene in the $i$-th cell, and $m_{ij}$ should be multiplied by $\pi_{ij}$ and $V_{ij}$ during the dropout process and normalized as below:

$$m_{ij} = \frac{\pi_{ij}V_{ij}}{\sum_{j=1}^{d}\pi_{ij}V_{ij}} \tag{12}$$

Obviously, $m$ cannot be calculated completely by statistical theory, hence we learn from the low-dimensional representation in the deep neural network according to Eq. (12), which is defined as follows:

$$M = \text{softmax}(W_m f_d(Z)) \tag{13}$$

where $f_d$ is a three-layer fully connected neural network with 128, 256 and 512 nodes in the hidden layers, respectively; $W_m$ represents the learned weights and $M$ is the parameter matrix. Given the latent embedded representation $Z$, $\widehat{X}$ can be regarded as conditionally independent, the negative log-likelihood of the multinomial distribution can be constructed as the loss of data matrix reconstruction as below:

$$\mathcal{L}_m = -\log\prod_{1=1}^{n}\frac{n_i!}{\widehat{X}_{i1}!\widehat{X}_{i2}!\ldots\widehat{X}_{id}!}\prod_{j=1}^{d}m_{ij}^{\widehat{X}_{ij}} \tag{14}$$

$$\propto -\sum_{i=1}^{n}\sum_{j=1}^{d}\widehat{X}_{ij}\log m_{ij} \tag{15}$$

## Deep embedded clustering with latent representation

To obtain better clustering performance, we present a self-training clustering method on the latent embedded representation of the graph convolutional autoencoder. The loss function takes the form of Kullback-Leibler (KL) divergence, which is defined as follows:

$$\mathcal{L}_c = KL(P||Q) = \sum_i\sum_u p_{iu}\log\frac{p_{iu}}{q_{iu}} \tag{16}$$

where $q_{iu}$ is the soft label of the embedded node $z_i$, using Student's $t$-distribution to calculate the similarity between $z_i$ and the cluster center $\mu_u$, which can be described as follows :

$$q_{iu} = \frac{(1+||z_i - \mu_u||^2)^{-1}}{\sum_k(1+||z_i - \mu_k||^2)^{-1}} \tag{17}$$

For the initial clustering center $\{\mu\}$, we use kmeans clustering based on data distribution and spectral clustering based on graph structure to obtain the results, and select the best ones for training. In addition, $p_{iu}$ is an auxiliary target distribution, to improve the high confidence in $q_{iu}$, which can be formulated as follows:

$$p_{iu} = \frac{q_{iu}^2/\sum_i q_{iu}}{\sum_k(q_{ik}^2/\sum_i q_{ik})} \tag{18}$$

Noticeably, the target distribution $P$ is defined on the basis of $Q$, and $Q$ approximates the target distribution $P$ in a self-optimizing manner to obtain better clustering performance.

## Joint embedding and clustering optimization

The training strategy of scMGCA is to jointly optimize the latent embedded representation of the graph convolutional autoencoder and cluster tasks, minimizing the following objective function:

$$\mathcal{L} = \gamma_1\mathcal{L}_r + \gamma_2\mathcal{L}_m + \gamma_3\mathcal{L}_c \tag{19}$$

where $\mathcal{L}_r$ denotes the cell graph reconstruction loss; $\mathcal{L}_m$ represents the loss of multinomial-based decoder; $\mathcal{L}_c$ is the embedded clustering loss; $\gamma_1, \gamma_2$ and $\gamma_3$ are weight coefficients assigned to each loss to control the balance of the total objective function. Stochastic gradient descent (SGD) and back propagation are used jointly to optimize the graph autoencoder and clustering centers. The gradient of the clustering loss $\mathcal{L}_c$ relative to the potential embedded node $z_i$ and the cluster center $\mu_u$ can be calculated as:

$$\frac{\partial\mathcal{L}_c}{\partial z_i} = 2\sum_u(1+||z_i - \mu_u||^2)^{-1}(p_{iu} - q_{iu})(z_i - \mu_u) \tag{20}$$

$$\frac{\partial\mathcal{L}_c}{\partial\mu_u} = -2\sum_i(1+||z_i - \mu_u||^2)^{-1}(p_{iu} - q_{iu})(z_i - \mu_u) \tag{21}$$

Given the learning rate $l_r$, the cluster center $\mu_u$ can be updated as follows:

$$\mu_u = \mu_u - \frac{l_r}{n}\sum_{i=1}^{n}\frac{\partial\mathcal{L}_c}{\partial\mu_u} \tag{22}$$

where $n$ denotes the total number of nodes. The polynomial coefficient matrix $W_e$ of scMGCA in encoder and the weight matrix $W_d$ in the

multinomial-based decoder are updated as follows:

$$\mathbf{W}_e = \mathbf{W}_e - \frac{\gamma_2 l_r}{n} \sum_{i=1}^{n} \frac{\partial \mathcal{L}_m}{\partial \mathbf{W}_e} \qquad (23)$$

$$\mathbf{W}_d = \mathbf{W}_d - \frac{l_r}{n} \sum_{i=1}^{n} \left( \gamma_1 \frac{\partial \mathcal{L}_r}{\partial \mathbf{W}_d} + \gamma_2 \frac{\partial \mathcal{L}_m}{\partial \mathbf{W}_d} + \gamma_3 \frac{\partial \mathcal{L}_c}{\partial \mathbf{W}_d} \right) \qquad (24)$$

The stopping condition is that the maximum number of iterations is achieved. Then, we can obtain the predicted clustering assignment of each cell via the $\mathbf{Q}$ after model training.

## Functional genomics interpretation of gene findings from latent embedded representations

In scMGCA, the normalized cell graph $\overline{\mathbf{A}}$ and scRNA-seq gene expression matrix $\hat{\mathbf{X}}$ preserve the cell-cell topology information into a low-dimensional latent embedded representation $\mathbf{Z}$ through two layers of GCN, as shown in Equation (8). Among them, the $d$-dimensional data is reduced to 15-dimensional at the beginning, and $d$ is the number of genes after pre-processing. Obviously, the weight matrix $\mathbf{W}$ contains the key information of these genes, as it plays a leading role in dimensionality reduction and is involved in model training. Therefore, we focus on the weight matrices $\mathbf{W}_1$ and $\mathbf{W}_2$, extracting the key information through the variance from the direction of $\mathbf{Z}^{n \times 15}$ to $\hat{\mathbf{X}}^{n \times d}$, and finally finding the biologically explanatory genes. In the remaining part, we introduce the specific computational process.

First, the latent embedded representation $\mathbf{Z}^{n \times 15}$ is derived from the weight matrix $\mathbf{W}_2^{128 \times 15}$ and the first-layer GCN output $\hat{\mathbf{X}}_1^{n \times 128}$. At this time, the key information of all genes contained in 15 dimensions of $\mathbf{Z}$ can be captured from $\mathbf{W}_2$. Therefore, we calculate the standard deviation of $\mathbf{W}_2$ and obtain a 128-dimensional result $r_1^{128 \times 1}$. Then, we select the top 50% ($128 \times 50\% = 64$) of highest variance from $r_1^{128 \times 1}$, and save their index $l_1^{1 \times 64}$. The $\hat{\mathbf{X}}_1^{n \times 128}$ is derived from the weight matrix $\mathbf{W}_1^{d \times 128}$ and the $\hat{\mathbf{X}}^{n \times d}$. In this step, we filter and retain the columns with high variance in $\hat{\mathbf{X}}_1^{n \times 128}$ (containing essential gene information) according to $l_1$, and finally we get $\overline{\mathbf{W}}_1^{d \times 64}$. We also perform the previous operation on $\overline{\mathbf{W}}_1^{d \times 64}$, that is, calculate the standard deviation, to obtain a $d$-dimensional result $r_2^{d \times 1}$. Finally, we extract the top 200 highly variable genes from $r_2^{d \times 1}$ and save the index $l_2^{1 \times 200}$, and then find 200 pivotal genes with biological explanations from $\hat{\mathbf{X}}^{n \times d}$ based on the index $l_2^{1 \times 200}$.

## Implementation details and comparisons with existing methods

scMGCA is implemented in Python, and the core model is built on the Tensorflow framework that is publicly available at https://pypi.org/project/scMGCA. To evaluate the performance of our scMGCA against other current scRNA clustering methods, we compared scMGCA with the following methods:

- Deep self-training K-means clustering method (scziDesk, https://github.com/xuebaliang/scziDesk)[26]. scziDesk combines the deep learning method and a soft self-training K-means algorithm to aggregate scRNA-seq data in a latent space.
- Single-cell model-based deep learning method (scDeepCluster, https://github.com/ttgump/scDeepCluster)[17]. scDeepCluster is a deep clustering method based on a single-cell model that uses domain knowledge to learn clustering and feature representation simultaneously.
- Deep count autoencoder network (DCA, https://github.com/theislab/dca)[16]. DCA takes into account the overdispersion and count distribution of scRNA-seq data and uses a zero-inflated negative binomial noise model.

- Unsupervised deep embedding clustering method (DEC, https://github.com/XifengGuo/DEC-keras)[27]. DEC applies a deep neural network to simultaneously learn cluster assignments and feature representations in a lower-dimensional space.
- Unsupervised deep embedding algorithm (DESC, https://github.com/eleozzr/desc)[18]. DESC clusters scRNA-seq data by iteratively optimizing the clustering objective function.
- Single-cell variational inference (scVI, https://github.com/YosefLab/scVI)[19]. scVI uses stochastic optimization and deep neural networks to aggregate information from similar cells and genes and approximate the distribution of the observed expression values, taking into account batch effects and limited sensitivity.
- Deep scRNA-seq clustering method via convolutional autoencoder and soft K-means (scCAEs, https://github.com/gushenweiz/scCAEs)[20]. scCAEs learns feature representation and clustering for scRNA-seq based on convolutional autoencoder embeddings and soft K-means.
- Single-cell graph neural network (scGNN, https://github.com/juexinwang/scGNN)[23]. scGNN provides a deep learning framework that formulates cell-cell relationships and applies a mixture Gaussian model to learn gene expression patterns.
- Single-cell graph autoencoder (scGAE, https://github.com/ZixiangLuo1161/scGAE)[28]. scGAE applies a multi-task oriented graph autoencoder to maintain the feature information and structural information of scRNA-seq data.
- GCN-based single-cell clustering model (GraphSCC, https://github.com/GeniusYx/GraphSCC)[24]. GraphSCC clusters cells by using a graphical convolutional network to illustrate the structural relationships between cells.
- R toolkit for single-cell genomics (Seurat, https://github.com/satijalab/seurat). Seurat is an R package design for the analysis and exploration of single cell RNA-seq data.
- Single-cell RNA-seq hyper-fast and accurate processing via ensemble random projection (SHARP, https://github.com/shibiaowan/SHARP). SHARP is a bioinformatics tool for processing and analyzing single-cell RNA-seq (scRNA-seq) data.

## Benchmarking metrics for scMGCA clustering and visualization

In this paper, we employed two widely used evaluation measures, normalized mutual information (NMI) and adjusted rand index (ARI), to estimate the performance of the different computational methods. Davies-Bouldin (DBI) and V-Measure are also introduced for a more comprehensive analysis when comparing to SCANPY and Seurat.

NMI measures the normalized mutual information between predicted labels and true labels. Let $l$ denote the predicted cluster labels and $l^G$ denote the ground-truth labels. The NMI is defined as:

$$NMI(l, l^G) = \frac{\sum_{i=1}^{n'} \sum_{j=1}^{n^G} n_{ij} \log \frac{n_{ij} n}{n_{i'} n_j^G}}{\sqrt{\sum_{i=1}^{n'} n_{i'} \log \frac{n_{i'}}{n} \sum_{j=1}^{n^G} n_j^G \log \frac{n_j^G}{n}}} \qquad (25)$$

ARI evaluates the similarity of two clustering results in a statistical way, which can be defined as:

$$ARI\left(l, l^G\right) = \frac{\sum_{i=1}^{n'} \sum_{j=1}^{n^G} \binom{n_{ij}}{2} - \sum_{i=1}^{n'} \binom{n_{i'}}{2} \cdot \sum_{j=1}^{n^G} \binom{n_j^G}{2} / \binom{n}{2}}{\sum_{i=1}^{n'} \binom{n_{i'}}{2} / 2 + \sum_{j=1}^{n^G} \binom{n_j^G}{2} / 2 - \sum_{i=1}^{n'} \binom{n_{i'}}{2} \cdot \sum_{j=1}^{n^G} \binom{n_j^G}{2} / \binom{n}{2}} \qquad (26)$$

where $n'$ is the number of clusters in the predicted clustering $l$, $n^G$ is the number of clusters in the ground-truth clustering $l^G$, $n_{i'}$ is the number of data objects in the $i$-th cluster of $l$, $n_j^G$ is the number of data objects in

the $j$-th cluster of $l^G$, and $n_{ij}$ is the number of data objects that belong to both cluster $i$ in $l$ and cluster $j$ in $l^G$.

We also adopt the average silhouette width (ASW) to test our method for dimensionality reduction and visualization, which is defined as below:

$$ASW(x) = \frac{1}{n}\sum_{i=1}^{n}\left(\frac{b(i)-a(i)}{max\{a(i),b(i)\}}\right) \qquad (27)$$

where $a(i)$ denotes the average distance from $x_i$ to all the other data points in the cluster to which $x_i$ belongs, and $b(i)$ denotes the minimum average distance from $x_i$ to all other clusters to which $o$ does not belong. The value of ASW is between $[-1, 1]$. If ASW is close to 1, it means that the clustering of the data object $x$ is reasonable; if ASW is close to -1, it means that the division of $x$ is inaccurate; if ASW is -0, it implies that many data points in $x$ are on the boundary of the two clusters. The ASW is a measure of the reasonableness and validity of the clustering results.

DBI measures the mean of the maximum similarity of each cluster, which can be formulated as:

$$DBI = \frac{1}{N}\sum_{i=1}^{N}\max_{j\neq i}\left(\frac{\overline{S_i}+\overline{S_j}}{E_{ij}}\right) \qquad (28)$$

where $N$ is the number of clusters, $\overline{S_i}$ represents the average distance from the data in the $i$-th cluster to the cluster centroid, and $E_{ij}$ is the Euclidean distance between the i-th cluster and the j-th cluster. The smaller the DBI value, the closer the clustering results are to the inside of the cluster, indicating that the clustering effect is better.

V-Measure is based entirely on the conditional entropy between two clusters, which can be defined as:

$$v = \frac{2hc}{h+c} = \frac{2\left(1-\frac{H(\mathbf{C}|\mathbf{K})}{H(\mathbf{C})}\right)\left(1-\frac{H(\mathbf{K}|\mathbf{C})}{H(\mathbf{K})}\right)}{\left(1-\frac{H(\mathbf{C}|\mathbf{K})}{H(\mathbf{C})}\right)+\left(1-\frac{H(\mathbf{K}|\mathbf{C})}{H(\mathbf{K})}\right)} \qquad (29)$$

where $\frac{H(\mathbf{C}|\mathbf{K})}{H(\mathbf{C})}$ is the conditional entropy of class division given the cluster division condition. $h$ and $c$ represent homogeneity and completeness measures, respectively. The V-Measure value $v$ is the harmonic mean of the $h$ and $c$, and the value is larger when the two categories are closely separated.

## Functional enrichment

We use the R package ClusterProfiler[69] to perform KEGG and GO enrichment analysis on the 200 genes selected from the latent embedded representation. We utilize the function 'enrichKEGG' for the KEGG analysis, where the parameters are set to "pAdjustMethod = fdr, pvalueCutoff = 0.01, qvalueCutoff = 0.05". The GO analysis is performed by function 'enrichGO', where the parameters are set to "ont = ALL, pAdjustMethod = BH, pvalueCutoff = 0.01, qvalueCutoff = 0.05".

## PPI network

We performed preliminary PPI network construction on the selected 200 genes using STRING[43] and extracted the part of the network with high interaction scores (>0.7) as input to Cytoscape for visualization. To identify the most important modules in the PPI network, we adopt MCODE[45] for the network and give the top three modules.

## Trajectory inference

The trajectory analysis was performed in the R package, Monocle3 (http://cole-trapnell-lab.github.io/monocle-release/monocle3/). In Supplementary Fig. 32, we extracted an expression matrix of 200 biologically interpretable genes as input and used the labels predicted by scMGCA clustering. After the monocle loads the data by function

'new_cell_data_set', the data is preprocessed by function 'pre-process_cds'. Then, the data is dimensionally reduced by function 'reduce_dimension', where the parameters are set to "reduction_method = UMAP, preprocess_method = PCA". Further, the cells are clustered by function 'cluster_cells', where the parameters are set to "reduction_method = UMAP", and the principal graph is learned by function 'learn_graph', where the parameters are set to "use_partition = F, close_loop = F, learn_graph_control = NULL, verbose = FALSE". Finally, the function 'plot_cells' is used to visualize the trajectory inference. In Fig. 7f, the heatmap of the pseudo-time analysis adopted the function 'plot_pseudotime_heatmap'. Monocle introduces a strategy of sorting individual cells in pseudo-time, using the asynchronous process of individual cells to place them on corresponding trajectories for biological processes such as cell differentiation.

## Identification of marker genes in cell clusters

We used the function 'FindAllMarkers' in the R package Seurat[12] to identify marker genes, where the parameters are set to "only.pos = True, min.pct = 0.25". Then, biological marker genes of each cell cluster were selected from the obtained genes with p_val_adj $\leqslant$ 0.05 and avg_logFC $\geqslant$ 0.5. Finally, the expression levels of biological marker genes in different cell clusters are visualized by the function 'DoHeatmap'.

## TF-motif analysis

TF motifs were found by the R package RcisTarget[51] with default parameters. For the analysis of human cancer samples, hg19 was used as the reference TF. By inputting the marker genes of each cell cluster into the function 'cisTarget', where the parameters are set to "motifAnnot = motifAnnotations_hgnc", the enriched motif corresponding to each cell cluster is obtained, and then the motif with the highest enrichment score is selected.

## Gene set enrichment analysis (GSEA)

GSEA is performed on the basis of biological marker genes, and the main process adopts the R package ClusterProfiler[69]. First, the genes obtained from the function 'FindAllMarkers' were converted by the function 'bitr', that is from SYMBOL to ENTREZID, and then the converted genes and their avg_log2FC are input into the function 'gse-KEGG' for enrichment analysis. Finally, the visualization of the results is performed by the function 'gseaplot2'.

## Gene set variation analysis (GSVA)

We mainly use the function 'gsva' from the R package GSVA, where the parameters are set to "method = ssgsea, abs.ranking = T". The related pathway analysis was used to assess the activation of hallmark and metabolic pathways, which is available in the MSigDB databases[70].

## CellChat analysis

CellChat analysis was performed using the R package CellChat[58]. First, we used the function 'createCellChat' to create a CellChat object for the overall gene expression matrix, and then addded the cell annotation information identified by scMGCA through the function 'addMeta'. We used CellChatDB.human provided by CellChat as the human ligand receptor reference. Finally, the cell-cell communication probability is inferred using the function 'computeCommunProb', and the communication probability at the level of each cell signaling pathway is inferred by the function 'computeCommunProbPathway'. The figure is generated with the function 'netVisual_aggregate', where the parameter is set to "layout = hierarchy".

## Statistics and reproducibility

The detailed statistical tests were explained in each figure legend. Sample data were obtained from public repositories. Sample size was not predetermined and is the maximum number of samples available

for each datasets. No data were excluded from the analyses. No experimental groups were assigned. Our study does not involve group allocation that requires blinding. To reproduce the results, please find the Source Data file we provided.

## Reporting summary

Further information on research design is available in the Nature Portfolio Reporting Summary linked to this article.

## Data availability

We collected 20 real and public scRNA-seq datasets across multiple platforms. These datasets come from different organs in humans and mouse, such as the brain, pancreas, trachea, and other. These datasets include 'Pollen' (SRP041736), 'Camp-Brain' (GSE75140), 'Camp-Liver' (GSE81252), 'Muraro' (GSE85241), 'Klein' (GSE65525), 'Romanov' (GSE74672), 'Adam' (GSE94333), 'Young' (EGAS00001002171), 'Plasschaert' (GSE102580), 'Chen' (GSE87544), 'Tosches turtle' (PRJNA408230), 'Bach' (GSE106273), and so on. The remaining 'QS' and 'Qx' datasets are derived from mouse scRNA-seq data generated by Smart-seq2 sequencing and 10x Genomics sequencing, respectively, in the Stanford University study[71]. The details of the 20 scRNA-seq datasets including the number of cells, number of cell types, and organs are tabulated in Supplementary Table 24, and can be downloaded from https://github.com/Philyzh8/scMGCA and https://hemberg-lab.github.io/scRNA.seq.datasets. The dataset 'Tabula Muris' (GSE109774) can be downloaded from https://doi.org/10.6084/m9.figshare.5968960.v3. The 1.3 million mouse brain cells dataset contains 1,306,127 cells and 27,998 genes, downloaded from the 10x Genomics website: https://support.10xgenomics.com/single-cell-gene-expression/datasets/1.3.0/1M_neurons. The human pancreatic islet datasets are generated from four different scRNA-seq protocols, including CEL-seq (GSE81076)[33], CEL-seq2 (GSE85241)[34], Fluidigm C1 (GSE86469)[35], and Smart-seq2 (E-MTAB-5061)[36]. The hESC dataset (GSE75748)[49] can be downloaded from https://www.ncbi.nlm.nih.gov/geo/query/acc.cgi?acc=GSE75748. The accession number for the PDAC data reported in this paper is GSA: CRA001160. These data have been deposited in the Genome Sequence Archive under project PRJCA001063. The PAAD data comes from TCGA and can be downloaded from https://github.com/Philyzh8/scMGCA/tree/master/PAAD. The MSigDB databases for gene set analysis can be downloaded from https://www.gsea-msigdb.org/gsea/msigdb. These datasets have been deposited in the Zenodo database (https://doi.org/10.5281/zenodo.7475687). All other relevant data supporting the key findings of this study are available within the article and its Supplementary Information files or from the corresponding author upon reasonable request. Source data are provided with this paper.

## Code availability

scMGCA is released as a python package at: https://pypi.org/project/scMGCA. The source code and usage tutorial at GitHub: https://github.com/Philyzh8/scMGCA[72].

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

## Acknowledgements

The work described in this paper was substantially supported by the National Natural Science Foundation of China under Grant No. 62076109 (X.L.). The work described in this paper was substantially supported by a grant from the Health and Medical Research Fund, the Food and Health Bureau, The Government of the Hong Kong Special Administrative Region [07181426] (K.-C.W.). This research was substantially sponsored by the research projects (Grant No. 32170654 and Grant No. 32000464) (K.-C.W.) supported by the National Natural Science Foundation of China.

## Author contributions

X.L. conceived and supervised the project. Z.Y. developed and implemented the algorithms under the guidance of X.L., and Z.Y. wrote the manuscript. Z.Y. conducted the experiments. X.L., Z.Y. and K.-C.W. did the biological interpretation. Z.Y., Y.S., and Y.Y. completed the figures and manuscript. Z.Y., X.L., and K.-C.W. revised the manuscript. Y.L., F.W., S.Z., and Y.C. provided advice in method development. All authors approved the manuscript.

## Competing interests

The authors declare no competing interests.
