## [Peer Review File · Nature Communications]

REVIEWER COMMENTS

Reviewer #1 (Remarks to the Author):

In this manuscript, Yu et al. provide a novel clustering method for scRNA-seq data based on a graph-embedding autoencoder. Their approach combines learning the normalised count data and topological information captured by positive pointwise mutual information. Overall, this is an interesting method that appears to be superior to existing approaches for at least some applications, and will therefore be of interest to the community. However, some shortcomings need to be addressed before publication.

Major comments:

1. There is insufficient information on run times and scalability of the method. Which dataset does Figure 2C refer to? What is the run time on the Tabula Muris dataset? How does run time scale with cell and gene numbers? How does the run time compare with non-GNN-based methods?
2. Unlike PCA, scMGCA as well as t-SNE and UMAP have many parameters. The choice of parameters can greatly affect the two-dimensional embedding. Therefore, the comparison of approaches based on a single choice of parameters in Fig. 3A is not very helpful to the reader. Instead, the authors should demonstrate that their approach outperforms t-SNE and UMAP across a wider parameter space.
3. How many highly variable genes (d) were selected for each of the comparisons? How do results change if all genes are included? It would be interesting to know if scMGCA can detect e.g. rare cell types and small clusters that would not be picked up by other methods.
4. The authors should provide additional explanations of their rationale for the chosen network architecture. For example, why do they use two consecutive layers of GCN? Have they tried other architectures, e.g. considering count data and PPMI matrix in parallel?
5. For some datasets, K-means appears to work better as initialisation method than spectral clustering, while for other datasets, the opposite is true. Can the authors suggest why this might be the case? Do the respective datasets have anything in common?
6. While the analysis in Fig. 5 demonstrates that clustering by scMGCA may be used for functional genomics studies, it provides no comparison of results with other clustering approaches. Does scMGCA enable detection of genomic features that would not be detected using standard approaches (e.g. the standard scanpy or Seurat workflows)?
7. More generally, the authors demonstrate overall that scMGCA provides more defined cell clusters compared to other methods, but otherwise gives results consistent with other methods. However, in order for this method to be truly interesting, they should show that scMGCA enables the discovery of novel biologically meaningful results that could not be obtained by other methods.
8. The analysis of the PDAC dataset is unsatisfactory. No tumour cells are annotated in Fig. 6A - were they excluded or are they called 'tissue stem cells' here? Why? The authors also need to explain why they think "vascular smooth muscle contraction, influenza A, complement and staphylococcus aureus infection" are "closely related to PDAC". As for MIF, can the authors explain why acinar cells might be the most significant source in Fig. 6H? The result on Ro60/SSA is potentially very interesting but needs to be further investigated. What are the genes contributing to SLE pathway overexpression in this dataset? What might be the mechanism connecting SLE and PDAC? Can this link be experimentally validated and/or evidence be found in large cohort studies?

Minor comments:

1. The manuscript contains some grammatical errors and would benefit from further proofreading.
2. In the introduction, the authors state that "annotating cell types by unsupervised

learning [is] called clustering” and that only “linear dimensionality reduction methods [...] exist in the toolkits”. Both of these statements are not quite correct and should be amended.

3. The authors write that “these deep embedding clustering methods usually ignore the structural information propagation” - what is meant by “structural information propagation”?

4. UMAP should be included as an additional method in Fig. 5B.

5. The authors state that “chemotaxis and migration of neutrophils [...] are the key processes in repairing muscle damage”. This is arguably not true and should be reworded.

Reviewer #2 (Remarks to the Author):

Yu et al. present a clustering algorithm for scRNA-seq data leveraging graph convolution networks (GCNs) to embed cells and reconstruct the adjacency matrix (cell graph in their case) while optimizing clustering loss and additional gene reconstruction loss. The authors compare their algorithm with the existing frameworks for clustering and dimensionality reduction. They also show a use case of their method that can help analyze PDAC data. scRNA-seq representation learning has been an active field for past years. Many papers (as rightfully cited by authors) have already tackled this problem, from modeling count data to (deep learning-based) clustering and embedding cells using adjacency matrix and GCNs or its variations. The main novelty here is transforming the adjacency matrix to the PPMI matrix (the multitasking has been done before using DESC (Li et al, Nat Com 2020) and other algorithms. However, this is just the preprocessing step, and others have already explored the main components. Additionally, applications including GRN inference, cellular communication analysis (cell-chat) and gene enrichment analysis can already be done using other methods (scVI or other latent spaces) or directly on gene expression. Therefore, it is not clear what extra analysis step or insight scMGCA brings that is not possible to do compared to existing methods.

Major points :

1- The author did not compare their method non GCN based approaches such as DESC or scVI/scanVI (for dim reduction and also combined with scanpy for clustering) and other existing such as (Hu et al. 2022) or (Srinivasan et al. 2020). It would be essential to compare some of these methods since they are very popular in scRNAseq analysis and widely used.

2- The primary metric to compare the performance is NMI; I think authors can add more metrics such as ASW, ARI, and many more that can be found in benchmarking papers (e.g biopreservation metrics here (Luecken et al. 2022). Additional metrics can help to demonstrate how the method works in clustering rare-cell types or similar cell types along a developmental trajectory.

3- How do time and scaling compare with Leiden/Louvain clustering method as the main clustering method used in scanpy and seurat pipelines? I would also go for a more extensive dataset since a 100k cells dataset is not considered massive, considering datasets with millions of cells that are routinely available.

3- It would be crucial to learn more about how each loss was weighted overall in loss function since this can drastically influence the performance and representation.

4- how does the model handle the integration and clustering of multiple datasets (or multiple batches within the same dataset) together, accounting for the batch effect?

5- how did the authors perform hyperparameter optimization, and how would this affect the user?

6- when does the algorithm break? It would be great to show some failure cases to help the users.

Minor:

1- There are some typos in the text (e.g evolution metric in section B)

2- It seems some visible lines are not part of any panel in figure 2, which might be the artifact of the graphical post-processing app.

References :

Hu, Hang, Zhong Li, Xiangjie Li, Minzhe Yu, and Xiutao Pan. 2022. "ScCAEs: Deep Clustering of Single-Cell RNA-Seq via Convolutional Autoencoder Embedding and Soft K-Means." *Briefings in Bioinformatics* 23 (1). <https://doi.org/10.1093/bib/bbab321>.
Luecken, Malte D., M. Büttner, K. Chaichoompu, A. Danese, M. Interlandi, M. F. Mueller, D. C. Strobl, et al. 2022. "Benchmarking Atlas-Level Data Integration in Single-Cell Genomics." *Nature Methods* 19 (1): 41–50.
Srinivasan, Suhas, Anastasia Leshchik, Nathan T. Johnson, and Dmitry Korkin. 2020. "A Hybrid Deep Clustering Approach for Robust Cell Type Profiling Using Single-Cell RNA-Seq Data." *RNA* 26 (10): 1303–19.

Reviewer #3 (Remarks to the Author):

Brief comments:

Correcting biological information loss in scRNA-Seq analysis is important challenge for the bioinformatic researchers. In this manuscript, the authors developed a method (scMGCA) to solve this kind of problems, specifically utilizing cell-graph autoencoder approach to remedy single-cell dropout events and to optimize dimensionality reduction and cell clustering analyses. Through comparison with other existing tools in multiple real data, the authors displayed the outperformance of scMGCA. However, the architecture of scMGCA autoencoder is similar to other existing tool scGNN, the authors may need to directly compare scMGCA with other cell-graph-neural-network tools (at least scGNN) in core steps and show the specificity of scMGCA. Moreover, convergence is one character of deep learning, the converged output of cell-graph autoencoder may weaken the cell heterogeneity of the same cell-type of cells. In the manuscript, the results of scMGCA showed some potential similar problems that the cells in transition status could not be visualized and identified demonstrably.

Major points:

(1) For my understanding, scMGCA and scGNN utilize similar architecture to build graph convolutional network, especially the key processes and relationships between cell-graph, cell clustering and feature imputation (the details in scMGCA Results part A and scGNN Results "The architecture of scGNN ..." as well as the corresponding Methods parts) 1. The authors might need to directly explain what difference in the autoencoder between scMGCA and scGNN and what advantages of scMGCA.

I understand the authors described the key steps and mathematic models of scMGCA in the manuscript, especially in Results part A and Discussion, but without explaining the superiority of these approaches by straightforward comparison with other tools, it is difficult for the audience to estimate the innovation of scMGCA. The authors might learn

from other single-cell tool publications. For instance, the architecture of scDeepCluster is similar to DCA, the scDeepCluster paper gives direct discussion and explain the difference (page 192) 2,3.

(2) According to Fig. 1 and Methods part H, the authors utilized random surfing model to generate PPMI matrix, which estimates co-occurrence probability between cells, for building cell-graph; then scMGCA constructed GCN encoder by the cell-graph and gene expression matrix. I respect the authors' efforts and work, but I am curious why the encoder cannot be from cell-graph by KNN with gene expression matrix weighted directly. As the author mentioned in Methods part H and equation 5~7, PPMI reflects the co-occurrence probability of cells and the higher PMI score represents the closer relationship between cells, "Therefore, the cells of the same type in the biological network have higher weights in the cell graph". But, for my understanding, the KNN graph can achieve this kind of similarity metrics directly 4. Because PPMI is important step for scMGCA to build GCN encoder, the authors may elaborate the significance.

(3) In cell clustering and dimensionality reduction parts (Results part B and C), the authors displayed the outperformance of scMGCA in segregating different types of cells. However, the results in these two parts show some potential problems that scMGCA may artificially enlarge the dissimilarity of cell types and the cell heterogeneity of the same type of cells cannot be preserved in scMGCA visualization. Specifically, "Camp-Brain" data in the original publication is about progenitor-to-neuron differentiation, and a part of the single-cell data were generated from induced stem cells (GSE75140) 5. In Fig. S2 and S7 of this manuscript, most tools, except scMGCA, showed the blurred boundaries between cell types and reflected the process of cell differentiation, consistent with the analysis of the original paper 5. By contrast, scMGCA separated cell type independently and scMGCA visualization may distort the cell relationships. Similar situation can also be found in the results of "Klein" data. I suggest the authors improve scMGCA performance in this area, at least displaying similar results to other single-cell tools or dimensionality reduction methods.

I noticed that the authors showed cell trajectory in Fig. 5h, but the cells in the panels do not have logical biological connections (i.e., the cells belong to different cell types, it is difficult to estimate whether trajectory connections in Fig.5h are appropriate) 6. Further, Fig.5h panels show the output of monocle-2 dimensionality reduction, but most scMGCA dimensionality reduction plots do not contain cell-cell connections (Fig. S7-S10). The authors could find the relative topics in UMAP paper 7 or Phate paper 8.

(4) Although correcting single-cell dropout issue is one of the targets in this study, I found the interpretation and analyses of scMGCA imputed data were very limited in this manuscript. It is difficult to estimate the performance of scMGCA in dropout-event correction.

(5) The authors evaluated the stability of scMGCA with different parameters in Results part D, but the discussion of how to build cell graph may not be sufficient. I encourage the authors could provide some suggestions and strategies for the users to optimize parameters to construct cell graph.

Minor points:

(1) Seurat and SCANPY employ other clustering algorithms in their toolkits. When comparing their cluster results (Fig. 2a and 3b), the authors need to clarify what algorithms is used in the manuscript.

For SCANPY, <https://scanpy.readthedocs.io/en/stable/api.html#clustering-and-trajectory-inference>

For Seurat, please notice the parameters "method" and "algorithm"
library(Seurat)
help("FindClusters")

(2) In Results, the authors mentioned "Monocle" in part B and "Monocle2" in part E, but in part M of Methods, the authors provided the link of "Monocle3". The authors may need to unify what Monocle version you used in the study. If using Monocle2, it also employs other cluster method.

<http://www.bioconductor.org/packages/release/bioc/manuals/monocle/man/monocle.pdf>

The link for Monocle2

<http://cole-trapnell-lab.github.io/monocle-release/docs/>

(3) In Introduction and Results part A, the authors' interpretation ("To address these limitations, deep embedding clustering ...") and ("scMGCA can learn the low-dimensional representation ...") are not consistent.

(4) The subtitle "scMGCA is the most accurate of ..." is not appropriate.

(5) In Results part D, the sentence "... when $s=2$ and $d=2$..." needs to be corrected.

Reference

1. Wang, J. et al. scGNN is a novel graph neural network framework for single-cell RNA-Seq analyses. *Nature Communications* 12, (2021).
2. Tian, T., Wan, J., Song, Q. & Wei, Z. Clustering single-cell RNA-seq data with a model-based deep learning approach. *Nature Machine Intelligence* 1, 191-198 (2019).
3. Eraslan, G., Simon, L., Mircea, M., Mueller, N. & Theis, F. Single-cell RNA-seq denoising using a deep count autoencoder. *Nature Communications* 10, (2019).
4. Eppstein, D., Paterson, M. & Yao, F. On Nearest-Neighbor Graphs. *Discrete & Computational Geometry* 17, 263-282 (1997).
5. Camp, J. et al. Human cerebral organoids recapitulate gene expression programs of fetal neocortex development. *Proceedings of the National Academy of Sciences* 112, 15672-15677 (2015).
6. The Tabula Muris Consortium., Overall coordination., Logistical coordination. et al. Single-cell transcriptomics of 20 mouse organs creates a Tabula Muris. *Nature* 562, 367-372 (2018).
7. Becht, E. et al. Dimensionality reduction for visualizing single-cell data using UMAP. *Nature Biotechnology* 37, 38-44 (2018).
8. Moon, K. et al. Visualizing structure and transitions in high-dimensional biological data. *Nature Biotechnology* 37, 1482-1492 (2019).

Reviewer #1:

In this manuscript, Yu *et al.* provide a novel clustering method for scRNA-seq data based on a graph-embedding autoencoder. Their approach combines learning the normalised count data and topological information captured by positive pointwise mutual information. Overall, this is an interesting method that appears to be superior to existing approaches for at least some applications, and will therefore be of interest to the community. However, some shortcomings need to be addressed before publication.

Response: We would like to thank you for your insightful comments and suggestions. For the past four months, we have duly followed your comments and suggestions to address all the raised concerns as elaborated in the following responses. In particular, we would like to thank you for your constructive comments that enabled us to improve our manuscript in numerous ways. Thank you very much for your precious time and kind attention.

1. *There is insufficient information on run times and scalability of the method. Which dataset does Figure 2C refer to? What is the run time on the Tabula Muris dataset? How does run time scale with cell and gene numbers? How does the run time compare with non-GNN-based methods?*

Response: Thank you very much for your critical comments. Figure 2C refers to the comparison of the running times between scMGCA and three other GNN-based clustering methods on those 20 datasets. To address your comments, we have now reconstructed the running time experiments for the comparisons with 12 other computational methods, including the GNN-based methods (scGNN, scGAE, and GraphSCC), non-GNN-based methods (scziDesk, scDeepCluster, DCA, DEC, DESC, scVI, and scCAEs) and traditional methods (Seurat and SHARP). For the test datasets, we adopted 20 small-scale datasets with 300–25,000 cells, and a large-scale dataset called ‘Tabula Muris’ with 70,118 cells. The experimental results are depicted in **Fig. 2c**. It is noteworthy that scMGCA divides the data into multiple batches for learning when clustering datasets with more than 25,000 cells ($n > 25,000$). The reason is that for small-scale datasets, the time and memory consumption are within a reasonable range, and batch learning can result in the loss of important topological information in the cell graph, influencing the clustering effect. In contrast to large-scale datasets, the cell graph consumes a lot of time and memory and generates redundant information; however, batch learning can greatly reduce the time and memory consumption while improving clustering effectiveness. As depicted in **Fig. 2c**, we observe that the running time curve of scMGCA drops and is within reasonable ranges (i.e. all less 1.5 hour). The slightly longer times for scMGCA compared to these non-GNN-based algorithms (including DEC, DESC, and scVI) is attributed to the fact that the GNN-based method (including scMGCA) additionally consider the cell graph at the expenses of extra time and memory consumption.

For the second concern, to evaluate how running time is changed along the number of cells and number of genes, we also tested the respective running time of scMGCA for different numbers of cells and different numbers of genes on ‘Tabula Muris’ (**Supplementary Fig. 9**). Specifically, we randomly selected 5000, 10,000, 20,000, 30,000, 40,000, 50,000, 60,000, and 70,000

cells from the original 70,118 cells, to evaluate the impact from the number of cells. For the number of genes, we selected 500, 1000, 2000, 3000, 4000, 5000, 6000, and 7000 highly variable genes using the SCANPY package, to investigate the effect of different numbers of genes. It can be observed from **Supplementary Fig. 9** that scMGCA demonstrates a monotonically linear increase in running time along with the number of cells and genes, demonstrating its computational scalability.

In addition, **Fig. 2c** shows the overall running time of scMGCA with 12 other computational methods, including GNN-based methods (scGNN, scGAE, and GraphSCC), non-GNN-based methods (scziDesk, scDeepCluster, DCA, DEC, DESC, scVI, and scCAEs) and traditional methods (Seurat and SHARP). We can clearly observe that the running time of the traditional methods, Seurat and SHARP, is always faster than all the deep learning methods. Compared to other GNN-based methods, our proposed model can provide a shorter running time. We also found that scGAE cannot be run on the datasets with more than 12k cells.

To further demonstrate the scalability of scMGCA, we applied scMGCA to the dataset composed of 1,306,127 mouse brain cells. Most comparison algorithms failed or produced no results within 24 hours for such a big dataset. Ultimately, only scMGCA, DEC, DESC, and traditional methods were able to run successfully. Therefore, we compared their running time and memory usage on different numbers of cells (**Supplementary Fig. 11**). From the experimental results, the running time of traditional methods is shorter than other methods, which is largely consistent with the evaluation in (1). However, we can observe that Seurat and SHARP require more than 60GB of memory, and SCANPY requires nearly 40GB of memory to analyze 1.3 million cells, rendering them inaccessible to the majority of researchers who lack enough computational resources. Meanwhile, although the running time of scMGCA is high, it is still within a reasonable range and scMGCA is the only GNN-based method (in contrast to scGNN, scGAE, and GraphSCC) that was able to be successfully run on an 1.3 million cell dataset. In terms of memory consumption, DESC outperforms the other methods. The memory consumption of DEC and scMGCA are both smaller than the traditional methods. Interestingly, the time and memory requirements of the deep learning method DESC on large-scale datasets is more balanced than other methods. Possibly, for the deep graph learning method scMGCA, the construction and calculation of cell graph can incur additional time and memory costs, resulting in the extra time and memory consumption accordingly. Moreover, since a small number of pre-training epochs may not be helpful for further analysis, our model increases the number of pre-training epochs to 200 as opposed to DESC that only evaluates pre-training epochs 10 times on large-scale data; therefore, it greatly increases the final running time.

Overall, we observe that, excluding the extra time and memory consumption from the cell graph computation, there is not much difference between scMGCA and the deep learning methods. Moreover, it is worth noting that among several graph neural networks compared in this paper, scMGCA is the only one that can run successfully; it can also be seen as an advancement in deep graph learning method development.

C

Fig. 2c. Comparison of the running time between scMGCA and other state-of-the-art clustering methods.

Tabula Muris

Supplementary Fig. 9. Running time of scMGCA with different numbers of cells or genes on 'Tabula Muris'.

Supplementary Fig. 11. Comparison of running time and memory usage of scMGCA to other methods with different numbers of cells.

2. Unlike PCA, scMGCA as well as t-SNE and UMAP have many parameters. The choice of parameters can greatly affect the two-dimensional embedding. Therefore, the comparison of approaches based on a single choice of parameters in Fig. 3A is not very helpful to the reader. Instead, the authors should demonstrate that their approach outperforms t-SNE and UMAP across a wider parameter space.

Response: Thank you very much for your comments. We agree with you for the need to demonstrate that our approach outperforms t-SNE and UMAP across a wider parameter space. As suggested, we have extended the parameter space of t-SNE and UMAP and conducted experiments to demonstrate the performance comparison. The overview of the different parameter cases for t-SNE and UMAP is summarized in **Supplementary Table 1** and the comparison results are tabulated in **Supplementary Tables 2-3, Fig. 3a and Supplementary Figs. 12-15**.

For t-SNE, there are three adjustable parameters including “perplexity”, “metric”, and “init” respectively. The “perplexity” is related to the number of nearest neighbors for the manifold learning algorithm, generally between 5 and 50. We enumerated the value among the set [10, 30, 50]. The “metric” is distance function between instances in the feature space, we enumerated four kinds of distance methods: Euclidean distance, cosine distance, Pearson correlation coefficient, and Minkowski distance. The “init” is the initialization method of embedding; it is mainly divided into random sampling and PCA. For UMAP, there are also

3. How many highly variable genes (d) were selected for each of the comparisons? How do results change if all genes are included? It would be interesting to know if scMGCA can detect e.g. rare cell types and small clusters that would not be picked up by other methods.

Response: Thank you very much for the constructive comments. According to the previous hyperparameter analysis, we have conducted an experiment to benchmark scMGCA on the 20 real datasets with different numbers of highly variable genes (d) ranging from 300 to 2000, as depicted in **Fig. 5b**. From the results, we find that 500 highly variable genes have the best performance for scMGCA.

Fig. 5b. The impact of different numbers of selected genes on scMGCA.

Based on your suggestion, we have also added the case of all genes into the comparison groups to demonstrate the performance in the **Fig. 5b**. It can be observed that the clustering performance of selected genes is generally better than all genes, due to the capability of filtering out unimportant genes and preventing the curse of dimensionality while screening for the genes with high variances. We compared these two sets of results using the t-test, with p -values of 4.1×10^{-4} (ARI) and 1.1×10^{-2} (NMI), respectively, indicating that the improvements were significant.

In addition, to investigate whether scMGCA can detect rare cell types and small clusters that cannot be detected by other methods, we compared scMGCA with the other deep learning methods including scziDesk, scDeepCluster, DCA, DEC, scGNN, scGAE, and GraphSCC on four datasets ('QS heart', 'Muraro', 'Qx limb Muscle', and 'Adam') that contain the rare cell types and small clusters for in-depth examination. To construct a fair and explicit comparison, we first applied t-SNE to project the raw single-cell data into a 2D space and visualized it using true labels. After that, we applied the cluster labels obtained by scMGCA and the other deep learning methods, including scziDesk, scDeepCluster, DCA, DEC, scGNN, scGAE, and GraphSCC (**Supplementary Fig. 8**) to visualize in the same 2D space. We observe that scMGCA can detect rare cell types and small clusters that the other methods cannot pick up on those datasets, as depicted in **Supplementary Fig. 8**. Indeed, for the 'QS Heart' dataset, only scMGCA was able to accurately detect two small clusters of smooth muscle cells and myofibroblast cells, which were aggregated by the other methods. A similar situation was seen in the 'Muraro' dataset, where only scMGCA and DCA could distinguish delta and gamma cells; the other methods either merged these two cell types into one group (e.g.,

scziDesk, scDeepCluster, and scGAE), or grouped the two cell types as in other larger cell populations (e.g., DEC, scGNN, and GraphSCC). Furthermore, scMGCA, scziDesk and scGAE could accurately detect T cells in the ‘Qx Limb Muscle’ dataset, while the remaining methods assigned T cells to other types of cell clusters. In particular, compared to most non-GNN-based methods (except DEC) that successfully identified delta cells in the ‘Adam’ dataset, all GNN-based methods failed, except scMGCA which still correctly identified delta cells in this situation. Overall, our proposed scMGCA either outperforms or complements existing approaches in identifying cell types and could reliably detect rare cell types and small clusters.

Supplementary Fig. 8. scMGCA can detect rare cell types and small clusters.

4. The authors should provide additional explanations of their rationale for the chosen network architecture. For example, why do they use two consecutive layers of GCN? Have they tried other architectures, e.g. considering count data and PPMI matrix in parallel?

Response: Thank you very much for the insights. We agree with you and have followed your comments to provide additional explanations on our rationale for the chosen network architecture. In addition, we have conducted an experiment to explore the effect of different numbers of consecutive layers in our proposed scMGCA. After that, we parallelized the multinomial-based autoencoder and the graph convolution autoencoder so that the gene expression matrix and the cellular PPMI graph could be inputted in parallel.

In the experiments, the number of consecutive layers in GCN has been varied to demonstrate why two layers are adopted at the end. For the number of consecutive layers of GCN, we have benchmarked four cases as one, two, three, and four layers with the number of neuron nodes [15], [128, 15], [256, 128, 15] and [512, 256, 128, 15], respectively, which are denoted as scMGCA1, scMGCA2 (our proposed scMGCA), scMGCA3, and scMGCA4 respectively. We compared the clustering performance of

these four different frameworks on 20 single-cell datasets with NMI and ARI as evaluation metrics (Supplementary Fig. 22). It can be clearly observed that the NMI and ARI of scMGCA2 (scMGCA) outperform the other frameworks on most datasets, and the average clustering performance on all 20 datasets is also the best. For the ‘QS Trachea’, ‘Qx Bladder’ and ‘Romanov’ datasets, the clustering performance of the two-layer GCN is significantly better than the other layers. The ARI score on the ‘Chen’ dataset is greatly affected by the number of layers of GCN. scMGCA4 has the worst clustering performance among all the comparison frameworks. For the reasons, it could be possible that the network structure with 3 or more layers could be over-complicated due to the excessive number of hidden layer nodes. It is more likely to fall into local minima during the back-propagation process with gradient vanishing, and the learning speed will also be too slow.

Supplementary Fig. 22. The clustering performance of four different frameworks on 20 single-cell datasets was evaluated using NMI and ARI as evaluation metrics.

In addition, as suggested by the reviewer, we have also designed a parallel framework-based scMGCA inspired by SEDR (2) called scMGCA-p, as shown in Supplementary Fig. 23 that considers count data and PPMI matrix in parallel. In scMGCA-p, we parallelize the multinomial-based autoencoder and the graph convolution autoencoder, allowing for the parallel inputs of the gene expression matrix and cell-PPMI graph. Indeed, the multinomial-based autoencoder simulates the dropout event of the gene expression matrix to obtain the latent representation z_1 while the graph convolutional autoencoder integrates the gene expression matrix and cell-PPMI graph capturing the cell topology to obtain the latent representation z_2 . Finally, z_1 and z_2 are concatenated and optimized with KL divergence-based clustering loss to obtain the final clustering result. The experimental

results comparing scMGCA and scMGCA-p are summarized in **Supplementary Fig. 24**, showing the clustering performance of the two architectures on 20 single-cell datasets with NMI and ARI as evaluation metrics. From the results, we can observe that the clustering performance of scMGCA-p is lower than that of scMGCA and then we compared the results of scMGCA and scMGCA-p using the pairwise t-test, with p -values of 5.4×10^{-3} (ARI) and 1.1×10^{-2} (NMI), indicating a significant difference. Nevertheless, we appreciate your providing us with such a framework, and it will be very interesting and important to study the parallel framework in the graph embedding clustering model with those citations in the future.

Supplementary Fig. 23. The network architecture of the scMGCA-p algorithm.

Supplementary Fig. 24. The comparison of the clustering performances of scMGCA and scMGCA-p on 20 single-cell datasets using NMI and ARI as evaluation metrics.

5. For some datasets, K-means appears to work better as initialisation method than spectral clustering, while for other datasets, the opposite is true. Can the authors suggest why this might be the case? Do the respective datasets have anything in common?

Response: Thank you very much for the critical and helpful comments. The reason is that the cell-PPMI graph of some single-cell datasets can preserve more cell topological information, so the information contained in the cell-PPMI graph will be richer than that in the latent embedding representation obtained only by pre-training. Therefore, using spectral clustering to initialize the the clustering centers can result in a better final clustering performance than K-means for these datasets. On the contrary, when the cell-PPMI graph of other single-cell datasets preserves less information than others, the latent embedding representation obtained by pre-training will supplement some gene expression information in topological information, and it is preferable to use K-means to initialize the cluster centers in this case.

To demonstrate this point, we first construct a truth cell graph \mathbf{T} using the ground-truth labels underlying the scRNA-seq data, which can be defined as follows:

$$T_{ij} = \begin{cases} 1, & l_j = l_i \\ 0, & l_j \neq l_i \end{cases} \quad (1)$$

where i and j represent the i -th cell and j -th cell, and l_i and l_j represent the true labels of the i -th cell and the j -th cell, respectively. According to Eq. (1), it is known that in the truth cell graph \mathbf{T} , only cells belonging to the same cluster are edge-connected; and there is not any edge connection between the cells from different clusters. Therefore, \mathbf{T} provides the real and valid topological information in single-cell data, and can be used as the "true labels" of topological information. Then, we calculate the occupancy (t) of the real topological information by measuring the cell-PPMI graph \mathbf{A} and the truth cell graph \mathbf{T} , defined as follows:

$$t = \frac{\text{sum}(\mathbf{A} \odot \mathbf{T})}{\text{sum}(\mathbf{A})} = \frac{\sum_{i=1}^n \sum_{j=1}^n (A_{ij} T_{ij})}{\sum_{i=1}^n \sum_{j=1}^n A_{ij}} \times 100\% \quad (2)$$

where \odot is the Hadamard product. The cell connection edges in \mathbf{A} that co-occur with the true topology \mathbf{T} are retained by $\mathbf{A} \odot \mathbf{T}$, and the proportion of the sum of the weights of these valid edges to the sum of all weights in \mathbf{A} is calculated to measure how much effective topology the cell-PPMI graph \mathbf{A} can retain. On this basis, we calculated the effective topological occupancy (t) of the cell-PPMI graph \mathbf{A} for each dataset, as well as their clustering performance (NMI) of initialized cluster centers using K-means and spectral clustering across the 20 datasets, as summarized in **Supplementary Table 4**.

From the table, we find that the difference in clustering performance between kmeans and spectral clustering as initialization methods is not very large on most of the datasets while there is a significant difference on a few datasets. For the "Camp-Liver", "QS Trachea" and "Qx Bladder" datasets (blue), the K-means model as an initialized clustering center is significantly better than spectral clustering, and the effective topological occupancy (t) of the cell-PPMI graph for these three datasets is below 90%. On the contrary, spectral clustering is significantly better than K-means as a method for initializing cluster centers across the 'Adam', 'Qx Limb Muscle', 'QS Heart', 'Plasschaert' and 'Qx Spleen' datasets (red), where the t of the cell-PPMI graph of these datasets is higher than 90%. For the rest, no matter whether K-means or spectral clustering is chosen as the initialized clustering center, they have similar results. In addition, on the whole, the effective topological occupancies (t) of the cell-PPMI graph of the datasets where spectral clustering is better than K-means are generally higher than those in the opposite

cases (datasets). These cases verify our previous hypothesis that when the true topological occupancy of the cell-PPMI graph constructed from the dataset is high, it preserves more topological information than the information obtained through pre-training in the latent representation, therefore spectral clustering would be better than K-means as the initialization method, while, when the true topological occupancy of the cell-PPMI graph constructed from the dataset is low, K-means becomes a better choice than spectral clustering.

Supplementary Table 4. Comparison of clustering performance and t with different initialization methods.

Methods	Pollen	Camp-Brain	Camp-Liver	QS Diaphragm	QS Limb Muscle
K-means	0.9263	0.5352	0.9224	0.9291	0.9586
Spectral	0.8971	0.5585	0.7730	0.9263	0.9586
t	84.97%	92.35%	62.37%	89.64%	92.60%
Methods	QS Trachea	QS Lung	Muraro	Qx Bladder	Klein
K-means	0.8736	0.8402	0.8255	0.9595	0.6971
Spectral	0.7369	0.8068	0.8385	0.8033	0.6694
t	87.15%	93.17%	92.05%	85.80%	82.84%
Methods	Romanov	Adam	Qx Limb Muscle	QS Heart	Young
K-means	0.7236	0.7894	0.8633	0.7999	0.7999
Spectral	0.6909	0.8593	0.9459	0.9020	0.8109
t	82.70%	92.50%	97.69%	96.38%	90.25%
Methods	Plasschaert	Qx Spleen	Chen	Tosches turtle	Bach
K-means	0.6962	0.7213	0.7241	0.7046	0.8195
Spectral	0.7732	0.7840	0.7640	0.7320	0.8241
t	95.22%	98.12%	89.02%	88.45%	94.98%

6. While the analysis in Fig. 5 demonstrates that clustering by scMGCA may be used for functional genomics studies, it provides no comparison of results with other clustering approaches. Does scMGCA enable detection of genomic features that would not be detected using standard approaches (e.g. the standard SCANPY or Seurat workflows)?

Response: Thank you very much for your insightful comments. We have followed your comments to provide a comparison of scMGCA with other clustering algorithms for functional genomics studies. Since our functional interpretation approach is used to explain deep learning models, we adopt six deep clustering algorithms as comparison methods including three non-GNN-based methods (scziDesk, scDeepCluster, and DCA) and three GNN-based algorithms (scGNN, scGAE, and GraphSCC).

In scMGCA, we propose an algorithm based on the standard deviation of the weight matrix in the encoder to uncover the key interpretable genes of the latent representation. To construct a fair comparison, we also applied this algorithm to explain all other compared methods, enabling each method to extract 200 highly expressed genes from ‘Qx Limb Muscle’. After that, we visualized the top 10 highly expressed genes in different cell clusters by various deep clustering algorithms, as depicted in **Supplementary Fig. 29**. It can be observed that scMGCA can clearly find the respective highly expressed genes from each cell cluster without confusion with other cell clusters, while the other clustering methods overlap or fail in clarifying certain cell clusters; for instance, for B cells and skeletal muscle cells, the expression levels of the highly expressed genes identified by scziDesk were not prominent. scDeepCluster, DCA and scGNN found that the top 10 highly expressed genes in some cell clusters were also highly expressed in other cell clusters, without the obvious identification of specific genes that were highly expressed in one cell cluster alone. The gene expression of GraphSCC in B cells and mesenchymal cells is not high. scGAE

essentially did not find highly expressed genes in macrophages, skeletal muscle cells, and B cells.

In addition, we further compared the capacity of scMGCA and other clustering methods to uncover differentially expressed genes. For each algorithm, we extracted 200 differentially expressed genes based on their predicted labels by performing the wilcoxon analysis of one group against the other groups. We then performed multiple hypothesis tests using the Benjamini-Hochberg correction to adjust for the correct p values ($pvals_adj$). Finally, we computed the $pvals_adj$ of the 200 differential genes of each method as the calling probability of the differential genes while those of the real differentially expressed genes obtained from the the annotated labels in the data are considered as the true measures. We then used the scikit-learn python package to evaluate the area under the ROC curve (AUC) for each method and plotted the ROC curve, as summarized in **Supplementary Fig. 30**. Overall, scMGCA (AUC = 0.89) is generally superior to other clustering methods.

Supplementary Fig. 29. Stacked violin plots of the top 10 genes expressed in each cluster for scMGCA and comparative clustering methods.

Supplementary Fig. 30. Accuracy measurements of differentially expressed genes by different clustering methods.

To further elaborate on whether scMGCA is able to detect genomic features that are not detected by the standard methods (Seurat and SCANPY), we added a comparison of scMGCA with SCANPY and Seurat for detecting genomic features. For SCANPY, we clustered the data using the function *scanpy.tl.leiden* with the parameter set to “*resolution = ‘0.2’*”, and detected differential genes using the function *scanpy.tl.rank_genes_groups* with the parameter set to “*method=‘wilcoxon’*”. For Seurat, we used the function *FindClusters* to cluster the data, where the parameters were set to “*resolution = ‘0.2’*”, and employed the function *FindAllMarkers* to detect differential genes, where the parameters were set to “*min.pct = ‘0.25’*”, *logfc.threshold = ‘0.5’*”.

We used SCANPY and Seurat, respectively, to detect differentially expressed genes with high gene expression levels in each cell cluster, and then compared with the highly expressed genes among the 200 genes extracted by scMGCA. Following the reviewer’s comment, we selected three important marker genes that only scMGCA could detect, and compared the expression of these three marker genes in the three methods using violin plots (**Supplementary Fig. 31**). The marker genes are *Dcn*, *Pdcd1* and *Il1rn*, respectively. The *Dcn* gene provides instructions for making a protein called decorin, which enhances *IFN- γ* - and *LPC*-dependent macrophage activation in mice (3). The *Pdcd1* correlates with clonal size of follicular T cells and is a known gene that positively correlates with clonal expansion (4). The expression of the *Il1rn* gene affects body weight and glucose uptake in skeletal muscle cells, providing a hypothesis for a functional link between obesity and insulin resistance (5). It can be clearly seen from **Supplementary Fig. 31** that the two genes *Pdcd1* and *Il1rn* were not detected using the SCANPY and Seurat methods. Although the gene *Dcn* was identified by all three methods, only scMGCA used it as a marker gene of the macrophage cell cluster.

Supplementary Fig. 31. Stacked violin plots of the selected genes expressed in each cluster for scMGCA, SCANPY, and Seurat.

7. More generally, the authors demonstrate overall that scMGCA provides more defined cell clusters compared to other methods, but otherwise gives results consistent with other methods. However, in order for this method to be truly interesting, they should show that scMGCA enables the discovery of novel biologically meaningful results that could not be obtained by other methods.

Response: Thank you very much for the constructive comments. We have added extensive analysis and experiments to illustrate the biological novelties of the algorithm. In particular, several novel biologically meaningful results have been obtained:

1. scMGCA has robust clustering and dimensionality reduction performance on single-cell datasets across different platforms. We compared scMGCA with 12 other single-cell analysis methods on 20 real scRNA-seq datasets across platforms (Results part B and the response to Reviewer #2's comment 1). The experimental results show that scMGCA has better clustering and dimensionality reduction performance whether compared with deep learning methods, graph deep learning methods or tradi-

tional methods. Furthermore, scMGCA still outperforms the other 12 comparison methods for data across multiple platforms. The results demonstrate its cross-platform capability in the wet-lab sense.

2. scMGCA can detect rare cell types and small clusters which cannot be picked up by other methods. We employ t-SNE to visualize the latent embedding representation of scMGCA, which is annotated and colored with the ground-truth labels of the data (the response to Reviewer #1's comment 3). The results show that scMGCA can clearly identify rare cell types and small clusters that are difficult to be isolated and distinguished by other methods. This also illustrates that scMGCA has the ability to identify special cell clusters in single-cell data; such discovery of cell heterogeneity is an important step in the downstream analysis of scRNA-seq data, which may be used to distinguish tissue-specific sub-types based on identified gene sets. In addition, we also thank the reviewers for their comments to make scMGCA more comprehensive than the previous version.

3. scMGCA is also a specially interpretable approach for functional genomics. scMGCA is the first GNN-based single-cell analysis method to elucidate functional genomic information through latent embedding representations and weight matrices in graph neural networks, and it is also a novel method to select important genes by standard deviation ranking of weights (Methods part M). Several experiments (Results part F and the response to Reviewer #1's comment 6) demonstrate that the latent embedding representation of scMGCA is biologically interpretable without any major sacrifices on its cell clustering performance. Furthermore, with the same interpretable gene selection method, scMGCA was able to discover differentially expressed genes that were evident in cell clusters, while other deep learning approaches might not. In particular, scMGCA is also able to discover potential marker genes in special cell clusters that cannot be detected by standard methods.

4. scMGCA can remove the batch effect in multiple datasets from different scRNA-seq protocols. We conducted the experiment to test the scMGCA on a combined dataset consisting of four publicly available datasets on human pancreatic tissue generated using four different scRNA-seq protocols, CEL-seq (6), CEL-seq2 (7), Fluidigm C1 (8), and Smart-seq2 (9) (Results part D and the response to Reviewer #2's comment 5). By comparison with other batch correction methods, it can be demonstrated that scMGCA not only accurately cluster cells but also simultaneously correct for batch effects in multiple datasets with strong batch differences.

5. As a deep embedding graph clustering algorithm, scMGCA can be effectively applied to large-scale datasets (more than one million cells). In **Supplementary Note 5** and the response to Reviewer #2's comment 3, among all the GNN-based algorithms, only scMGCA was able to successfully cluster the 1.3 million dataset. Furthermore, the time and memory consumption of scMGCA is also within a reasonable range; and it does not exceed the deep learning algorithm too much due to the construction and calculation of the cell graph. Therefore, we believe that scMGCA can also be regarded as an advancement in deep graph learning on large-scale scRNA-seq datasets.

6. scMGCA can elucidate the underlying regulatory mechanisms of pancreatic ductal carcinoma (PDAC). We applied scMGCA to PDAC data for cell clustering and defined cell types by marker genes in the cell clusters. Interestingly, all type 2 ductal cells identified by scMGCA were derived from tumor cells (**Fig. 7b**). Further, we found that type 2 ductal cells were enriched for several pathways essentially associated with PDAC or cancer, and most of them were consistent with the pathways enriched in tumor samples from clinical data (Results part G and the response to Reviewer #1's comment 8). This demonstrates the ability of scMGCA to elucidate the underlying regulatory mechanisms of complex diseases and deliver biologically meaningful results.

8. *The analysis of the PDAC dataset is unsatisfactory. No tumour cells are annotated in Fig. 6A - were they excluded or are they called 'tissue stem cells' here? Why? The authors also need to explain why they think "vascular smooth muscle contraction, influenza A, complement and staphylococcus aureus infection" are "closely related to PDAC". As for MIF, can the authors explain why acinar cells might be the most significant source in Fig. 6H? The result on Ro60/SSA is potentially very interesting but needs to be further investigated. What are the genes contributing to SLE pathway overexpression in this dataset? What might be the mechanism connecting SLE and PDAC? Can this link be experimentally validated and/or evidence be found in large cohort studies?*

Response: Thank you very much for your constructive comments. We apologize for those confusions arisen from the analysis of the PDAC dataset. The primary reason is because, to ensure unbiased cell type annotations, SingleR (10) was adopted to perform unbiased cell type recognitions instead of the reliance on the original authors' manual cell annotations which are vulnerable to human errors. Therefore, it is expected to have a difference between the unbiased cell type annotations and human-annotated cell types in the PDAC dataset. In the revised manuscript, to address your concerns, we have adopted the original cancer and control cell type annotations of the PDAC data for re-performing the downstream analysis again.

The PDAC data were first preprocessed and clustered by scMGCA (**Supplementary Note 16**), and then cell types were annotated with marker genes in each cell cluster, as summarized in **Supplementary Table 5**. Finally, we identified 10 types of cells, including acinar cells, B cells, type 1 ductal cells, type 2 ductal cells, endocrine cells, endothelial cells, fibroblasts, macrophages, stellate cells, and T cells (**Fig. 7a**). Among them, the largest proportion of type 2 duct cells and the smallest proportion of endocrine cells were found. Notably, type 2 ductal cells were all detected in tumor cells as seen in **Fig. 7b**. We then visualized the expression levels of the top 10 differential genes in each cell cluster (**Fig. 7c**) and interpreted the significant marker genes of each cell cluster in detail, including their expression in all cell types distribution (**Fig. 7d**) and the difference in their expression levels between each cell cluster (**Fig. 7e**).

On the other hand, we re-performed the GSEA analysis in **Fig. 7g**. The highest enrichment score was for pancreatic secretion; it is consistent with the fact that dysregulated pancreatic duct fluid secretion is the main process related to PDAC. The digestion and absorption of proteins and fats are also the processes significantly enriched in PDAC, as evidenced by other PDAC studies (11–13). The pancreas contains different cells that produce enzymes important for digestion, including trypsin and chymotrypsin that digest protein, and lipase that breaks down fats. Therefore, the abnormality of the pancreas will indirectly or directly affect the digestion and absorption of proteins and fats. In addition, protein digestion and absorption have been reported to be associated with the development of pancreatic neuroendocrine tumors (14). Other pathways, such as natural killer cell-mediated cytotoxicity and complement and coagulation cascades, have been confirmed in PDAC-related research (15, 16).

Systematically, we used CellChat (17) to infer cell-cell communication detected by scMGCA. A variety of signaling pathways were found by CellChat, of which the insulin signaling pathway had the highest level of signaling. Due to the plethora of metabolic derangements caused by hyperinsulinemia, there is reason to believe that insulin signaling plays a potentially decisive role in tumors, including PDAC (18). From the hierarchical plot of cell interactions of the insulin signaling pathway in **Fig. 7h**, we observe that endothelial cells, fibroblasts, macrophages, and stellate cells in addition to duct cells, have a high

percentage of cells that serve as sources of the insulin signaling pathway. Meanwhile, several studies have indeed demonstrated that the insulin signaling pathway active in these cells plays an important role in the progression of PDAC (19).

To verify whether the signaling pathways identified by the single-cell PDAC data are indeed clinically relevant, we independently adopted the TCGA's database public of invasive pancreatic ductal adenocarcinomas and its variants (PAAD) to validate the insight consistency. Specifically, since the type 2 ductal cells identified by scMGCA were all derived from tumor cells, it is reasonable to assume that type 2 ductal cells and tumors are extremely correlated. On this basis, we performed GSVA enrichment on the type 2 ductal cells identified by scMGCA and 178 tumor samples from the TCGA data, respectively, and then compared them (**Supplementary Fig. 32**). It can be observed that the pathways enriched from PAAD tumor samples are mostly the same as those enriched from the type 2 ductal cells identified by scMGCA, and these pathways are essentially associated with PDAC. Among them, the co-enriched p53 (signaling) pathway is an important pathway in PDAC. As a sequence-specific transcription factor, the p53 protein is a major tumor suppressor (20). Under oncogenic stress, p53 is activated to induce multiple programs, including cell cycle exit, apoptosis and replicative senescence, aiming to limit malignant cell proliferation (21). In addition to this, it was mainly enriched for tumor-related functions such as glycolysis, protein secretion and pathways in cancer, leading to the malignant state of the type 2 ductal cells. In conclusion, the experimental results indicate that the results of scMGCA analysis of PDAC can be validated by another independent cohort study and that scMGCA can elucidate the potential regulatory mechanisms of PDAC.

Fig. 7. PDAC dataset (CRA001160) analysis by scMGCA. **a** Visualization of cell clusters identified by scMGCA via t-SNE (left) and the proportion of different cell types (right). **b** Visualization of the distribution of normal cells and cancer cells via t-SNE. **c** Heatmap of top 10 marker gene expression levels for each cell cluster, and the corresponding DNA binding motif for each cell cluster. **d** Visualization of expression distribution of the top marker gene of each cell cluster. **e** Violin plot of expression levels of top marker genes in each cell cluster. **f** Heatmap for pseudo-time analysis of marker genes. **g** GSEA on representative Gene Ontology (GO) terms for PDAC. **h** Hierarchical plot of communication between cell clusters in the Insulin signaling pathway.

Supplementary Fig. 32. GSVA enrichment pathway diagram of the result of single-cell RNA-seq data (left) and TCGA dataset (right).

Minor comments:

1. *The manuscript contains some grammatical errors and would benefit from further proofreading.*

Response: Thank you very much for the constructive comments. To fix the grammar mistakes, we have asked a native English speaker to proofread the revised manuscript. Moreover, we (the first author and the corresponding authors) have proofread the manuscript in multiple rounds again. English grammatical errors and typos have been fixed to our best effort.

2. *In the introduction, the authors state that “annotating cell types by unsupervised learning [is] called clustering” and that only “linear dimensionality reduction methods [...] exist in the toolkits”. Both of these statements are not quite correct and should be amended.*

Response: Thank you very much for your sharp catches. We agree with you and have followed your comments to amend those sentences. It is copied below for your easy reference:

“annotating cell types by unsupervised learning [is] called clustering” is amended to “As an unsupervised learning method, cluster analysis has become a key step in defining cell types based on transcriptome data as well as the basis for downstream analyses.”

“linear dimensionality reduction methods [...] exist in the toolkits” is amended to “current dimensionality reduction methods often either suffer from multiple simultaneous techniques and biological variability or ignore the intrinsic data distribution, resulting in the issue of overcrowding in the latent space and thus inaccurate cell clustering.”

3. *The authors write that “these deep embedding clustering methods usually ignore the structural information propagation” -*

what is meant by “structural information propagation”?

Response: Thank you very much for your insightful comments. We apologize for the unclear description. In our study, the meaning of “structural information propagation” is the cell topological information and the heterogeneity among cell populations that is propagated through the encoder in those deep embedding clustering methods, and then preserved in the latent embedding representation. To avoid such confusion, we have revised those sentences and proofread the revised manuscript to correct this kind of situation. It is copied below for your easy reference:

“However, these deep embedding clustering methods typically ignore cell topological information and the heterogeneity among cell populations.”

4. UMAP should be included as an additional method in Fig. 5B.

Response: Thank you very much for the suggestions. We agree with you and have followed your comments to include UMAP in revised Fig. 6b. It is copied below for your easy reference:

Fig. 6b. Cell topology preservation in the latent embedded representation of scMGCA compared with methods using PCA, t-SNE and UMAP. The upper triangular part shows the Pearson correlation coefficient of the corresponding two spaces, while the diagonal part represents the correlation heatmap of paired cells.

5. The authors state that “chemotaxis and migration of neutrophils [...] are the key processes in repairing muscle damage”. This is arguably not true and should be reworded.

Response: Thank you very much for your suggestion. We apologize for it and have followed your comments to correct the sentence as follows:

“The GO terms are mainly enriched in the chemotaxis and migration of neutrophils in the biological processes (BPs). Neutrophils are the frontline cells of the innate immune system with an important role in inflammation and tissue wound healing (41, 42).”

[41] De Oliveira, S., Rosowski, E. E., Huttenlocher, A. (2016). Neutrophil migration in infection and wound repair: going forward in reverse. *Nature Reviews Immunology*, 16(6), 378-391.

[42] Metzemaekers, M., Gouwy, M., Proost, P. (2020). Neutrophil chemoattractant receptors in health and disease: double-edged swords. *Cellular molecular immunology*, 17(5), 433-450.

Reviewer #2:

Yu *et al.* present a clustering algorithm for scRNA-seq data leveraging graph convolution networks (GCNs) to embed cells and reconstruct the adjacency matrix (cell graph in their case) while optimizing clustering loss and additional gene reconstruction loss. The authors compare their algorithm with the existing frameworks for clustering and dimensionality reduction. They also show a use case of their method that can help analyze PDAC data. scRNA-seq representation learning has been an active field for past years. Many papers (as rightfully cited by authors) have already tackled this problem, from modeling count data to (deep learning-based) clustering and embedding cells using adjacency matrix and GCNs or its variations. The main novelty here is transforming the adjacency matrix to the PPMI matrix (the multitasking has been done before using DESC (Li *et al.*, Nat Com 2020) and other algorithms. However, this is just the preprocessing step, and others have already explored the main components. Additionally, applications including GRN inference, cellular communication analysis (cell-chat) and gene enrichment analysis can already be done using other methods (scVI or other latent spaces) or directly on gene expression. Therefore, it is not clear what extra analysis step or insight scMGCA brings that is not possible to do compared to existing methods.

Response: Thank you very much for your critical comments. We understand your concerns about the novelty of scMGCA and hope that the novelties of scMGCA can be observed from the following points. For the past four months, we have significantly revised the study according to your constructive comments. Firstly, for high-dimensional sparse single-cell sequencing data, autoencoder-based deep embedding clustering is a good approach to cluster and analyze single-cell data. The reason is that the high-dimensional sparse single-cell data can be compressed and encoded through the encoder and result in the low-dimensional embedding representation which can cluster and analyze the single-cell data efficiently. Many excellent single-cell analysis methods are also based on deep embedded clustering patterns, such as DESC, scDeepCluster, DCA, etc. However, as emerging graph neural networks (GNNs) have been shown to naturally model heterogeneous cell-cell relationships and complex gene expression patterns from compressed latent spaces, deep graph embedded clustering methods have also been proposed. Our proposed scMGCA is also a deep graph embedded clustering method with its biggest difference from the deep embedding clustering methods is that scMGCA also considers the topology between cells while characterizing gene expression information, so that cell-cell relationships and complex gene expression patterns are orthogonally combined to obtain robust latent representations.

Compared with deep graph embedding clustering methods, such as the most representative method scGNN, scMGCA also has its own contribution. For the model framework, scMGCA adopts the cell-PPMI graph and introduces a decoder based on multinomial distribution. scMGCA uses the cell-PPMI graph as the cell graph to characterize cell topological information, instead of using traditional KNN or SNN to construct the cell graph. The cell-PPMI graph uses the random surfing method to calculate the transition probability of cells under a finite number of steps to obtain a transition matrix, and then calculates the PPMI to represent the co-occurrence probability between cells. The cell-PPMI graph characterizes the underlying, complex cell-cell relationships and topology across entire biological networks, not just two individual cells. scMGCA also introduces a multinomial distribution to characterize high-probability dropout events for single-cell data. scMGCA uses the multinomial-based decoder to decode the latent embedding representation, resulting in the distribution of gene expression data via the multinomial distribution. This enables the key gene expression information in the latent embedding representation to fit the single-cell data distribution for training, and can be effectively combined with the cell topology information to learn a robust low-dimensional

representation.

In particular, the novelties are summarized below:

1. **scMGCA has robust clustering and dimensionality reduction performance on single-cell datasets across different platforms.** We compared scMGCA with 12 other single-cell analysis methods on 20 real scRNA-seq datasets across platforms (Results part B and the response to Reviewer #2's comment 1). The experimental results show that scMGCA has better clustering and dimensionality reduction performance whether compared with deep learning methods, graph deep learning methods or traditional methods. Furthermore, scMGCA still outperforms the other 12 comparison methods for data across multiple platforms. The results demonstrate its cross-platform capability in the wet-lab sense.
2. **scMGCA can detect rare cell types and small clusters which cannot be picked up by other methods.** We employ t-SNE to visualize the latent embedding representation of scMGCA, which is annotated and colored with the ground-truth labels of the data (the response to Reviewer #1's comment 3). The results show that scMGCA can clearly identify rare cell types and small clusters that are difficult to be isolated and distinguished by other methods. This also illustrates that scMGCA has the ability to identify special cell clusters in single-cell data; such discovery of cell heterogeneity is an important step in the downstream analysis of scRNA-seq data, which may be used to distinguish tissue-specific subtypes based on identified gene sets. In addition, we also thank the reviewers for their comments to make scMGCA more comprehensive than the previous version.
3. **scMGCA is also a specially interpretable approach for functional genomics.** scMGCA is the first functional genomics interpretability analysis of latent embedding representations on a deep graph embedding clustering method. Unlike other methods, we propose a ranking and screening method based on the standard deviation of the parameter matrix in the GCN encoder to select 200 highly expressed genes from the latent embedding representation in scMGCA (Methods part M). From the functional analysis of these genes (Results part F and the response to Reviewer 1's comment 6), we found that the latent embedding representation of scMGCA is biologically interpretable without any major sacrifices on its cell clustering performance. In addition, scMGCA is able to discover significantly expressed differential genes that are evident in cell clusters, while other deep learning methods might not, and discover potential marker genes in special cell clusters that were not detected by standard methods (SCANPY and Seurat). These experimental results not only verify that the underlying embedding representation of scMGCA is biologically interpretable, but also provide a new way for deep graph clustering to explore biological interpretability.
4. **scMGCA can remove batch effects in multiple datasets from different scRNA-seq protocols.** We conducted the experiment to test the scMGCA on a combined dataset consisting of four publicly available datasets on human pancreatic tissue generated using four different scRNA-seq protocols, CEL-seq (6), CEL-seq2 (7), Fluidigm C1 (8), and Smart-seq2 (9) (Results part D and the response to Reviewer #2's comment 5). By comparison with other batch correction methods, it can be demonstrated that scMGCA not only accurately cluster cells but also simultaneously correct for batch effects in multiple datasets with strong batch differences.
5. **As a deep embedding graph clustering algorithm, scMGCA can be effectively applied to large-scale datasets (more than one million cells).** In **Supplementary Note 5** and the response to Reviewer #2's comment 3, among all the GNN-based algorithms, only scMGCA was able to successfully cluster the 1.3 million dataset. Furthermore, the time and

memory consumption of scMGCA is also within a reasonable range; and it does not exceed the deep learning algorithm too much due to the construction and calculation of the cell graph. Therefore, we believe that scMGCA can also be regarded as an advancement in deep graph learning on large-scale scRNA-seq datasets.

In addition, we also demonstrate that scMGCA can elucidate the underlying regulatory mechanisms of pancreatic ductal carcinoma (PDAC). We applied scMGCA to PDAC data for cell clustering and defined cell types by marker genes in the cell clusters. Interestingly, all type 2 ductal cells identified by scMGCA were derived from tumor cells (**Fig. 7b**). Further, we found that type 2 ductal cells were enriched for several pathways essentially associated with PDAC, and most of them were consistent with the pathways enriched in tumor samples from clinical data (Results part G and the response to Reviewer #1's comment 8). This demonstrates the ability of scMGCA to elucidate the underlying regulatory mechanisms of complex diseases and deliver biologically meaningful results.

Lastly, we would like to re-emphasize the importance and the thoroughness of the above analyses carried out in our current study. For scRNA-seq data analysis, scMGCA is the first **deep graph learning method** that can simultaneously perform cell clustering, dimensionality reduction, batch effect correction, million-level data analysis, interpretability analysis and single-cell data analysis of joint clinical data, which is extremely rare in the current field of graph neural networks (to the best of our knowledge). Inspired by this comment, we have revised the **Discussion** to emphasize the above points in highlighting the novelties of our approach.

Major points :

1. *The author did not compare their method non GCN based approaches such as DESC or scVI/scanVI (for dim reduction and also combined with SCANPY for clustering) and other existing such as (Hu et al. 2022) or (Srinivasan et al. 2020). It would be essential to compare some of these methods since they are very popular in scRNA-seq analysis and widely used.*

Response: Thank you very much for the helpful comments. We agree with you and have followed your comments to add three methods, including DESC, scVI and scCAEs, into our comparison group on 20 real single-cell datasets across different platforms, and the evaluation metrics were NMI, ARI and ASW (**Fig. 2a, Supplementary Figs. 1-2**). In addition, we also visualize the low-dimensional embedded representations of scMGCA and other embedded clustering methods, including those three newly added algorithms, in two dimensions using UMAP (**Supplementary Figs. 3-7**). It is copied below for your easy reference:

“ As shown in those figures, for the 20 datasets, scMGCA has the highest NMI, ARI and ASW values for 16, 15 and 15 of them, respectively. Overall, across the 20 datasets, scMGCA has the highest mean NMI, ARI and ASW values, reaching 0.8304, 0.8278, and 0.5827, respectively. After scMGCA, scziDesk has superior NMI and ARI values, while scVI has superior ASW values. In addition, we also compared the overall clustering performance (by NMI, ARI and ASW) of the methods across various data platforms, including plate-based platform, flow-cell-based platform, Smart-Seq2, SMARTer, 10X Genomics, and Drop-seq. The experimental results indicated that scMGCA outperformed the other 12 clustering algorithms on across multiple platforms, and demonstrating the effectiveness and precision of scMGCA in clustering across multiple platforms.”

In addition, we also emphasize, present, and add explicit citations to those three newly added algorithms in **Section N. Implementation details and comparisons with existing methods.**

- Unsupervised deep embedding algorithm (**DESC**, <https://github.com/eleozzr/desc>) (19). DESC clusters scRNA-seq data by iteratively optimizing the clustering objective function.
- Single-cell variational inference (**scVI**, <https://github.com/YosefLab/scVI>) (20). scVI uses stochastic optimization and deep neural networks to aggregate information from similar cells and genes and approximate the distribution of the observed expression values, taking into account batch effects and limited sensitivity.
- Deep scRNA-seq clustering method via convolutional autoencoder and soft K-means (**scCAEs**, <https://github.com/gushenweis/scCAEs>) (21). scCAEs learns feature representation and clustering for scRNA-seq based on convolutional autoencoder embeddings and soft K-means.

Fig. 2a. Comparisons of NMI values between scMGCA and 12 single-cell clustering methods on 20 real scRNA-seq datasets across multiple platforms (left) and on different platforms (right).

Supplementary Fig. 1. Comparisons of ARI values between scMGCA and 12 single-cell clustering methods on (a) 20 real scRNA-seq datasets across multiple platforms and (b) on different platforms. The blank bar chart in figures generated by scGAE are due to the fact that scGAE cannot run them even with large memory.

Supplementary Fig. 2. Comparisons of ASW values between scMGCA and 12 single-cell clustering methods on **(a)** 20 real scRNA-seq datasets across multiple platforms and **(b)** on different platforms. The blank bar chart in figures generated by scGAE are due to the fact that scGAE cannot run them even with large memory.

2. The primary metric to compare the performance is NMI; I think authors can add more metrics such as ASW, ARI, and many more that can be found in benchmarking papers (e.g biopreservation metrics here (Luecken et al. 2022)). Additional metrics can help to demonstrate how the method works in clustering rare-cell types or similar cell types along a developmental trajectory.

Response: Thank you very much for the constructive suggestions. We have read the reference (22) with keen interest and agree that additional metrics can help to demonstrate how the method works in clustering rare-cell types or similar cell types along a developmental trajectory. Therefore, we have followed your comments to add ASW and ARI as evaluation metrics for clustering and batch effect performance. The experimental results of the two new evaluation metrics are presented in the reply to **your comment (#1)**. These evaluation metrics are described in **Section O. Benchmarking metrics for scMGCA clustering and visualization**, and they are copied below for your easy reference:

“NMI measures the normalized mutual information between predicted labels and true labels. Let π' denote the predicted clustering and π^G denote the ground-truth clustering. The NMI is defined as:

$$NMI(\pi', \pi^G) = \frac{\sum_{i=1}^{n'} \sum_{j=1}^{n^G} n_{ij} \log \frac{n_{ij} n}{n_i' n_j^G}}{\sqrt{\sum_{i=1}^{n'} n_i' \log \frac{n_i'}{n} \sum_{j=1}^{n^G} n_j^G \log \frac{n_j^G}{n}}} \quad (3)$$

ARI evaluates the similarity of two clustering results in a statistical way, which can be defined as:

$$ARI(\pi', \pi^G) = \frac{\sum_{i=1}^{n'} \sum_{j=1}^{n^G} \binom{n_{ij}}{2} - \sum_{i=1}^{n'} \binom{n_i'}{2} \cdot \sum_{j=1}^{n^G} \binom{n_j^G}{2} / \binom{n}{2}}{\sum_{i=1}^{n'} \binom{n_i'}{2} / 2 + \sum_{j=1}^{n^G} \binom{n_j^G}{2} / 2 - \sum_{i=1}^{n'} \binom{n_i'}{2} \cdot \sum_{j=1}^{n^G} \binom{n_j^G}{2} / \binom{n}{2}} \quad (4)$$

where n' is the number of clusters in the predicted clustering π' , n^G is the number of clusters in the ground-truth clustering π^G , n_i' is the number of data objects in the i -th cluster of π' , n_j^G is the number of data objects in the j -th cluster of π^G , and n_{ij} is the number of data objects that belong to both cluster i in π' and cluster j in π^G .

We also adopt the average silhouette width (ASW) to test our method for dimensionality reduction and visualization, which is defined as below:

$$ASW(x) = \frac{1}{n} \sum_{i=1}^n \left(\frac{b(i) - a(i)}{\max\{a(i), b(i)\}} \right) \quad (5)$$

where $a(i)$ denotes the average distance from x_i to all the other data points in the cluster to which x_i belongs, and $b(i)$ denotes the minimum average distance from x_i to all other clusters to which x_i does not belong. The value of ASW is between $[-1, 1]$. If ASW is close to 1, it means that the clustering of the data object x is reasonable; if ASW is close to -1, it means that the division of x is inaccurate; if ASW is approximately 0, it implies that many data points in x are on the boundary of the two clusters. The ASW is a measure of the reasonableness and validity of the clustering results."

3. How do time and scaling compare with Leiden/Louvain clustering method as the main clustering method used in SCANPY and seurat pipelines? I would also go for a more extensive dataset since a 100k cells dataset is not considered massive, considering datasets with millions of cells that are routinely available.

Response: Thank you very much for the constructive comments. We have followed your comments to apply scMGCA to the mouse brain dataset with 1,306,127 cells and 27,998 genes to compare the running time and memory consumption with other comparative methods, including GNN-based methods (scGNN, scGAE, and GraphSCC), non-GNN-based methods (scziDesk, scDeepCluster, DCA, DEC, DESC, scVI, and scCAEs) and traditional methods (Seurat, SHARP, and SCANPY). With limited computing resources, we set the algorithm to terminate when the running time exceeded 24 hours. We found that most comparison algorithms could not succeed or produce results within 24 hours for such a vast dataset. Ultimately, only scMGCA, DEC, DESC, Seurat, SCANPY, and SHARP were able to run successfully. We present the comparison results in **Supplementary Note 5: Analysis of the mouse brain data with 1.3 million cells**, which is copied below for your easy reference:

Supplementary Note 5: Analysis of the mouse brain data with 1.3 million cells

The original dataset was downloaded from the 10x Genomics website which contains 1,306,127 cells and 27,998 genes: https://support.10xgenomics.com/single-cell-gene-expression/datasets/1.3.0/1M_neurons.

Data preprocessing: We used SCANPY to preprocess the dataset: 1. Cells and genes were filtered using “*scanpy.pp.filter_cells*” with (min_counts=200) and “*scanpy.pp.filter_genes*” with (min_counts=20), respectively. 2. Gene expression levels were normalized using a “*scanpy.pp.normalize_total*” with (target_sum=1e4). 3. The data was log-transformed using “*scanpy.pp.log1p*”. 4. The data after normalization was used with “*scanpy.pp.highly_variable_genes*” (min_mean=0.0125, max_mean=3, min_disp=0.5, n_top_genes = 1000) to identify the top 1000 highly variable genes.

Parameters setting: For the 1.3 million dataset, we used the GCN-based encoder of scMGCA comprising two layers, with the first hidden layer containing 32 nodes and the second hidden layer including 20 nodes. The decoder and encoder are symmetrical, thus the neural network architecture is [1000, 32, 20, 32, 1000]. The initialization center adopted the Leiden algorithm, and the resolution was set to 0.2. The remaining settings included pretrain_epoch = 200, maxiter = 100, and batch = 2000. It is worth noting that the number of training epochs of DEC was the same as that of scMGCA, and other parameters were used by default. DESC used the parameters set in its paper for analyzing 1.3 million dataset. For Seurat, SHARP and SCANPY, we followed the default parameters.

Analysis of the mouse brain data containing 1.3 million cells. Supplementary Fig. 10 depicts the 2D visualization of the latent embedding representation of scMGCA using t-SNE, with the Leiden method identifying 14 clusters. In addition, we compared the running time and memory usage of scMGCA and the compared methods on the 1.3 million cell dataset, including DEC, DESC, Seurat, SHARP, and SCANPY. For additional comparisons, we randomly selected 10,000, 30,000, 50,000, 1,000,000, 1,300,000 cells from the 1.3 million cell dataset, and evaluated the running time and memory usage in these 5 different cases (**Supplementary Fig. 11**). For the traditional methods, the running time is shorter than other methods, which is largely consistent with the evaluation in (1). However, we can observe that Seurat and SHARP require more than 60GB of memory and SCANPY requires nearly 40GB of memory to analyze 1.3 million cells, rendering them inaccessible to the majority of researchers who lack enough computational resources. Meanwhile, although the running time of scMGCA is high, it

is still within a reasonable range and scMGCA is the only GNN-based method (in contrast to scGNN, scGAE, and GraphSCC) that was able to be successfully run on an 1.3 million cell dataset. In terms of memory consumption, DESC outperforms the other methods. The memory consumption of DEC and scMGCA are both smaller than the traditional methods. Interestingly, the time and memory requirements of the deep learning method DESC on large-scale datasets is more balanced than other methods. Possibly, for the deep graph learning method scMGCA, the construction and calculation of cell graph can incur additional time and memory costs, resulting in the extra time and memory consumption accordingly. Moreover, since a small number of pretrain_epoch may not be helpful for further analysis, our model increases the number of pretrain_epoch to 200 as opposed to DESC, which only evaluates pretrain_epoch 10 times on large-scale data; therefore, it greatly increases the final running time.

In summary, we can observe that, excluding the extra time and memory consumption aspects caused by cell graph computation, there is not much difference between scMGCA and deep learning methods. Moreover, it is worth noting that among several graph neural networks compared in this paper, scMGCA is the only one that can run successfully; it can also be regarded as an advancement in deep graph learning method development.

Supplementary Fig. 10. 2D visualization for clustering results of scMGCA on 1.3 million mouse dataset via t-SNE.

Supplementary Fig. 11. Comparison of running time and memory usage of scMGCA to other methods with different numbers of cells.

4. It would be crucial to learn more about how each loss was weighted overall in loss function since this can drastically influence the performance and representation.

Response: Thank you for your critical comments. In our study, the total loss function trained and optimized by scMGCA is defined as follows:

$$\mathcal{L} = \gamma_1 \mathcal{L}_r + \gamma_2 \mathcal{L}_m + \gamma_3 \mathcal{L}_c \quad (6)$$

where \mathcal{L}_r represents the reconstruction loss between the original cell graph (cell-PPMI graph) and the reconstructed cell graph obtained by the inner product decoder; \mathcal{L}_m is the multinomial-based loss that simulates the distribution of the single-cell data by the multinomial-based decoder; \mathcal{L}_c denotes the KL divergence-based clustering loss simultaneously optimized with other losses; γ_1 , γ_2 and γ_3 are weight coefficients assigned to each loss.

As you know, it is extremely difficult to set the weights of different losses in deep learning. Most research is based on the experience of the researchers themselves or on the papers of earlier studies as a guide. Indeed, in this paper, the weight coefficients were chosen (after some preliminary experiments) to give robust results. We collected and summarized different weight coefficients from previous studies (23, 24). On this basis, γ_1 , γ_2 , and γ_3 were selected from [0.3, 0.6, 0.9], [1, 1.5,

2], and [1, 1.5, 2] respectively. Then, to provide comprehensive performance evaluations, we enumerated them to obtain 27 distinct loss weight assignments and compared the clustering performance in those scenarios (**Supplementary Fig. 20**). From **Supplementary Fig. 20**, we can observe that variation in the loss weight over a certain range has little effect on the clustering performance of scMGCA across the majority of datasets. Of these three datasets, ‘QS Trachea’, ‘QS Lung’ and ‘Chen’ are sensitive to the weight coefficients of the loss function with noticeable performance fluctuations. Overall, when γ_1 , γ_2 , and γ_3 are set to [0.3, 1, 1.5], the average NMI and ARI values are the best among those 27 cases.

Supplementary Fig. 20. The clustering performance of different loss weight assignments on 20 single-cell datasets evaluated using NMI and ARI evaluation metrics.

5. how does the model handle the integration and clustering of multiple datasets (or multiple batches within the same dataset) together, accounting for the batch effect?

Response: Thank you very much for the insightful comments. We understand your concerns and have followed your comments to add the experiment to test scMGCA on a combined dataset consisting of four publicly available datasets on human pancreatic tissue generated from four different scRNA-seq protocols, CEL-seq (6), CEL-seq2 (7), Fluidigm C1 (8), and Smart-seq2 (9). The revised manuscript now includes the corresponding paragraphs entitled "**scMGCA can remove the batch effect in human pancreatic data from different scRNA-seq protocols**" on the revised manuscript. We can observe that scMGCA not only exhibits a promising clustering performance for the combined human pancreatic dataset but also effectively removes batch effects. This section has been copied below for your easy reference:

D. scMGCA can remove the batch effect in human pancreatic data from different scRNA-seq protocols. Batch effect correction and clustering of data generated by multiple different scRNA-seq protocols is challenging due to the strong batch differences among different scRNA-seq protocols. To investigate the batch effect of data generated by different scRNA-seq protocols, we combined four publicly available human pancreas datasets generated from CEL-seq (34), CEL-seq2 (35), Fluidigm C1 (36), and Smart-seq2 (37). For benchmark comparison, we selected five state-of-the-art batch effect correction methods including DECS (19), Harmony (38), MNN (39), scVI (20), and Scanorama (40).

In **Fig. 4a**, we observe that scMGCA can effectively merge datasets from different scRNA-seq protocols and remove the batch effect. Compared to other batch effect correction methods, both DESC and Harmony can substantially complete correction of the batch effect, but they do not mix all cells uniformly. scVI could only mix the datasets from CEL-seq and CEL-seq2, with no correction of Fluidigm C1 and Smart-seq2. Scanorama did not remove the batch effect from Smart-seq2, while MNN completely separated the data generated by the four scRNA-seq protocols. Furthermore, **Fig. 4a** also reveals that scMGCA can effectively aggregate cells of the same type and separate cells of different types. DESC also aggregated most cells, with the exception of beta cells which were mixed with alpha cells, acinar cells, and ductal cells. Harmony mixed alpha cells are mixed with acinar cells, delta cells, and beta cells. The three methods (MNN, scVI, and Scanorama), failed in effectively separating the different cell types, resulting in downstream confusion. These findings are also confirmed by the comparison of the clustering performance as depicted in **Fig. 4b**. Overall, scMGCA and DESC are superior to other batch effect correction methods, whereas MNN, scVI, and Scanorama perform poorly across those three metrics: NMI, ARI, and ASW.

To explore the process by which scMGCA removes batch effects, we visualize the latent embedding representation of scMGCA at different epochs. As shown in **Fig. 4c**, scMGCA is capable of aggregating the cells and gradually mixing the datasets generated by the various scRNA-seq protocols. This demonstrates that scMGCA not only accurately clusters cells but also simultaneously corrects for batch effect in multiple datasets with strong batch differences.

Fig. 4. Comparison of batch effect correction methods on the pancreatic islet data generated by different scRNA-seq protocols for batch effect correction via UMAP. The cell batches are colored in the top row and cell types in the bottom row. **b** Comparison of clustering performance with evaluation metrics NMI, ARI and ASW. **c** UMAP plots showing scMGCA removes the batch effect gradually over iterations.

6. how did the authors perform hyperparameter optimization, and how would this affect the user?

Response: Thank you very much for the insightful comments. Since the other reviewers also had some comments on hyperparameter optimization, we evaluated the impact of all the hyperparameter choices on the performance of scMGCA to investigate the effect of each hyperparameter. In scMGCA, there are several hyperparameters that need to be adjusted, including the number of highly variable genes (d), the initial number of neighbors (k) in constructing the cell-PPMI graph, the way of calculating

the distance, the transfer step (s), the number of GCN layers of scMGCA, and the three weights in the scMGCA loss function.

Firstly, we conducted an experiment to examine scMGCA on different numbers of highly variable genes (d) in six cases (300, 500, 1000, 1500, 2000, and all genes), as depicted in **Fig. 5b**, which is in the response to Reviewer #1's comment 3. Then, we also evaluated three hyperparameters required to build the cell-PPMI graph, namely the initial number of neighbors (k), the way to calculate the distance, and the transfer step (s). We enumerated these three parameters into 36 different cases and compared the clustering performance of each case. The experimental results as depicted in **Supplementary Fig. 21** are shown in the response to Reviewer #3's comment 5. Another hyperparameter is the number of GCN layers of scMGCA that was also evaluated, and the 1, 2, 3 and 4 layers were benchmarked to compare the clustering performance, and the experimental results as summarized in **Supplementary Fig. 22** were shown in the response to Reviewer #1's comment 4. Finally, for the analysis of the three weights in the scMGCA loss function, a total of 27 different situations were enumerated to compare clustering performance, and the experimental results as visualized in **Supplementary Fig. 20** are presented in the response to Reviewer #2's comment 4.

In general, these parameters have no obvious changes in the effect on the model within reasonable ranges. However, it might be difficult for users who have no extended expertise in graph deep learning to design optimal graph deep learning architectures for their own biomolecular data of interest. Therefore, we also implemented and provided an automatic hyperparameter search interface (**the code is available at <https://github.com/Philyzh8/scMGCA>**) for users that do not have domain knowledge to optimize the set of hyperparameters. Even then, with a large number of hyperparameter combinations, this inevitably leads to an excessive amount of tests, consuming significant computational and time costs. On this basis, we recommend that users who are unfamiliar with graph deep learning use the automatic hyperparameter search to optimize hyperparameters if they have sufficient computational and time resources, whereas other users are encouraged to tune hyperparameters around the default settings we have carefully set.

7. *when does the algorithm break? It would be great to show some failure cases to help the users.*

Response: Thank you very much for the critical suggestions. As suggested by the reviewers, we have added the following suggestions for the user-friendly uses of scMGCA. Those suggestions are also added to the documentation along with the source code.

Dataset: We recommend that users to adopt the standard h5 format data (also known as HDF5), which is common among data preprocessing and feature selection. Data in other formats (such as .csv, .txt) can be converted into h5 format data for input.

Pre-processing: In most cases, the scRNA-seq datasets always need different data preprocessing steps, such as cell gene filtering, logarithmization, highly-variable genes selection. These are aggregated into an internal function named *normalize*. It is worth noting that the rule for selecting the number of highly variable genes is not less than 50 since less than 50 can cause errors in building the cell-PPMI graph. The reason being that in order to calculate the KNN graph faster when building the cell graph, we performed the PCA operation on the data and reduced the dimension to 50 dimensions. Therefore, if the data itself

is less than 50 dimensions, the code will report the *ValueError*. However, in reality, the use cases with less than 50 genes are pretty rare in practice. One of the workarounds is to modify the PCA source code to be less than 50 dimensions.

Cell graph: The cell-PPMI graph is obtained by calculating the co-occurrence probability of cells through PPMI which is a symmetric undirected graph. Therefore, if the user wants to use their own cell graph for input, we recommend a symmetric undirected graph as the cell-PPMI graph. Furthermore, it is necessary to ensure that the parameter k in the function ‘*get_adj*’ is less than or equal to the number of cells in the dataset ($n_neighbors \leq n_samples$); otherwise, it will report the *ValueError*.

Batch settings: scMGCA performs batch learning for the datasets with more than 25,000 cells. Therefore, when using scMGCA to cluster the dataset with more than 25,000 cells, the user should select the function ‘*get_adj_batch*’ in “graph_function.py” to construct the cell graph and the class ‘*SCMGCAL*’ in “scmgca.py” as the model for calculation; otherwise, an out of memory (OOM) error can occur if memory resources are not enough.

Minor:

1. *There are some typos in the text (e.g evolution metric in section B)*

Response: Thank you very much for the helpful comments. We apologize for the typos in the text, we have checked and corrected. To fix the grammar mistakes, we have asked a native English speaker to proofread the revised manuscript. Moreover, we (the first author and the corresponding authors) have proofread the manuscript for multiple rounds again. English grammatical errors and typos have been fixed to our best effort.

2. *It seems some visible lines are not part of any panel in figure 2, which might be the artifact of the graphical post-processing app.*

Response: Thank you very much for the critical comments. We apologize for the problem with **Fig. 2**. Following the reviewers’ comments, we have checked the issues and redrawn **Fig. 2**.

Fig. 2. Cell clustering performance of scMGCA compared to other state-of-the-art clustering methods. **a** Comparisons of NMI values between scMGCA and 12 single-cell clustering methods on 20 real scRNA-seq datasets across multiple platforms (left) and on different platforms (right). **b** Comparison of clustering results with 2D visualization by UMAP on the 'Qx Limb Muscle' dataset. **c** Comparison of the running time between scMGCA and other state-of-the-art clustering methods. **d** The clustering performance of all 13 methods on the 'Tabula Muris' dataset. The blank bar chart in figures generated by scGAE are due to the fact that scGAE cannot run them even with large memory.

Reviewer #3:

Correcting biological information loss in scRNA-Seq analysis is an important challenge for the bioinformatic researchers. In this manuscript, the authors developed a method (scMGCA) to solve this kind of problems, specifically utilizing cell-graph autoencoder approach to remedy single-cell dropout events and to optimize dimensionality reduction and cell clustering analyses. Through comparison with other existing tools in multiple real data, the authors displayed the outperformance of scMGCA. However, the architecture of scMGCA autoencoder is similar to other existing tool scGNN, the authors may need to directly compare scMGCA with other cell-graph-neural-network tools (at least scGNN) in core steps and show the specificity of scMGCA. Moreover, convergence is one character of deep learning, the converged output of cell-graph autoencoder may weaken the cell heterogeneity of the same cell-type of cells. In the manuscript, the results of scMGCA showed some potential similar problems that the cells in transition status could not be visualized and identified demonstrably.

Response: Thank you very much for the critical and helpful comments. For the past four months, we have duly followed your comments and suggestions to address all the concerns as elaborated in the following responses. In particular, we would like to thank you for your critical comments which have improved the manuscript in numerous ways. Thank you very much for your precious time and kind attention.

1. For my understanding, scMGCA and scGNN utilize similar architecture to build graph convolutional network, especially the key processes and relationships between cell-graph, cell clustering and feature imputation (the details in scMGCA Results part A and scGNN Results “The architecture of scGNN . . . ” as well as the corresponding Methods parts) 1. The authors might need to directly explain what difference in the autoencoder between scMGCA and scGNN and what advantages of scMGCA.

I understand the authors described the key steps and mathematic models of scMGCA in the manuscript, especially in Results part A and Discussion, but without explaining the superiority of these approaches by straightforward comparison with other tools, it is difficult for the audience to estimate the innovation of scMGCA. The authors might learn from other single-cell tool publications. For instance, the architecture of scDeepCluster is similar to DCA, the scDeepCluster paper gives direct discussion and explain the difference (page 192) 2,3.

Response: Thank you very much for the suggestions. We appreciate and agree with you. We have followed your comments to directly explain the difference in the autoencoder between scMGCA and scGNN and the advantages of scMGCA.

Compared with scGNN, our algorithm has three important differences. Firstly, scGNN uses the traditional KNN graph as a cell graph, while scMGCA proposes the cell-PPMI graph as a cell graph to apply the random surfing model on the basis of the KNN graph to capture the cell topology information and generate a probabilistic co-occurrence matrix. After that, we calculate its positive pointwise mutual information (PPMI) matrix to further augment the relationship of similar cells. Indeed, compared to the Euclidean distance between cells in scGNN, the co-occurrence probability between cells obtained by random surfing and the PPMI matrix can effectively capture the important relationships and topology between cells, which is also verified in the subsequent experimental analysis, as elucidated in **Supplementary Fig. 26**.

Secondly, single-cell clustering models are often challenged by the high probability of dropout events in single-cell data due to low capture rates and insufficient sequencing depth. To address this limitation, scMGCA proposes a multinomial-based graph convolutional autoencoder to address this problem, whereas scGNN uses an imputation autoencoder with the additional L1 regularizer to generate the reconstructed gene expression values, which is the difference between our algorithm and scGNN. In particular, the multinomial-based graph convolutional autoencoder adopts a multinomial distribution to capture high-probability dropout events in single cell data and then uses the negative log-likelihood of the multinomial distribution as the objective function to optimize and train the entire network when decoding the latent representation. As a result, the GCN can extract the latent representation that fits the distribution of the single-cell data to improve clustering performance, which is also confirmed in the dropout analysis, as demonstrated in **Supplementary Figs. 27-28**.

Finally, from the perspective of combining deep learning and clustering algorithms, scGNN is typically a two-stage deep clustering method; in the first stage, graph autoencoders are used to learn the features of the data, and in the second stage, a traditional clustering algorithm such as Louvain or Hierarchical Clustering is chosen to cluster the latent representation yielded by the graph autoencoders. In contrast to the two-stage deep clustering method (scGNN), our scMGCA is a joint deep clustering method specific for scRNA-seq data to train the model and then simultaneously learns cell-cell topology representations and cluster assignments. In particular, we designed three training losses, including the clustering loss, multinomial-based loss, and cell graph reconstruction loss, to jointly optimize deep graph embedding and clustering to elucidate scRNA-seq data across different platforms. Thus, we can see that scMGCA optimizes feature learning and clustering as a whole, whereas scGNN separates feature learning and clustering. Indeed, according to the previous literature review (23, 25), the clustering performance of joint learning is commonly superior to that of segmented algorithms.

2. According to Fig. 1 and Methods part H, the authors utilized random surfing model to generate PPMI matrix, which estimates co-occurrence probability between cells, for building cell-graph; then scMGCA constructed GCN encoder by the cell-graph and gene expression matrix. I respect the authors' efforts and work, but I am curious why the encoder cannot be from cell-graph by KNN with gene expression matrix weighted directly. As the author mentioned in Methods part H and equation 5 7, PPMI reflects the co-occurrence probability of cells and the higher PMI score represents the closer relationship between cells, "Therefore, the cells of the same type in the biological network have higher weights in the cell graph". But, for my understanding, the KNN graph can achieve this kind of similarity metrics directly 4. Because PPMI is important step for scMGCA to build GCN encoder, the authors may elaborate the significance.

Response: Thank you very much for your insightful comment. Yes, you are perfectly right that the KNN graph can also be constructed directly for the similarity metrics. However, there are several differences in both the methodology principles and molecular motivations between the two graphs to achieve this goal of bringing homogeneous cells close together and separating heterogeneous cells. To address your comments, the differences and advantages of the cell-PPMI graph compared to the KNN map are briefly summarized below. After that, a comparison of scMGCA(PPMI) and scMGCA(KNN) is provided for your easy reference.

For constructing a cell graph through KNN, the distance between each cell and other cells is calculated, and the top k closest

cells selected as neighbors, assigned higher weight edge connections, and they have no edge connections with the other remaining cells (weight is 0). Indeed, the KNN graph, as the reviewer points out, is straightforward in its construction. However, according to some previous studies (23, 24), the KNN graph, which directly assigns weights only by the distance between two nodes, is more suitable for completely random networks. For the single-cell network composed of cells with homogeneity and consistency, the KNN graph may not be compatible with the single-cell network properties from different platforms. The reason could be that the KNN graph only focuses on the relationship between two cells and does not perceive the underlying information of surrounding cells (e.g. cell clusters), which often also plays an important role in the single-cell network.

The cell-PPMI graph in scMGCA can address the shortcomings of KNN graph and can also strengthen the aggregated cell neighborhood. The construction of the cell-PPMI graph is mainly divided into three parts. First, the cell-PPMI graph starts from an unweighted KNN graph \mathbf{K} , which is defined as follows:

$$K_{ij} = \begin{cases} 1, & j \in \mathcal{N}_k(i) \\ 0, & j \notin \mathcal{N}_k(i) \end{cases} \quad (7)$$

where $\mathcal{N}_k(i)$ represents the neighborhood containing the nearest k nodes with the i -th node as the center. This step mainly uses the KNN algorithm to initialize the neighborhood of each cell for the cell-PPMI graph. Then, we use the random surfing algorithm to process \mathbf{K} to obtain the transition matrix \mathcal{P}^s , which is defined as follows:

$$\begin{aligned} \mathcal{P}^s &= \alpha \cdot \mathcal{P}^{s-1} \mathbf{K} + (1 - \alpha) \mathcal{P}^0 \\ &= \alpha^s \cdot \mathcal{P}^0 \mathbf{K}^s + (1 - \alpha) \sum_{t=1}^s \alpha^{s-t} \cdot \mathcal{P}^0 \mathbf{K}^{s-t} \end{aligned} \quad (8)$$

where s is the transfer step ($s \leq 3$), and \mathcal{P}^0 is the initial transition matrix equal to the identity matrix ($\mathcal{P}^0 = \mathcal{I}$). The probability of returning to the initial vertex and restarting the process is set to $(1 - \alpha)$. \mathcal{P}_{ij}^s represents the probability of the i -th node reaching the j -th node after s steps of transitions. By transferring and walking in the network, the random surfing algorithm obtains the local and global topology among neighboring cells. Finally, we sum all the transition matrices after s steps to get the probability matrix $\mathbf{M} = \sum_{s=1}^S$, and use PPMI to further augment the graph, which is defined as follows:

$$\begin{aligned} PPMI(x, y) &= \max(PMI(x, y), 0) \\ &= \max\left(\log\left(\frac{p(x, y) \cdot |\mathbf{S}|}{p(x)p(y)}\right), 0\right) \end{aligned} \quad (9)$$

where $|\mathbf{S}| = \sum_x \sum_y p(x, y)$, x , and y represent different nodes; $p(x)$ and $p(y)$ denote the probability of x and y , respectively; $p(x, y)$ represents the co-occurrence probability of x and y . These probabilities are all obtained from \mathbf{M} , so the final cell-PPMI graph \mathbf{A} is defined as follows:

$$\mathbf{A} = \max\left(\log\left(\frac{\mathbf{M}\Theta}{\text{row}(\mathbf{M})\text{col}(\mathbf{M})}\right), 0\right) \quad (10)$$

where Θ denotes the sum of all elements in \mathbf{M} . $\text{row}(\mathbf{M})$ and $\text{col}(\mathbf{M})$ are the column vector of the sum of each row of \mathbf{M} and the row vector of the sum of each column of \mathbf{M} , respectively. To verify our point, we first demonstrate that the cell-PPMI graph can uncover underlying information in the cell network that cannot be perceived by the KNN method in the analytical manner.

Mathematically, the KNN graph directly assigns weights by the distance between two cells, and it is easy to ignore potential neighbor information (such as neighbors' neighbors) due to the selection of the number of neighbors and the calculation of the distance. Suppose that in the KNN graph cell i has an edge with cell j , and cell j has an edge with cell m , but i has no edge with m and m is a potential neighbor of i . We now demonstrate that the cell-PPMI graph is capable of discovering cell k . According to equation 8, since $(1 - \alpha)\mathcal{P}^0$ only changes the value of the diagonal of \mathcal{P}^s , it does not change the matrix structure of \mathcal{P}^s , and does not affect matrix multiplication, so we can conclude that \mathcal{P}^s is equivalent to \mathbf{K}^s , that is:

$$\mathcal{P}^s \iff \mathcal{P}^0(\underbrace{\mathbf{K} \cdot \mathbf{K} \cdots \mathbf{K}}_s) \iff \mathbf{K}^s \quad (11)$$

According to the previous assumption, both K_{ij} and K_{jm} are equal to 1 and K_{im} is equal to 0. However, when \mathbf{K} is exponentiated, the potential neighbor cell m of cell i is found, as follows:

$$\begin{aligned} (\mathbf{K} \cdot \mathbf{K})_{im} &= \sum_{c=1}^n K_{ic} \cdot K_{cm} \\ &= \sum_{\substack{c=1 \\ c \neq j}}^n K_{ic} \cdot K_{cm} + K_{ij} \cdot K_{jm} \\ &\geq K_{ij} \cdot K_{jm} > 0 \end{aligned} \quad (12)$$

where n is the total number of cells. According to equation 12, it can be induced that the random surfing algorithm finds the information of cell i and cell m that cannot be found in the KNN graph through their common neighbor cell j when transferring and walking.

We now deduce that the cell-PPMI graph can aggregate neighboring cells to augment the cell graph. Equation 12 can also be rewritten as follows:

$$\begin{aligned} (\mathbf{K} \cdot \mathbf{K})_{ij} &= \sum_{c=1}^n K_{ic} \cdot K_{cj} \\ &= \sum_{\substack{c=1 \\ c \neq a}}^n K_{ic} \cdot K_{cj} + \sum_{\substack{a \in \mathcal{N}_k(i) \\ a \in \mathcal{N}_k(j)}}^n K_{ia} \cdot K_{aj} \\ &\geq \sum_{\substack{a \in \mathcal{N}_k(i) \\ a \in \mathcal{N}_k(j)}}^n K_{ia} \cdot K_{aj} \end{aligned} \quad (13)$$

where cell a is the common neighbor of cell i and cell j . According to equation 13, it is proved that the weight between cell i and cell j in the random surfing algorithm is affected by the number of common neighbor cells. In other words, the random surfing algorithm amplifies the weights between closely related cells, aggregates neighboring cells, and propagates this information to the probability matrix \mathbf{M} . Furthermore, according to equation 10, PPMI can stretch the low-weight part by logarithmic transformation and remove it to augment the cell graph.

Apart from this, we also provide the comparison results for scMGCA(PPMI) and scMGCA(KNN) on those 20 scRNA-seq

data, as depicted in Supplementary Fig. 26. From the results, the clustering performance of scMGCA(PPMI) outperforms scMGCA(KNN) on most datasets. It further demonstrates that the cell-PPMI graph is more suitable as a cell graph than the KNN graph.

Supplementary Fig. 26. Comparison of NMI and ARI between scMGCA(PPMI) and scMGCA(KNN) on 20 datasets.

3. In cell clustering and dimensionality reduction parts (Results part B and C), the authors displayed the outperformance of scMGCA in segregating different types of cells. However, the results in these two parts show some potential problems that scMGCA may artificially enlarge the dissimilarity of cell types and the cell heterogeneity of the same type of cells cannot be preserved in scMGCA visualization. Specifically, “Camp-Brain” data in the original publication is about progenitor-to-neuron differentiation, and a part of the single-cell data were generated from induced stem cells (GSE75140) 5. In Supplementary Fig. 2 and S7 of this manuscript, most tools, except scMGCA, showed the blurred boundaries between cell types and reflected the process of cell differentiation, consistent with the analysis of the original paper 5. By contrast, scMGCA separated cell type independently and scMGCA visualization may distort the cell relationships. Similar situation can also be found in the results of “Klein” data. I suggest the authors improve scMGCA performance in this area, at least displaying similar results to other

single-cell tools or dimensionality reduction methods.

I noticed that the authors showed cell trajectory in Fig. 5h, but the cells in the panels do not have logical biological connections (i.e., the cells belong to different cell types, it is difficult to estimate whether trajectory connections in Fig.5h are appropriate) 6. Further, Fig.5h panels show the output of monocle-2 dimensionality reduction, but most scMGCA dimensionality reduction plots do not contain cell-cell connections (Supplementary Fig. 7-S10). The authors could find the relative topics in UMAP paper 7 or Phate paper 8.

Response: Thank you very much for the insightful comments which are also of our interests to examine. Those boundaries are actually generated from the adopted visualization technique (**UMAP**) but not from the algorithmic core of scMGCA.

The algorithmic core of scMGCA is a deep-embedded graph clustering method based on graph autoencoders, and its loss function is designed to allow related cells to aggregate and unrelated cells to separate. The scMGCA loss function mainly includes three parts: (1) the reconstruction loss between the original cell graph and the reconstructed cell graph obtained by the inner product decoder; (2) the multinomial-based loss that simulates the distribution of single-cell data by the multinomial-based decoder; (3) the KL divergence-based clustering loss simultaneously optimized with other losses. Through the optimization of the combined loss function, scMGCA obtains a low-dimensional latent representation that preserves the key gene expression information and topology of the original data. We use **UMAP** to visualize the latent representations in 2D.

The reviewer pointed out that the boundaries generated by cell differentiation are missing from the visualization. We believe that the reason is because of the parameter setting of **UMAP** but not scMGCA. To examine and address this, we did the following experiments. Specifically, we changed the parameters of the UMAP function from $min_dist = 0.1$ to $min_dist = 1$ as visualized below (**Fig. R1**). As observed from the figure, scMGCA retains the boundaries of different types of cells, but the boundaries are “lost” due to the parameters of the visualization method in use.

In addition, we also adjusted the parameters of UMAP to visualize the ‘Qx Limb Muscle’ dataset in **Fig. R2**. It can be observed that the clustered ‘Qx Limb Muscle’ dataset by scMGCA still has biological connections. Further, to address your concerns, we performed trajectory inference and pseudo-temporal analysis on the gene expression matrix composed of those 200 interpretable genes extracted by scMGCA on the ‘Qx Limb Muscle’ dataset (**Fig. 6h**). The analysis of the experimental results in the paper has also been revised, and they are copied for your easy reference:

Finally, we applied the Monocle3 (16) algorithm to the gene expression matrix consisting of these 200 interpretable genes to analyze the developmental trajectory of the ‘Qx Limb Muscle’ dataset in pseudo-time (**Fig. 6h**). By measuring the progression of biological processes based on transcriptional similarity, we found that the developmental trajectory mainly starts with mainly mesenchymal cells, then moves to B cells, endothelial cells, and T cells.

Fig. R1. Visualization of latent embedding representations of scMGCA with different parameters of UMAP.

Fig. R2. Visualization of latent embedding representations of scMGCA with different parameters of UMAP on the 'Qx Limb Muscle' dataset.

h

Fig. 6h. Trajectory inference of selected genes.

4. Although correcting single-cell dropout issue is one of the targets in this study, I found the interpretation and analyses of scMGCA imputed data were very limited in this manuscript. It is difficult to estimate the performance of scMGCA in dropout-event correction.

Response: Thank you very much for your comments. We agree that more work is required to estimate the performance of scMGCA in dropout-event correction. Therefore, we have added experiments to compare the clustering performance of scMGCA with that of 10 other deep learning clustering approaches as the single-cell data dropout rate is increased. Following the reference (26), we evaluated the different methods by randomly masking the non-zero gene expression information of each cell (replacing with zeros) on single-cell data and then simulating the increase in the probability of data dropout as the masking probability increases. We test the 20 real single-cell datasets where the rate of increase in dropout events is set to [0%, 5%, 10%, 20%, 50%]. The comparison results are shown in **Supplementary Figs. 27-28**. On the majority of the datasets, it can be observed that scMGCA has a small decrease in NMI and ARI values and outperforms all other compared methods. In particular, the clustering performance of scGNN decreases significantly when the dropout events is increased from 20% to 50% on many datasets. These experimental findings confirm that scMGCA is sufficiently stable and robust with promising performance in dropout event correction.

Supplementary Fig. 27. The NMI values of scMGCA with different probabilities of artificial dropout events on 20 real single-cell datasets.

Supplementary Fig. 28. The ARI values of scMGCA with different probabilities of artificial dropout events on 20 real single-cell datasets.

5. The authors evaluated the stability of scMGCA with different parameters in Results part D, but the discussion of how to build cell graph may not be sufficient. I encourage the authors could provide some suggestions and strategies for the users to optimize parameters to construct cell graph.

Response: Thank you very much for the constructive suggestions. The cell graph (cell-PPMI graph) in scMGCA is mainly divided into three parts: 1. Convert the gene expression matrix into an unweighted KNN graph through the KNN algorithm, and determine the preliminary neighborhood of each cell; 2. Capture the transition probabilities between different cells from the KNN graph structural information using a random surfing model and then aggregate the adjacent cells; 3. The transition probability matrix is summed within a finite transfer step to obtain the probability matrix, and its positive point mutual information (PPMI) matrix is calculated to characterize similar cells, and finally the cell graph is further enhanced by assigning co-occurrence attention mechanisms.

To demonstrate clearly the performance of different parameters in the cell graph construction, we first supplemented the experiments by adjusting the three parameters in the construction of cell graph. The first parameter is the initial number of neighbors (k) when building the KNN graph, which we choose from [10, 15, 20]. The second parameter is the distance function (m) to determine the neighbors of each cell when building the KNN graph, where we have four distance functions to choose including the Euclidean distance, the cosine distance, the Pearson correlation coefficient, and the Minkowski distance. The last parameter is the number of transfer steps (s) in the random surfing to determine the scope of search and aggregation, which we choose from [1, 2, 3] for our analysis. To provide comprehensive comparisons, we compare the clustering performance of different combinations of these three parameters (36 cases) using the NMI and ARI evaluation metrics (**Supplementary Fig. 21**). The experimental results showed that the average NMI and ARI for the 20 datasets were ranged from 0.794 to 0.831 and from 0.748 to 0.828, respectively. Indeed, we can observe that the clustering effect is the best at “ $k=15$, $m=Euclidean$, $s=2$ ”, with the highest average NMI and ARI values. Therefore, we adopted this parameter setting when constructing the cell-PPMI graph.

For users in similar fields, we recommend using the default parameters we set for graph construction or fine-tune around the default parameters we set based on other biomolecular data characteristics. In addition, as per another reviewer’s suggestion, we also implemented and provided an automatic hyperparameter search interface (**the code is available at <https://github.com/Philyzh8/scMGCA>**) for users that have no extended expertise in graph deep learning to optimize the hyperparameters.

Supplementary Fig. 21. The clustering performance of different parameters for building the cell-PPMI graph on 20 single-cell datasets evaluated by NMI and ARI evaluation.

Minor points:

1. *Seurat and SCANPY employ other clustering algorithms in their toolkits. When comparing their cluster results (Fig. 2a and 3b), the authors need to clarify what algorithms is used in the manuscript. For SCANPY, <https://SCANPY.readthedocs.io/en/stable/api.html#clustering-and-trajectory-inference> For Seurat, please notice the parameters “method” and “algorithm” library(Seurat) help(“FindClusters”)*

Response: Thank you very much for your insightful comments. We agree with you and have followed your comments to describe the algorithms and parameter settings used in SCANPY and Seurat in **Supplementary Note 8**, which is copied below for your easy reference:

Supplementary Note 8: Clustering algorithms and parameter settings for SCANPY and Seurat

SCANPY: The version of SCANPY used is 1.6.0. For preprocessing the data, we used “*scanpy.pp.normalize_per_cell*” function to normalize gene expression levels for each cell and “*scanpy.pp.log1p*” function to log-transform the data. Then, after normalization the data used “*scanpy.pp.highly_variable_genes*” function with (min_mean=0.0125, max_mean=3, min_disp=0.5, n_top_genes = 500) to select the top 500 highly variable genes. For clustering the data, we first used “*scanpy.tl.pca*” function with (svd_solver='arpark') and “*scanpy.pp.neighbors*” function with (n_neighbors=10, n_pcs=40) to find the neighborhood of the data, and then used “*scanpy.tl.leiden*” function with (resolution = 1, random_state = 0) for clustering.

Seurat: The version of Seurat used is 4.0.0. For preprocessing data, we used “*log2*” and “*ScaleData*” functions to normalize the data. After normalization, the data used “*RunPCA*” and “*FindNeighbors*” functions with (dims = 1:10) to find the neighborhood of the data, and then used “*FindClusters*” function with (resolution = 0.8, method = "matrix", algorithm = 1) for clustering.

2. In Results, the authors mentioned “*Monocle*” in part B and “*Monocle2*” in part E, but in part M of Methods, the authors provided the link of “*Monocle3*”. The authors may need to unify what Monocle version you used in the study. If using Monocle2, it also employs other cluster method. <http://www.bioconductor.org/packages/release/bioc/manuals/monocle/man/monocle.pdf>
The link for Monocle2 <http://cole-trapnell-lab.github.io/monocle-release/docs/>

Response: Thank you very much for your insightful comments. We apologize for this mistake. In our previous manuscript, we used the Monocle2 for clustering and trajectory inference. Regarding your comment (3), since the Monocle3 can provide a more intuitive trajectory inference and pseudo-temporal analysis to verify the biological connections, we have changed the Monocle to version 3 and updated the reference link in this revised manuscript. The revised section "**Trajectory inference**" is copied below for your easy reference:

R. Trajectory inference. The trajectory analysis was performed in the R package, Monocle3 (<http://cole-trapnell-lab.github.io/monocle-release/monocle3/>). In **Fig. 6h**, we extracted an expression matrix of 200 biologically interpretable genes as input and used the labels predicted by scMGCA clustering. After the monocle loads the data by function “*new_cell_data_set*”, the data is preprocessed by function “*preprocess_cds*”. Then, the data is dimensionally reduced by function “*reduce_dimension*”, where the parameters are set to “*reduction_method = ‘UMAP’, preprocess_method = ‘PCA’*”. Further, the cells are clustered by function “*cluster_cells*”, where the parameters are set to “*reduction_method = ‘UMAP’*”, and the principal graph is learned by function “*learn_graph*”, where the parameters are set to “*use_partition = ‘F’, close_loop = ‘F’, learn_graph_control = ‘NULL’, verbose = ‘FALSE’*”. Finally, the function “*plot_cells*” is used to visualize the trajectory inference.

3. In Introduction and Results part A, the authors' interpretation ("To address these limitations, deep embedding clustering ...") and ("scMGCA can learn the low-dimensional representation ...") are not consistent.

Response: Thank you very much for the sharp catch. We have revised the sentence "scMGCA can learn the low-dimensional representation ... " to make it consistent with the previous sentence. Moreover, we have proofread the manuscript several times to avoid similar representations in the revised manuscript. It is copied below for your easy reference:

"scMGCA can learn the low-dimensional representation from high-dimensional and sparse scRNA-seq data".

4. The subtitle "scMGCA is the most accurate of ..." is not appropriate.

Response: Thank you very much for the helpful comments. We have replaced the "scMGCA is the most accurate of the single-cell clustering methods across multiple platforms" with "scMGCA provides better performance than other single-cell clustering methods across multiple platforms" in the revised manuscript.

5. In Results part D, the sentence "... when $s=2$ and $d=2$..." needs to be corrected.

Response: Thank you very much for the helpful comments. We apologize for the oversight and have corrected this sentence in an exact manner.

"The results demonstrate that scMGCA has the best overall clustering performance when $s=2$ and $d=500$ compared to other conditions. "

References

1. Xiangjie Li, Kui Wang, Yafei Lyu, Huize Pan, Jingxiao Zhang, Dwight Stambolian, Katalin Susztak, Muredach P Reilly, Gang Hu, and Mingyao Li. Deep learning enables accurate clustering with batch effect removal in single-cell rna-seq analysis. *Nature communications*, 11(1):1–14, 2020.
2. Huazhu Fu, Hang Xu, Kelvin Chong, Mengwei Li, Kok Siong Ang, Hong Kai Lee, Jingjing Ling, Ao Chen, Ling Shao, Longqi Liu, et al. Unsupervised spatially embedded deep representation of spatial transcriptomics. *Biorxiv*, 2021.
3. Mònica Comalada, Marina Cardó, Jordi Xaus, Annabel F Valledor, Jorge Lloberas, Francesc Ventura, and Antonio Celada. Decorin reverses the repressive effect of autocrine-produced $\text{tgf-}\beta$ on mouse macrophage activation. *The Journal of Immunology*, 170(9):4450–4456, 2003.
4. Elliot H Akama-Garren, Theo van den Broek, Lea Simoni, Carlos Castrillon, Cees E van der Poel, and Michael C Carroll. Follicular t cells are clonally and transcriptionally distinct in b cell-driven mouse autoimmune disease. *Nature communications*, 12(1):1–19, 2021.
5. E Somm, P Cettour-Rose, C Asensio, A Charollais, M Klein, C Theander-Carrillo, CE Juge-Aubry, J-M Dayer, MJH Nicklin, P Meda, et al. Interleukin-1 receptor antagonist is upregulated during diet-induced obesity and regulates insulin sensitivity in rodents. *Diabetologia*, 49(2):387–393, 2006.
6. Dominic Grün, Mauro J Muraro, Jean-Charles Boisset, Kay Wiebrands, Anna Lyubimova, Gitanjali Dharmadhikari, Maaike van den Born, Johan Van Es, Erik Jansen, Hans Clevers, et al. De novo prediction of stem cell identity using single-cell transcriptome data. *Cell stem cell*, 19(2):266–277, 2016.
7. Mauro J Muraro, Gitanjali Dharmadhikari, Dominic Grün, Nathalie Groen, Tim Dielen, Erik Jansen, Leon Van Gurp, Marten A Engelse, Françoise Carlotti, Eelco Jp De Koning, et al. A single-cell transcriptome atlas of the human pancreas. *Cell systems*, 3(4):385–394, 2016.
8. Nathan Lawlor, Joshy George, Mohan Bolisetty, Romy Kursawe, Lili Sun, V Sivakamasundari, Ina Kycia, Paul Robson, and Michael L Stitzel. Single-cell transcriptomes identify human islet cell signatures and reveal cell-type-specific expression changes in type 2 diabetes. *Genome research*, 27(2):208–222, 2017.
9. Åsa Segerstolpe, Athanasia Palasantza, Pernilla Eliasson, Eva-Marie Andersson, Anne-Christine Andréasson, Xiaoyan Sun, Simone Picelli, Alan Sabirsh, Maryam Clausen, Magnus K Bjursell, et al. Single-cell transcriptome profiling of human pancreatic islets in health and type 2 diabetes. *Cell metabolism*, 24(4):593–607, 2016.
10. Dvir Aran, Agnieszka P Looney, Leqian Liu, Esther Wu, Valerie Fong, Austin Hsu, Suzanna Chak, Ram P Naikawadi, Paul J Wolters, Adam R Abate, et al. Reference-based analysis of lung single-cell sequencing reveals a transitional profibrotic macrophage. *Nature immunology*, 20(2):163–172, 2019.
11. Kyung Cho Cho, Ralph Hruban, Hui Zhang, Chen Huang, Bing Zhang, Daniel Cui Zhou, Li Ding, Emily Boja, and Investigators from the Clinical Proteomic Tumor Analysis Consortium. Abstract a60: Integrated proteogenomic characterization of pancreatic ductal adenocarcinoma. *Cancer Research*, 79(24_Supplement):A60–A60, 2019.

12. Yiping He, Yan Liu, Jianping Gong, Changan Liu, Hua Zhang, and Hao Wu. Identification of key pathways and candidate genes in pancreatic ductal adenocarcinoma using bioinformatics analysis. *Oncology letters*, 17(4):3751–3764, 2019.
13. Nan Li, Xin Zhao, and Shengyi You. Identification of key regulators of pancreatic ductal adenocarcinoma using bioinformatics analysis of microarray data. *Medicine*, 98(2), 2019.
14. DD Wang, ZW Liu, MM Han, ZM Zhu, YL Tu, CQ Dou, X Jin, SW Cai, and N Du. Microarray based analysis of gene expression patterns in pancreatic neuroendocrine tumors. *Eur Rev Med Pharmacol Sci*, 19(18):3367–3374, 2015.
15. Jonas RM Van Audenaerde, Jorrit De Waele, Elly Marcq, Jinthe Van Loenhout, Eva Lion, Johan MJ Van den Bergh, Ralf Jesenofsky, Atsushi Masamune, Geert Roeyen, Patrick Pauwels, et al. Interleukin-15 stimulates natural killer cell-mediated killing of both human pancreatic cancer and stellate cells. *Oncotarget*, 8(34):56968, 2017.
16. Jianyu Yang, Ping Lin, Minwei Yang, Wei Liu, Xueliang Fu, Dejun Liu, Lingye Tao, Yanmiao Huo, Junfeng Zhang, Rong Hua, et al. Integrated genomic and transcriptomic analysis reveals unique characteristics of hepatic metastases and pro-metastatic role of complement c1q in pancreatic ductal adenocarcinoma. *Genome biology*, 22(1):1–20, 2021.
17. Suoqin Jin, Christian F Guerrero-Juarez, Lihua Zhang, Ivan Chang, Raul Ramos, Chen-Hsiang Kuan, Peggy Myung, Maksim V Plikus, and Qing Nie. Inference and analysis of cell-cell communication using cellchat. *Nature communications*, 12(1):1–20, 2021.
18. Michael Pollak. Insulin and insulin-like growth factor signalling in neoplasia. *Nature Reviews Cancer*, 8(12):915–928, 2008.
19. Ayse Ceren Mutgan, H Erdinc Besikcioglu, Shenghan Wang, Helmut Friess, Güralp O Ceyhan, and Ihsan Ekin Demir. Insulin/igf-driven cancer cell-stroma crosstalk as a novel therapeutic target in pancreatic cancer. *Molecular cancer*, 17(1):1–11, 2018.
20. Kathryn T Biegging and Laura D Attardi. Deconstructing p53 transcriptional networks in tumor suppression. *Trends in cell biology*, 22(2):97–106, 2012.
21. Arnold J Levine and Moshe Oren. The first 30 years of p53: growing ever more complex. *Nature reviews cancer*, 9(10):749–758, 2009.
22. Malte D Luecken, Maren Büttner, Kridsakorn Chaichoompu, Anna Danese, Marta Interlandi, Michaela F Müller, Daniel C Strobl, Luke Zappia, Martin Dugas, Maria Colomé-Tatché, et al. Benchmarking atlas-level data integration in single-cell genomics. *Nature methods*, 19(1):41–50, 2022.
23. Zixiang Luo, Chenyu Xu, Zhen Zhang, and Wenfei Jin. scgae: topology-preserving dimensionality reduction for single-cell rna-seq data using graph autoencoder. *bioRxiv*, 2021.
24. Juexin Wang, Anjun Ma, Yuzhou Chang, Jianting Gong, Yuexu Jiang, Ren Qi, Cankun Wang, Hongjun Fu, Qin Ma, and Dong Xu. scgnn is a novel graph neural network framework for single-cell rna-seq analyses. *Nature communications*, 12(1):1–11, 2021.
25. Chun Wang, Shirui Pan, Ruiqi Hu, Guodong Long, Jing Jiang, and Chengqi Zhang. Attributed graph clustering: A deep attentional embedding approach. *arXiv preprint arXiv:1906.06532*, 2019.
26. Cédric Arisdakessian, Olivier Poirion, Breck Yunits, Xun Zhu, and Lana X Garmire. Deepimpute: an accurate, fast, and scalable deep neural network method to impute single-cell rna-seq data. *Genome biology*, 20(1):1–14, 2019.

REVIEWER COMMENTS

Reviewer #1 (Remarks to the Author):

In response to my comments, the authors performed substantial additional analyses. I appreciate their efforts and the paper now appears technically sound.

However, it is still not clear to me that scMGCA would offer substantial benefit to colleagues in the field to warrant publication in Nature Communications. It performs batch correction and dimensionality reduction, similar to other methods. The authors demonstrate improved clustering and detection of small clusters in some cases, but their method is so much slower than current widely used standards in the field (e.g. SCANPY and Seurat), as well as some other deep learning based approaches, that this would only be relevant to a small number of researchers. While the authors argue that their approach requires less computational resources, especially memory, from my understanding this is hardly a bottleneck for most researchers given the availability of affordable cloud computing.

Moreover, while the re-analysis of the PDAC dataset gave results more in line with what one might expect based on current knowledge and literature, it is at least surprising that they differ so substantially from the first analysis. For example, the proposed link to SLE no longer appears, which was not commented upon in the revised version or the rebuttal. Therefore, the stability of predictions from scMGCA is not clear to me, begging the question whether it really is an improved approach for functional genomics, as the authors claim.

Reviewer #2 (Remarks to the Author):

Thanks very much for all the hard work addressing my comments! I am still not convinced about the novelty of the work, but I will not be a deal breaker here.

I have two small comments:

1- It is bizarre how your method outperforms all other methods. have you also optimized other methods or yours per dataset?

2- I still think adding a use case or figure with failure would improve the paper.

Reviewer #3 (Remarks to the Author):

Thanks for authors' diligent work on this manuscript, I agree with the authors' major revisions. Here I just provide some suggestions with my way of thinking.

(1) In major comment 2, the authors mentioned

"Indeed, the KNN graph, as the reviewer points out, is straightforward in its construction. However, according to some previous studies (23, 24), the KNN graph, which directly assigns weights only by the distance between two nodes, is more suitable for completely random networks. For the single-cell network composed of cells with homogeneity and consistency, the KNN graph may not be compatible with the single-cell network properties from different platforms. The reason could be that the KNN graph only focuses on the relationship between two cells and does not perceive the underlying information of surrounding cells (e.g. cell clusters), which often also plays an important role in the single-cell network."

I want to highlight that weighted KNN graph has been widely used in cell clustering¹ (Seurat's default

cluster method 'snn') and dimension reduction (see UMAP part)^{2,3} for some years. In addition, I want to make sure which KNN your supplementary Fig. 26 is based on, unweighted or weighted KNN? If the conclusion is from unweighted KNN, I think it may not be appropriate.

Moreover, in scGNN (your citation 24) method part 'Cell graph and pruning', scGNN generates weighted edges between node and its k nearest neighbors and prunes outliers by isolation forest. Actually, I agree that "information of surrounding cells" is important in single-cell analysis, but how to define "surrounding cells" is a question for different tools, obviously scGNN does not think long-distance (dissimilar) cells need to be considered. Further, for my understanding, the PPMI is deduced from steps of transitions which are constructed by unweighted KNN that is generated by cell-cell transcriptomic similarities. I can understand the PPMI has different approach to estimate cell-cell relationship, but it is difficult for me to understand the essential difference between the PPMI and weighted KNN, both depending on KNN graph built by cell-cell transcriptomic similarity.

(2) In my major comment 3, I suggested that

"I noticed that the authors showed cell trajectory in Fig. 5h, but the cells in the panels do not have logical biological connections (i.e., the cells belong to different cell types, it is difficult to estimate whether trajectory connections in Fig.5h are appropriate)."

I think the authors do not modify this, and I do not know why the authors ignore this comment. Building cell trajectory with completely different cell types may not be a logical strategy (e.g. between B cell and Mesenchymal cell), and I am worried that how many people will accept the authors' conclusion,

"By measuring the progression of biological processes based on transcriptional similarity, we found that the developmental trajectory mainly starts with mainly mesenchymal cells, then moves to B cells, endothelial cells, and T cells" (in manuscript, page 11).

I strongly recommend the authors improve this part, at least using some reasonable data to explain the ability of scMGCA in cell trajectory.

About the authors' explanation of the incorrect boundaries, Fig. R1 and R2 only display extreme UMAP distance, $\text{min_dist} = 0.1$ or 1 ; it is difficult to estimate the performance. SCANPY v1.9.1 (default $\text{min_dist} = 0.5$) and Seurat v4.2.0 (default $\text{min_dist} = 0.3$).

Reference:

1. Xu, C. & Su, Z. Identification of cell types from single-cell transcriptomes using a novel clustering method. *Bioinformatics* 31, 1974–1980 (2015).
2. Xiang, R. et al. A comparison for dimensionality reduction methods of single-cell RNA-seq data. *Front. Genet.* 12, 646936 (2021).
3. Becht, E. et al. Dimensionality reduction for visualizing single-cell data using UMAP. *Nat. Biotechnol.* 37, 38–44 (2018).

Reviewer #1:

In response to my comments, the authors performed substantial additional analyses. I appreciate their efforts and the paper now appears technically sound.

Response: Thank you very much for the positive and constructive comments. We have duly followed your comments and suggestions to address all the concerns as elaborated in the following responses. In particular, we would like to thank you for your critical comments which have improved the manuscript in numerous ways. Thank you very much for your precious time and kind attention.

1. *However, it is still not clear to me that scMGCA would offer substantial benefit to colleagues in the field to warrant publication in Nature Communications. It performs batch correction and dimensionality reduction, similar to other methods. The authors demonstrate improved clustering and detection of small clusters in some cases, but their method is so much slower than current widely used standards in the field (e.g. SCANPY and Seurat), as well as some other deep learning based approaches, that this would only be relevant to a small number of researchers. While the authors argue that their approach requires less computational resources, especially memory, from my understanding this is hardly a bottleneck for most researchers given the availability of affordable cloud computing.*

Response: Thank you very much for pointing this out. We agree that there are non-deep learning and deep learning methods that have been developed for clustering single-cell RNA-seq data. This highlights not only the importance of the problem, but also the unresolved challenges. In addition, each of the existing methods has its own advantages and disadvantages. Our main contributions lie in the topological identification and interpretation ability across multiple platforms. In addition, through the substantial revisions with 3 independent reviewers in more than half year, we are also confident that the resultant experimental results in this study can also serve as an independent and comprehensive benchmark beyond the methodology contributions.

As you have pointed out, earlier non-deep-learning methods such as SCANPY (1) and Seurat (2) have been developed for a simplified and easy-to-use scRNA-seq data analysis; it relies on linear dimensional reduction such as principal component analysis on gene expression profiles to capture the relationships among genes and cells. However, the rationality of the linear assumption of gene expression data remained hypothetical and speculative (3). In addition, our evaluation shows that SCANPY and Seurat perform poorly with increasing cell size, both in terms of cell clustering and the removal of batch effects; it prompted the proposal of efficient deep learning algorithms. Deep learning and deep graph learning algorithms offer an alternative way to model nonlinear features of gene expression profiles, especially for single-cell data with many cells and samples; for instance, DESC enables accurate clustering by removing batch effects in single-cell RNA-seq analysis; scGNN formulates and aggregates cell-cell relationships via a graph neural network for scRNA-seq analysis. The novelty in these works lies in their applications and biological functional analysis, as supported by the comprehensive experiments. Moreover, thanks to multiple

reviewers' comments, we have also demonstrated that our scMGCA is the first **deep graph learning method** that can simultaneously perform many single-cell data analysis tasks on the basis of their underlying cell-to-cell heterogeneity, including cell clustering, dimensionality reduction, batch effect correction, million-level data analysis, interpretability analysis, and clinical data integration, which is extremely rare in the current field of graph neural networks (to the best of our knowledge). Therefore, from a multitasking perspective, our model can offer substantial benefit to colleagues in the field.

From the technical contribution perspective, the key advantage in scMGCA is "Topological Identification and Interpretation". We have surveyed on lots of modeling techniques before. To incorporate information on cell-to-cell heterogeneity in genomic data, we have narrowed our scope to a deep graph learning model and found that scGNN published in Nature Communications is also a graph neural network based model. To demonstrate the advantages in topology identification and information augmentation clearly, we directly list the differences in the autoencoders between scMGCA and scGNN and the advantages of scMGCA including three important differences, which are summarized below:

1. Firstly, scGNN uses the traditional KNN graph as a cell graph, while scMGCA proposes the cell-PPMI graph as a cell graph to apply the random surfing model on the basis of the KNN graph to capture the cell topology information and generate a probabilistic co-occurrence matrix. After that, we calculate its positive pointwise mutual information (PPMI) matrix to further augment the relationship of similar cells. Indeed, compared to the Euclidean distances between cells in scGNN, the co-occurrence probabilities between cells obtained by random surfing and the PPMI matrix can effectively capture the important relationships and topology between cells, which is also verified through the subsequent theoretical experimental analysis, as elucidated in **Supplementary Note 12** and **Supplementary Fig. 26**.
2. Secondly, single-cell clustering models are often challenged by the high probability of dropout events in single-cell data due to low capture rates and insufficient sequencing depth. To address this limitation, scMGCA proposes a multinomial-based graph convolutional autoencoder to address this problem, whereas scGNN uses an imputation autoencoder with the additional L1 regularizer to generate the reconstructed gene expression values; it is also the difference between our algorithm and scGNN. In particular, the multinomial-based graph convolutional autoencoder adopts a multinomial distribution to capture high-probability dropout events in single cell data and then uses the negative log-likelihood of the multinomial distribution as the objective function to optimize and train the entire network when decoding the latent representation. As a result, the GCN can extract the latent representation that fits the distribution of the single-cell data to improve clustering performance, which is also confirmed in the dropout analysis, as demonstrated in **Supplementary Figs. 27-28**.
3. Then, from the perspective of combining deep learning and clustering algorithms, scGNN is a typical two-stage deep clustering method; in the first stage, graph autoencoders are used to learn the features of the data, and in the second stage, a traditional clustering algorithm such as Louvain or Hierarchical Clustering is chosen to cluster the latent representation yielded by the graph autoencoders. In contrast to the two-stage deep clustering method (scGNN), our scMGCA is a joint deep clustering method specific for scRNA-seq data to train the model and then simultaneously learns cell-cell topology representations and cluster assignments. In particular, we designed three training losses, including the clustering loss, multinomial-based loss, and cell graph reconstruction loss, to jointly optimize deep graph embedding and clustering to elucidate scRNA-seq data across different platforms. Thus, we can see that scMGCA optimizes feature learning

and clustering as a whole, whereas scGNN separates feature learning and clustering. Indeed, according to the previous literature review (4, 5), the clustering performance of joint learning is commonly superior to that of segmented algorithms.

4. Finally, scMGCA interprets the key topological information in the deep graph learning framework, which is not considered by other deep graph learning methods. We focus on the variation of parameter matrices in the GCN encoder because these parameter matrices play an important role in extracting and preserving key topological information of the data. Therefore, we propose a ranking and screening method based on the standard deviation of the parameter matrices in the GCN encoder to find data dimensions with key topological information through high-weight sorting in the parameter matrix, thereby selecting interpretable highly expressed genes. This is not only an interpretable work on the black-box model of the graph neural network, but also an interpretable functional genomics study based on topological knowledge from graph neural network. From the functional analysis of these genes, it is demonstrated that the topological interpretability of scMGCA can effectively enable the downstream analysis in the field and give novel biologically meaningful information not found by other methods (**Fig. 6**).

From the runtime cost perspective, we agree with the reviewers that scMGCA does not provide better runtime than other non-deep learning models. Based on the availability of affordable cloud computing, memory is hardly a bottleneck for most inquirers. In fact, memory is not the only difficulty; non-deep learning models also cannot perform as well as they ought to when data sizes continue to grow. In contrast to the memory requirements of non-deep learning models, deep learning also has high demands on GPUs. However, due to the current limited GPU computing resources, deep learning algorithms generally take longer to execute. This is a challenging problem in deep learning but it does not affect the efficiency since deep learning techniques can learn potential features from high-dimensional sparse data, thus providing considerable performance trade-off. Overall, according to the "no free lunch" theory (6–8), the balance of computation time and performance between different computational methods could still be an open issue.

As we demonstrated in our experiments, this model formulation allows us to accomplish the following tasks more successfully compared to scGNN:

- **Embedding based on topological identification:** scMGCA identifies the topology between cells while characterizing gene expression information, so that cell-cell relationships and complex gene expression patterns are orthogonally combined to obtain robust latent representations. Specially, scMGCA adopts the cell-PPMI graph as the cell graph to characterize cell topological information, which can better characterize the underlying, complex cell-cell relationship and topology in the entire biological network. scMGCA can effectively identify the topological information of cell-PPMI and learn the topology-preserving latent embedding representation, so as to obtain good clustering and dimensionality reduction performance. Furthermore, it is still robust on cross-platform single-cell data (**Figs. 2-3**).
- **Batch effect correction:** scMGCA is a joint deep graph learning method that simultaneously learns cell-cell topological representations and cluster assignments by iteratively optimizing a clustering objective function with a self-trained objective distribution. This iterative procedure moves each cell to its nearest cluster centroid, balances biological and technical differences between clusters, and gradually reduces the influence of batch effects in multiple datasets from different scRNA-seq protocols (**Fig. 4**). In particular, scMGCA also enables soft clustering by assigning cluster-specific

probabilities to each cell, which facilitates cell clustering with high confidence even while correcting for batch effects.

- **Dropout correction:** scMGCA proposes a multinomial-based graph convolutional autoencoder to capture high-probability dropout events in single cell data and then uses the negative log-likelihood of the multinomial distribution as the objective function to optimize and train the entire network when decoding the latent representation. Therefore, scMGCA is sufficiently stable and robust with promising performance in dropout event correction (**Supplementary Figs. 27-28**).
- **Functional genomics based on topological interpretability:** scMGCA has its unique functional genomic interpretability analysis of latent embedding representations. We propose a ranking and screening method based on the standard deviation of the parameter matrices in the GCN encoder to select interpretable highly expressed genes from the latent embedding representation in scMGCA (**Methods part M**). From the functional analysis of these genes, it is demonstrated that the key information analyzed by scMGCA from the central hidden layer of the graph neural network is functionally and pathologically interpretable (**Fig. 6**), which provide a new way for deep graph clustering to explore biological interpretability.
- **Elucidation of cancer mechanisms:** scMGCA can elucidate the underlying regulatory mechanisms of pancreatic ductal carcinoma (PDAC). We applied scMGCA to PDAC data for cell clustering and defined cell types by marker genes in the cell clusters. Interestingly, the type 2 ductal cells identified by scMGCA were derived from tumor cells (**Fig. 7**). Further, we found that type 2 ductal cells were enriched for several pathways essentially associated with PDAC, and most of them were consistent with the pathways enriched in tumor samples from clinical data (**Supplementary Note 18**). It demonstrates the ability of scMGCA to elucidate the underlying regulatory mechanisms of complex diseases and deliver biologically meaningful results.
- **Highly scalable package:** As a multi-task single-cell analysis method, scMGCA integrates cell clustering, dimensionality reduction, batch effect correction, million-level data analysis, interpretability analysis and single-cell data analysis of joint clinical data. To enable users to better use scMGCA for various analysis, we wrap the source code into a python package form, and provide users with some examples of different analyses of scMGCA and step-by-step tutorials to fully explain our algorithm (**Figs. R2-3**). In addition, we have developed a more detailed tutorial for users at <https://philyzh8.github.io> (**Fig. R1**).

Lastly, we would like to re-emphasize the importance and the thoroughness of the above analyses carried out in our current study. For scRNA-seq data analysis, previous algorithms mainly focus on 1-2 tasks, but scMGCA is the first **deep graph learning method** that can simultaneously perform multiple tasks, including cell clustering, dimensionality reduction, batch effect correction, million-level data analysis, interpretability analysis and single-cell data analysis of joint clinical data. To the best of our knowledge, this is extremely rare in the current field of graph neural networks.

2. Moreover, while the re-analysis of the PDAC dataset gave results more in line with what one might expect based on current knowledge and literature, it is at least surprising that they differ so substantially from the first analysis. For example, the proposed link to SLE no longer appears, which was not commented upon in the revised version or the rebuttal. Therefore, the stability of predictions from scMGCA is not clear to me, begging the question whether it really is an improved approach for functional genomics, as the authors claim.

Response: Thank you very much for your insightful comments. We apologize for the confusions on the analysis of SLE in the revised manuscript and for not providing any explanation. Since we expanded the PDAC dataset in the revised manuscript to include 57,530 cells from primary PDAC tumors and control pancreas, corresponding to 6.5 times the 8,921 cells in the first analysis, which can avoid the amplification of some unrepresentative local information. In addition, we used manual cell annotation instead of SingleR (9), an unbiased cell type annotation method, to annotate different cell types. As a result of the variation in these two experimental settings, the cell types obtained by the different annotation methods in the two versions are not exactly the same, which leads to inconsistencies in the differential genes, also resulting in variations in the results of the subsequent GSEA analysis.

Nonetheless, the pathway for systemic lupus erythematosus (SLE) did not disappear in the second analysis, but its enrichment score was decreased (**Supplementary Fig. 34**). Moreover, the p -values of the pathway for SLE were also increased to be greater than 0.05. Therefore, we did not analyze it as the main pathway. To address your concerns, we have extracted the genes enriched in the SLE pathway and employed STRING (10) to construct the PPI network for the selected genes (**Supplementary Fig. 35a**). Then, we used molecular complex detection (MCODE) (11) in Cytoscape (12) to identify three important modules in the PPI network (**Supplementary Fig. 35b**). Further, we selected the top three genes in the PPI network through CytoHubba (13), which were also in the module with the highest score; thus, we use these three genes as the hub genes of SLE (**Supplementary Fig. 35c**). Finally, we studied the differential expression of these three hub genes between tumors and adjacent normal tissues in the TCGA pan-cancer dataset (**Supplementary Fig. 36**). It can be observed from **Supplementary Fig. 36** that the gene expression levels of these three genes on the PAAD data are relatively significant, and the p -values calculated by the Wilcoxon test are all less than 0.01. Therefore, we believe that SLE and PDAC are potentially related.

To verify whether the SLE signaling pathways identified by the single-cell PDAC data are indeed clinically relevant, we also independently adopted the TCGA's database public of invasive pancreatic ductal adenocarcinomas and its variants (PAAD) to validate the insight consistency. Specifically, we performed GSVA enrichment of SLE-related genes on tumor cells from PDAC and 178 tumor samples from TCGA data, respectively, and then compared them (**Supplementary Fig. 37**). It can be found that the SLE pathway can be enriched in both PDAC and its TCGA data, and both are expressed in tumor cells. In particular, some important signaling pathways in pathogenic process of SLE were also enriched, including coagulation cascade and complement system (14–16). These results demonstrate that an association of SLE with PDAC can be found in large cohort studies. In addition, several studies have shown that SLE has been studied as a risk factor for pancreatic cancer (17, 18), and that SLE may be associated with an increased risk of pancreatic cancer (19, 20). In summary, we have revealed a potential relationship between SLE and PDAC, which has also been demonstrated in clinical data.

Supplementary Fig. 34. GSEA on KEGG pathways for PDAC.

Supplementary Fig. 35. Protein-protein interaction (PPI) network complex and modular analysis of genes enriched in the SLE pathway. **a** PPI network of genes enriched in the SLE pathway. **b** PPI networks in MCODE analysis. **c** Top three hub genes in CytoHubba analysis.

*: p -value < 0.05 **: p -value < 0.01 ***: p -value < 0.001

Supplementary Fig. 36. The differential expression of three hub genes between tumors and adjacent normal tissues in the TCGA pan-cancer dataset.

Supplementary Fig. 37. GSEA enrichment pathway diagram of the result of single-cell RNA-seq data (top) and TCGA dataset (down).

Reviewer #2:

Thanks very much for all the hard work addressing my comments! I am still not convinced about the novelty of the work, but I will not be a deal breaker here. I have two small comments.

Response: Thank you very much for the positive and considerate comments. We have duly followed your comments and suggestions to address the remaining comments as elaborated in the following responses. In particular, we would like to thank you for your critical comments which have improved the manuscript in numerous ways. Thank you very much for your precious time and kind attention.

1. *It is bizarre how your method outperforms all other methods. have you also optimized other methods or yours per dataset?*

Response: Thank you very much for your comments. Indeed, the parameters of our model and all the compared methods were optimized (after preliminary experiments). To address your concerns further, we constructed an additional parameter analysis to follow our model parameter analysis procedure for the other algorithms as well. We have included those results and analyses in **Supplementary Tables 1-18** of the revised manuscript. It is copied below for your easy reference:

"To ensure robustnesses in the experiments, we compare the best clustering results of other deep learning algorithms by exploring potential parameter settings in a brute-force manner on 20 real datasets in terms of NMI, ARI and ASW. We set the parameters of other algorithms according to the same adjustment method of scMGCA, mainly including the number of highly variable genes and the number of network layers. Specifically, the number of highly variable genes is selected from [300, 500, 1000, 1500, 2000], and the number of network layers is adjusted according to the default framework of different algorithms. The experimental results are tabulated in **Supplementary Tables 1-18**. We chose the parameter settings that obtained the best average clustering performance for each method in comparison with scMGCA, and then updated **Fig. 2a** and **Supplementary Figs. 1-7**."

Supplementary Table 1. Comparison performance of scziDesk with different numbers of highly variable genes measured by NMI, ARI and ASW

Dataset	NMI					ARI					ASW				
	300	500	1000	1500	2000	300	500	1000	1500	2000	300	500	1000	1500	2000
Pollen	0.8982	0.8810	0.9365	0.9314	0.8846	0.8776	0.8476	0.9630	0.8486	0.8403	0.5317	0.4982	0.5869	0.5045	0.4865
Camp-Brain	0.5153	0.5232	0.4978	0.4925	0.4964	0.3960	0.4151	0.3949	0.3898	0.3678	0.1926	0.2123	0.1633	0.1423	0.1678
Camp-Liver	0.7332	0.7343	0.7317	0.7802	0.7293	0.5443	0.6005	0.5436	0.6250	0.5382	0.4327	0.4954	0.4868	0.4849	0.4107
QS Diaphragm	0.8951	0.9210	0.9362	0.9239	0.8985	0.9113	0.9517	0.9741	0.9714	0.9185	0.7169	0.6466	0.8657	0.8349	0.8837
QS Limb Muscle	0.8942	0.9468	0.9563	0.9562	0.9535	0.9232	0.9743	0.9745	0.9709	0.9771	0.6770	0.6567	0.8524	0.8471	0.8504
QS Trachea	0.6439	0.7341	0.7536	0.7066	0.7510	0.7662	0.8085	0.8340	0.7650	0.8291	0.4428	0.5108	0.5131	0.6253	0.6954
QS Lung	0.8063	0.7543	0.7580	0.7530	0.7664	0.7784	0.7401	0.7397	0.5778	0.5868	0.4642	0.5295	0.4476	0.6097	0.6172
Muraro	0.5233	0.7349	0.7352	0.8880	0.8131	0.3544	0.6784	0.6846	0.9301	0.7233	0.0793	0.3478	0.6776	0.3844	0.6940
Qx Bladder	0.9101	0.9707	0.9648	0.9653	0.9910	0.9411	0.9858	0.9837	0.9843	0.9968	0.7318	0.7206	0.7734	0.7477	0.7171
Klein	0.7430	0.7777	0.7837	0.8448	0.8142	0.7902	0.7984	0.8068	0.8749	0.8128	0.5297	0.5775	0.6404	0.7127	0.7100
Romanov	0.7201	0.7274	0.6660	0.6715	0.6903	0.7566	0.7603	0.6299	0.6711	0.7080	0.3233	0.4166	0.4027	0.4167	0.4114
Adam	0.8502	0.8509	0.8313	0.8429	0.7418	0.8413	0.8680	0.8214	0.8400	0.6301	0.5776	0.6074	0.5876	0.5843	0.6049
Qx Limb Muscle	0.8338	0.9131	0.9269	0.8587	0.9263	0.8746	0.9441	0.9376	0.7915	0.9313	0.7295	0.8028	0.7841	0.825	0.8428
QS Heart	0.7357	0.8723	0.8941	0.8613	0.9214	0.6035	0.9324	0.9557	0.9042	0.9585	0.5059	0.6121	0.7882	0.7554	0.7208
Young	0.7765	0.7394	0.7343	0.7976	0.7366	0.6924	0.6836	0.6940	0.7361	0.6884	0.3701	0.3519	0.4139	0.4880	0.4432
Plasschaert	0.5748	0.7899	0.7867	0.6561	0.6747	0.4111	0.8634	0.8591	0.5058	0.5858	0.3628	0.3657	0.4129	0.3888	0.4117
Qx Spleen	0.8046	0.8289	0.8212	0.8014	0.8646	0.8956	0.9197	0.9291	0.8906	0.9294	0.6909	0.5368	0.5573	0.6519	0.6481
Chen	0.6511	0.7101	0.7319	0.7117	0.7538	0.5738	0.6677	0.7337	0.3132	0.7498	0.3168	0.5338	0.5659	0.5853	0.5623
Tosches turtle	0.5879	0.6082	0.5731	0.5789	0.5379	0.6017	0.6165	0.5901	0.6019	0.5517	0.4526	0.4930	0.4237	0.4538	0.4103
Bach	0.7953	0.8343	0.8134	0.8269	0.8645	0.7406	0.8738	0.8293	0.8385	0.8646	0.5406	0.5308	0.6231	0.5996	0.5763
Average	0.7446	0.7926	0.7916	0.7924	0.7905	0.7137	0.7965	0.7939	0.7515	0.7594	0.4834	0.5223	0.5783	0.5821	0.5932

Supplementary Table 17. Comparison performance of GraphSCC with different numbers of highly variable genes measured by NMI, ARI and ASW

Dataset	NMI					ARI					ASW				
	300	500	1000	1500	2000	300	500	1000	1500	2000	300	500	1000	1500	2000
Pollen	0.9195	0.9148	0.9174	0.8818	0.9233	0.8826	0.9240	0.9040	0.8458	0.9167	0.6954	0.6813	0.6481	0.6906	0.7625
Camp-Brain	0.4658	0.4347	0.4706	0.4728	0.4683	0.2817	0.2498	0.3111	0.3215	0.3119	0.2422	0.1383	0.1633	0.1055	0.1016
Camp-Liver	0.7637	0.6688	0.6997	0.6144	0.6354	0.5661	0.4315	0.4517	0.4011	0.4033	0.4737	0.4645	0.4504	0.2867	0.3308
QS Diaphragm	0.6664	0.8966	0.6453	0.8888	0.8752	0.4671	0.9519	0.4663	0.9474	0.9421	0.5572	0.6932	0.6588	0.7977	0.7363
QS Limb Muscle	0.7025	0.7009	0.9214	0.9050	0.9080	0.5329	0.5303	0.9571	0.9314	0.9412	0.4684	0.7577	0.7568	0.7189	0.6572
QS Trachea	0.6080	0.7102	0.8010	0.8104	0.8123	0.6784	0.8020	0.8127	0.8476	0.8468	0.4368	0.5169	0.5069	0.5737	0.6606
QS Lung	0.6586	0.6824	0.7334	0.6629	0.7569	0.5189	0.5180	0.7084	0.6050	0.7348	0.2598	0.4450	0.4630	0.4692	0.4942
Muraro	0.5830	0.5723	0.7632	0.7731	0.7885	0.3789	0.3391	0.7454	0.7539	0.7741	0.1549	0.4759	0.5889	0.6053	0.5351
Qx Bladder	0.3825	0.4541	0.3572	0.9508	0.9264	0.1603	0.1638	0.1612	0.9815	0.9605	0.6448	0.5697	0.4663	0.5159	0.5270
Klein	0.6401	0.7484	0.7391	0.7243	0.7202	0.6638	0.7789	0.7518	0.7227	0.7090	0.4645	0.5936	0.6526	0.6814	0.6918
Romanov	0.3808	0.6080	0.6241	0.6138	0.5752	0.2296	0.4501	0.4564	0.4436	0.4297	0.1997	0.2498	0.3196	0.3244	0.3805
Adam	0.5533	0.5606	0.5698	0.5776	0.5779	0.2381	0.2444	0.2482	0.2602	0.2633	0.4619	0.5022	0.5033	0.4603	0.4754
Qx Limb Muscle	0.4378	0.6501	0.8182	0.7920	0.7763	0.2092	0.4534	0.8066	0.7405	0.6421	0.5165	0.7410	0.7731	0.7400	0.7073
QS Heart	0.6614	0.8350	0.8449	0.8421	0.8474	0.6643	0.8908	0.8963	0.8953	0.9008	0.3843	0.5709	0.5621	0.4993	0.5206
Young	0.5633	0.6047	0.6290	0.6533	0.6460	0.3712	0.3795	0.4153	0.4470	0.4183	0.2345	0.2033	0.2444	0.3828	0.3246
Plasschaert	0.4508	0.7489	0.7994	0.8035	0.7832	0.2619	0.7965	0.8681	0.8648	0.8396	0.1621	0.2541	0.3073	0.3358	0.3762
Qx Spleen	0.4496	0.6581	0.5050	0.4420	0.4243	0.3974	0.7874	0.5633	0.4387	0.4138	0.3627	0.4504	0.4623	0.3596	0.4346
Chen	0.5144	0.6497	0.7046	0.6979	0.7039	0.2531	0.4543	0.5743	0.5727	0.5696	0.1774	0.3166	0.2956	0.2988	0.3266
Tosches turtle	0.6442	0.6937	0.7027	0.6940	0.7102	0.4986	0.5555	0.6176	0.5321	0.6318	0.1947	0.3035	0.2832	0.2784	0.3599
Bach	0.4966	0.6950	0.5369	0.5446	0.5370	0.3897	0.6546	0.4012	0.4063	0.3988	0.2092	0.3468	0.3323	0.3897	0.3780
Average	0.5771	0.6744	0.6891	0.7173	0.7198	0.4322	0.5678	0.6059	0.6480	0.6524	0.3650	0.4637	0.4719	0.4757	0.4890

Supplementary Table 18. Comparison performance of GraphSCC with different hidden layer numbers measured by NMI, ARI and ASW. GraphSCC1 is with ([Input layer]-64-10), GraphSCC2 is with ([Input layer]-256-64-10), and GraphSCC3 is with ([Input layer]-512-256-64-10)

Dataset	NMI			ARI			ASW		
	GraphSCC1	GraphSCC2	GraphSCC3	GraphSCC1	GraphSCC2	GraphSCC3	GraphSCC1	GraphSCC2	GraphSCC3
Pollen	0.9107	0.8032	0.9233	0.9274	0.8962	0.9167	0.7512	0.7594	0.7625
Camp-Brain	0.2866	0.3964	0.4683	0.2236	0.2636	0.3119	0.119	0.0930	0.1016
Camp-Liver	0.5608	0.5791	0.6354	0.3618	0.3884	0.4033	0.3086	0.2701	0.3308
QS Diaphragm	0.8637	0.8764	0.8752	0.9346	0.9478	0.9421	0.7471	0.7447	0.7363
QS Limb Muscle	0.8514	0.8635	0.9080	0.9128	0.9136	0.9412	0.6182	0.6189	0.6572
QS Trachea	0.8381	0.8134	0.8123	0.8586	0.6782	0.8468	0.5514	0.5869	0.6606
QS Lung	0.7354	0.7117	0.7569	0.7207	0.7053	0.7348	0.5020	0.4925	0.4942
Muraro	0.6932	0.6969	0.7885	0.6488	0.6531	0.7741	0.5318	0.5337	0.5351
Qx Bladder	0.9129	0.9306	0.9264	0.9600	0.9677	0.9605	0.5144	0.5446	0.5270
Klein	0.7112	0.7804	0.7202	0.7206	0.7683	0.7090	0.6585	0.6941	0.6918
Romanov	0.4460	0.4430	0.5752	0.4346	0.4109	0.4297	0.3887	0.4050	0.3805
Adam	0.4292	0.4525	0.5779	0.3508	0.3665	0.2633	0.4435	0.4546	0.4754
Qx Limb Muscle	0.7398	0.7982	0.7763	0.6169	0.6819	0.6421	0.6877	0.7291	0.7073
QS Heart	0.9038	0.9102	0.8474	0.8406	0.8628	0.9008	0.6011	0.5964	0.5206
Young	0.6025	0.5332	0.6460	0.4084	0.3223	0.4183	0.4114	0.2594	0.3246
Plasschaert	0.8148	0.7655	0.7832	0.8541	0.8295	0.8396	0.3660	0.3472	0.3762
Qx Spleen	0.5497	0.5848	0.4243	0.5120	0.5615	0.4138	0.4012	0.4788	0.4346
Chen	0.7829	0.7002	0.7039	0.6014	0.5237	0.5696	0.2653	0.3739	0.3266
Tosches turtle	0.6513	0.3906	0.7102	0.6223	0.5762	0.6318	0.2483	0.3395	0.3599
Bach	0.7637	0.6650	0.5370	0.7904	0.7191	0.3988	0.4673	0.4501	0.3780
Average	0.7024	0.6847	0.7198	0.6650	0.6518	0.6524	0.4791	0.4886	0.4890

2. I still think adding a use case or figure with failure would improve the paper.

Response: Thank you very much for the constructive comments, which greatly enhances the quality of the scMGCA algorithm. We agree with you and have followed your comments to provide users with the scMGCA website, different analysis examples, and step-by-step tutorials to fully explain our algorithm. It is copied below for your easy reference:

1) Developing a more detailed tutorial site on scMGCA:

We have developed a more detailed tutorial for users at <https://philyzh8.github.io> (Fig. R1). There are three sections on the website: the homepage, installation, and tutorial. Home: This part is about the overview and framework of scMGCA, allowing users to understand the basic functions and principles of scMGCA; Installation: This part describes the installation of scMGCA's operating environment, and users can use this installation instructions to successfully run scMGCA in Python; Tutorial: This part provides a comprehensive tutorial on scMGCA for cluster analysis and batch effect correction, allowing users to reproduce the results in the paper.

Fig. R1. The page of scMGCA's tutorial website.

2) Giving some examples of different analyses of scMGCA:

We have provided some examples of different analyses of scMGCA and created a tutorial folder (**Fig. R2**) for storage on GitHub. Among them, ‘demo.py’ is the example for single-cell data clustering; ‘demo_batch.py’ is the example for correcting the batch effect of data from different scRNA-seq protocols; ‘demo_scale.py’ is the example for analyzing mouse brain data containing 1.3 million cells; ‘demo_para.py’ is an automatic hyperparameter search algorithm; and ‘tutorial.md’ provides suggestions for the user-friendly uses of scMGCA. Users can adopt these examples and suggestions to perform various analyses using scMGCA.

File Name	Last Update	Time Ago
demo.ipynb	Update demo.ipynb	12 hours ago
demo.py	Update demo.py	12 hours ago
demo_batch.ipynb	Update demo_batch.ipynb	12 hours ago
demo_batch.py	Update demo_batch.py	12 hours ago
demo_para.py	Update demo_para.py	12 hours ago
demo_scale.py	Update demo_scale.py	12 hours ago
tutorial.md	Update tutorial.md	12 hours ago

Fig. R2. A summary of some examples on scMGCA analysis.

3) Providing step-by-step tutorials:

In addition, we have provided step-by-step guidances for users in the updated tutorials (‘demo.ipynb’ and ‘demo_batch.ipynb’) to ensure that scMGCA works without obstacles on analyzing the scRNA-seq data (**Fig. R3** and **Fig. R4**). These two tutorials introduce the specific codes of scMGCA in the process of cluster analysis and batch effect correction. Specifically, it involves importing packages, parameter settings, data processing, training and so on. Meanwhile, informative notes are provided so that the user can easily understand the configurations and rationale behind each step in the analysis process.

Cluster analysis

In this tutorial, we will perform the entire scMGCA cluster analysis using the Qx Bladder dataset (can be downloaded here).

Import python package

```
In [4]: import os
import argparse
import pandas as pd
import tensorflow as tf
from spektral.layers import GraphConv
from sklearn import metrics
from numpy.random import seed
seed(1)
tf.random.set_seed(1)
# Remove warnings
import warnings
warnings.simplefilter(action='ignore', category=FutureWarning)
tf.compat.v1.logging.set_verbosity(tf.compat.v1.logging.ERROR)
# scMGCA module
from scMGCA.preprocess import *
from scMGCA.utils import *
from scMGCA.scmgca import SCMGCA, SCMGCAL
from scMGCA.losses import *
from scMGCA.graph_function import *
```

Parameter settings

```
In [13]: parser = argparse.ArgumentParser(description="train", formatter_class=argparse.ArgumentDefaultHelpFormatter)
parser.add_argument("--dataname", default="Quake_10x_Bladder", type=str)
parser.add_argument("--highly_genes", default=500, type=int)
parser.add_argument("--pretrain_epochs", default=1000, type=int)
parser.add_argument("--maxiter", default=300, type=int)
args = parser.parse_args(args=["--dataname", "Quake_10x_Bladder", "--highly_genes", "500", "--pretrain_epochs", "1000", "--maxiter", "300"])
```

Read data

```
In [14]: data = './dataset/' + args.dataname + '/data.h5'
x, y = preprocess(data)
cluster_number = len(np.unique(y))
print("Cell number:", x.shape[0])
print("Gene number", x.shape[1])
print("Cluster number:", cluster_number)

Cell number: 2500
Gene number 23341
Cluster number: 4
```

Data preprocessing

```
In [15]: x = np.cell(x).astype(np.int)
adata = sc.AnnData(x)
adata.obs['Group'] = y
adata = normalize(adata, copy=True, highly_genes=args.highly_genes, size_factors=True, normalize_input=True, logtrans_input=True)
```

Construct cell graph

```
In [ ]: count = adata.X
adj = get_adj(count)
adj_n = GraphConv.preprocess(adj)
```

Pre-training

```
In [17]: model = SCMGCA(count, adj=adj, adj_n=adj_n)
model.pre_train(epochs=args.pretrain_epochs)
latent_pre = model.embedding(count, adj_n)

Epoch 990 Mult_loss: -2.6953423 A_rec_loss: 0.009539412
Epoch 1000 Mult_loss: -2.6975663 A_rec_loss: 0.009536003
Pre_train Finish!
```

Training

```
In [18]: centers = init_center(args.latent_pre, adj_n, cluster_number)
Cluster_predicted=model.train(epochs=args.maxiter, centers=centers)

Epoch 270 Mult_loss: -2.7136111 A_rec_loss: 0.009289946 cluster_loss: 0.041201342
Epoch 280 Mult_loss: -2.7191837 A_rec_loss: 0.009289946 cluster_loss: 0.040637758
Epoch 290 Mult_loss: -2.7230098 A_rec_loss: 0.009289946 cluster_loss: 0.040089436
```

Evaluation

```
In [19]: y = list(map(int, y))
Cluster_predicted_y_pred = np.array(Cluster_predicted.y_pred)
nmi = metrics.normalized_mutual_info_score(y, Cluster_predicted.y_pred)
ari = metrics.adjusted_rand_score(y, Cluster_predicted.y_pred)
print('NMI= %.4f, ARI= %.4f' % (nmi, ari))

NMI= 0.9802, ARI= 0.9900
```

Fig. R3. The step-by-step guidance for users on cluster analysis ('demo.ipynb').

Batch effect correction

In this tutorial, we will perform the entire scMGCA batch effect correction using the human pancreatic dataset (can be downloaded here).

Import python package

```
In [1]: import os
import argparse
import pandas as pd
import tensorflow as tf
from spektral.layers import GraphConv
from sklearn import metrics
from numpy.random import seed
seed(1)
tf.random.set_seed(1)
# Remove warnings
import warnings
warnings.simplefilter(action='ignore', category=FutureWarning)
tf.compat.v1.logging.set_verbosity(tf.compat.v1.logging.ERROR)
# scMGCA module
from scMGCA.preprocess import *
from scMGCA.utils import *
from scMGCA.scgca import SCMGCA, SCMGCAI
from scMGCA.losses import *
from scMGCA.graph_function import *
```

Parameter settings

```
In [2]: parser = argparse.ArgumentParser(description='train', formatter_class=argparse.ArgumentDefaultsHelpFormatter)
parser.add_argument("--dataset", default = 'pancreas', type = str)
parser.add_argument("--highly_genes", default = 2000, type=int)
parser.add_argument("--pretrain_epochs", default = 1000, type=int)
parser.add_argument("--maxiter", default = 300, type=int)
args = parser.parse_args(args=[ "--dataset", 'pancreas', "--highly_genes", '2000', "--pretrain_epochs", '1000', "--maxiter", '300'])
```

Read data

```
In [3]:adata = read_pancreas('./dataset/pancreas', cache=True)
y = np.array(adata.obs['celltype'].values, dtype=str)
cluster_number = len(np.unique(y))
print('Cell number:', adata.X.shape[0])
print('Gene number:', adata.X.shape[1])
print('Cluster number:', cluster_number)

View of AnnData object with n_obs × n_vars = 6321 × 21215
obs: 'celltype', 'tech', 'n_genes', 'percent_auto', 'n_counts'
var: 'genesetname', 'n_cells'
Cell number: 6321
Gene number: 21215
Cluster number: 13
```

Data preprocessing

```
In [4]: count = normalize_batch(adata, batch_key = 'tech', n_high_var = args.highly_genes)
```

Construct cell graph

```
In [5]: adj = get_adj(count)
adj_n = GraphConv.preprocess(adj)
```

Pre-training

```
In [6]: model = SCMGCA(count, adj=adj, adj_n=adj_n, latent_dim=20, dec_dim=[128])
model.pre_train(epochs=args.pretrain_epochs, V_a=0.6, lr=5e-4)
latent_pre = model.embedding(count, adj_n)
```

```
Epoch 1000 Mult_loss: -3.3980877 A_rec_loss: 0.0051390654
Pre_train Finish!
```

Training

```
In [7]: adata_latent = sc.AnnData(latent_pre)
sc.pp.neighbors(adata_latent, n_neighbors = 15, use_rep='X')
resolution = find_resolution(adata_latent, 6, 0)
adata_latent = sc.tl.ligand(adata_latent, resolution = resolution, random_state = 0, copy = True)
Y_pred_init = np.asarray(adata_latent.obs['leiden'], dtype=int)
features = pd.DataFrame(adata_latent.X, index = np.arange(0, adata_latent.shape[0]))
Group = pd.Series(Y_pred_init, index = np.arange(0, adata_latent.shape[0]), name='Group')
Mergefeature = pd.concat([features, Group], axis=1)
centers = np.asarray(Mergefeature.groupby('Group').mean())
Cluster_predicted=model.train(epochs=args.maxiter, V_a=0.6, centers=centers)
```

```
Epoch 290 Mult_loss: -3.284503 A_rec_loss: 0.005131225 cluster_loss: 0.06597743
```

Evaluation

```
In [8]: y = list(map(str, y))
Cluster_predicted_y_pred = np.array(Cluster_predicted_y_pred)
nmi = metrics.normalized_mutual_info_score(y, Cluster_predicted_y_pred)
ari = metrics.adjusted_rand_score(y, Cluster_predicted_y_pred)
print('NMI: %4f, ARI: %4f' % (nmi, ari))
```

```
NMI= 0.9276, ARI= 0.9686
```

Visualization

```
In [12]: import matplotlib.pyplot as plt
import umap
latent = Cluster_predicted_latent
reducer = umap.UMAP(n_neighbors=20, metric='cosine', min_dist=0.01, random_state=42)
embedding = reducer.fit_transform(latent)
for i, x in enumerate(adata.obs['tech']):
    if x == 'caltech':
        s1 = plt.scatter(embedding[i][0], embedding[i][1], s=5, color='#019133')
    elif x == 'celseq2':
        s2 = plt.scatter(embedding[i][0], embedding[i][1], s=5, color='#D96400')
    elif x == 'fluidigm':
        s3 = plt.scatter(embedding[i][0], embedding[i][1], s=5, color='#8D9901')
    elif x == 'martsini2':
        s4 = plt.scatter(embedding[i][0], embedding[i][1], s=5, color='#00BAC2')

plt.xticks(())
plt.yticks(())
plt.xlabel('UMAP1', fontsize=30)
plt.ylabel('UMAP2', fontsize=30)

ax = plt.gca()
ax.spines['right'].set_color('black')
ax.spines['top'].set_color('black')
ax.spines['left'].set_color('black')
ax.spines['bottom'].set_color('black')
ax.spines['bottom'].set_linewidth(2)
ax.spines['left'].set_linewidth(2)
ax.spines['right'].set_linewidth(2)
ax.spines['top'].set_linewidth(2)
plt.show()
```

Fig. R4. The step-by-step guidance for users on batch effect correction ('demo_batch.ipynb').

Reviewer #3:

Thanks for authors' diligent work on this manuscript, I agree with the authors' major revisions. Here I just provide some suggestions with my way of thinking.

Response: Thank you very much for the positive and thoughtful comments. We have duly followed your comments and suggestions to address the concerns as elaborated in the following responses. In particular, we would like to thank you for your critical comments which have improved the manuscript in numerous ways. Thank you very much for your precious time and kind attention.

1. *In major comment 2, the authors mentioned “Indeed, the KNN graph, as the reviewer points out, is straightforward in its construction. However, according to some previous studies (23, 24), the KNN graph, which directly assigns weights only by the distance between two nodes, is more suitable for completely random networks. For the single-cell network composed of cells with homogeneity and consistency, the KNN graph may not be compatible with the single-cell network properties from different platforms. The reason could be that the KNN graph only focuses on the relationship between two cells and does not perceive the underlying information of surrounding cells (e.g. cell clusters), which often also plays an important role in the single-cell network.”*

I want to highlight that weighted KNN graph has been widely used in cell clustering1 (Seurat's default cluster method 'snn') and dimension reduction (see UMAP part)2,3 for some years. In addition, I want to make sure which KNN your supplementary Fig. 26 is based on, unweighted or weighted KNN? If the conclusion is from unweighted KNN, I think it may not be appropriate.

Response: Thank you very much for your insights. In Supplemental Figure 26, we employed a weighted KNN graph in which the weights between cells are specified as the min-max normalized Euclidean distance subtracted from 1, and the model employing this weighted graph is denoted as scMGCA(wKNN1). As suggested by the reviewer, we also added the SNN graph constructed in Seurat (21) and the weighted KNN graph constructed in UMAP (22) to the comparison experiments; and the models with those two weighted graphs are denoted as scMGCA(SNN) and scMGCA(wKNN2), respectively. To examine the unique benefits of cell-PPMI graph, we compared scMGCA(SNN), scMGCA(wKNN1) and scMGCA(wKNN2) with scMGCA on 20 datasets as depicted on the revised **Supplementary Fig. 26**.

Supplementary Fig. 26. Comparison of NMI and ARI between scMGCA, scMGCA(SNN), scMGCA(wKNN1) and scMGCA(wKNN2) on 20 datasets.

2. Moreover, in scGNN (your citation 24) method part ‘Cell graph and pruning’, scGNN generates weighted edges between node and its k nearest neighbors and prunes outliers by isolation forest. Actually, I agree that “information of surrounding cells” is important in single-cell analysis, but how to define “surrounding cells” is a question for different tools, obviously scGNN does not think long-distance (dissimilar) cells need to be considered. Further, for my understanding, the PPMI is deduced from steps of transitions which are constructed by unweighted KNN that is generated by cell-cell transcriptomic similarities. I can understand the PPMI has different approach to estimate cell-cell relationship, but it is difficult for me to understand the essential difference between the PPMI and weighted KNN, both depending on KNN graph built by cell-cell transcriptomic similarity.

Response: Thank you very much for your insights. The most essential difference between the KNN graph and the cell-PPMI graph lies in different strategies for information propagation. The KNN graph calculates the distances between each cell and other cells, and selects the top k closest cells as neighbors. **No matter how the weight is computed**, the KNN graph focuses only on the selected neighbor cell information and aggregates this information only in the graph convolutional networks (GCN), which belong to a 1 -hop graph (23). The cell-PPMI graph, on the other hand, uses the random surfing algorithm so that information can be propagated between cells beyond 1 hop. The transfer step s in the cell-PPMI graph sets the range of information propagation. Therefore, the cell-PPMI graph can be regarded as an s -hop graph (To avoid excessive transfers, s is taken as 2 in the experiment). Therefore, when the cell-PPMI graph is involved in network training, the GCN is also able to inherit and extract the information propagated by the cells beyond the 1 -hop range, thus jointly optimizing cell features and capturing the complete topology between cells. In addition, the cell-PPMI graph is also based on the information propagated between cells when measuring the similarity between cells, and adopts the concept of co-occurrence probability. In contrast to the KNN graph where cell similarity is calculated by distance, it can capture potentially complex and nonlinear relationships between different cells.

To clearly compare the difference between the weighted KNN graph and the cell-PPMI graph in terms of information dissemination, we analyze a small graph composed of seven nodes as an example for analysis (**Fig. R5**). Since the KNN graph only considers the distance of 1 -hop neighbors, x_1 can only obtain x_3 , x_4 and x_5 information, and x_2 can only get x_3 , x_6 and x_7 information. Even though both neighborhoods share the node x_3 , only the 1 -hop neighborhood data from the KNN graph is captured and aggregated during GCN training. In contrast, although the initial situation of the cell-PPMI graph is identical to that of the KNN graph, because of the random surfing method, x_1 can receive the information of x_2 through x_3 , and x_1 and x_2 will become each other’s 2 -hop neighbors. Therefore, the information of the 2 -hop neighborhood can be exploited and aggregated under the cell-PPMI graph during GCN training, so that x_1 and x_2 has additional neighborhood information for training, thereby improving the learning of low-dimensional representations.

In addition, we have surveyed lots of definitions of "surrounding cells" before and found that in the single cell community, the vast majority of them are based on KNN to define "surrounding cells", however, they use different similarity distance calculations including Euclidean, cosine, Pearson, and Laplacian kernel distance. Meanwhile, shared nearest neighbor (SNN) is also the most commonly graph construction method in single-cell communities such as Seurat. To further elaborate the performance of different definitions of "surrounding cells" in the context of single cell RNA-seq data, we compared the clustering performance of the cell-PPMI graph with weighted KNN graphs using different distances on 20 scRNA-seq datasets (**Fig.**

R6). Among them, KNN(euc), KNN(cos), KNN(pea) and KNN(lap) represent weighted KNN graphs using Euclidean, cosine, Pearson and Laplacian kernel distance, respectively; scMGCA(snn) represent the scMGCA with SNN. As shown in **Fig. R6**, the clustering performance of the cell-PPMI graph outperforms other graphs on most datasets, it further demonstrates that the cell-PPMI graph is effective.

Fig. R5. An example of information transfer comparison between KNN graph and cell-PPMI graph.

Fig. R6. Comparison of NMI and ARI between scMGCA, scMGCA(snn), KNN(euc), KNN(cos), KNN(pea) and KNN(lap) on 20 datasets.

3. In my major comment 3, I suggested that “I noticed that the authors showed cell trajectory in Fig. 5h, but the cells in the panels do not have logical biological connections (i.e., the cells belong to different cell types, it is difficult to estimate whether trajectory connections in Fig.5h are appropriate).”

I think the authors do not modify this, and I do not know why the authors ignore this comment. Building cell trajectory with completely different cell types may not be a logical strategy (e.g. between B cell and Mesenchymal cell), and I am worried that how many people will accept the authors’ conclusion, “By measuring the progression of biological processes based on transcriptional similarity, we found that the developmental trajectory mainly starts with mainly mesenchymal cells, then moves to B cells, endothelial cells, and T cells” (in manuscript, page 11). I strongly recommend the authors improve this part, at least using some reasonable data to explain the ability of scMGCA in cell trajectory.

Response: Thank you very much for your valuable suggestion. We apologize for our previous response and agree with you that data with reasonable cell trajectory information should be utilized to explain the ability of scMGCA in cell trajectories, rather than data with completely different cell types as in the "Qx Limb Muscle" dataset. Therefore, to follow your recommendation in the last sentence, we have added two time-series datasets (‘Klein’ and hESC dataset) in the **Supplementary Note 16** and removed the Fig. 5h to illustrate the capacity of scMGCA in cell trajectory.

The ‘Klein’ dataset (GSE65525) is composed of mouse embryonic stem (ES) cells with time-series information on the ES cell differentiation process (24). Specifically, a total of 2717 cells were profiled at 0 (cell number $n = 933$), 2 ($n = 303$), 4 ($n = 683$) and 7 ($n = 798$) days after ES cell differentiation. To demonstrate that the interpretable genes selected by scMGCA have the ability to annotate cell trajectories, we compared trajectory inference and pseudo-time analysis of the ‘Klein’ dataset using the original gene expression matrix and the gene expression matrix consisting of 200 genes selected by scMGCA (**Supplementary Fig. 32a**). From **Supplementary Fig. 32a**, it can be observed that the raw data does not lead to the trajectory path well and reveals the wrong stage of cell differentiation, whereas the cell trajectories inferred by the interpretable genes picked by scMGCA were closely related to the true stages of cell differentiation accurately: from Day 0 to Day 2, to Day 4, and finally to Day 7.

In addition, we also analyzed a time-course hESC dataset (GSE75748) derived from the differentiation of H1 ESC into definitive endoderm cells (DEC) (25). A total of 758 cells were profiled at 0 (cell number $n = 92$), 12 ($n = 102$), 24 ($n = 66$), 36 ($n = 172$), 72 ($n = 138$) and 96 ($n = 188$) hours after inducing the differentiation from H1 ESCs to DECs. We also compared the trajectory inference and pseudo-time analysis using the original gene expression matrix and the gene expression matrix consisting of 200 genes selected by scMGCA (**Supplementary Fig. 32b**). As shown in **Supplementary Fig. 32b**, in contrast to the raw data showing incorrect cell differentiation stages, the interpretable genes picked by scMGCA inferred cell trajectories with the annotated cell differentiation stages and accurately followed the true differentiation stages: from Hour 0 to Hour 12, to Hour 24, to Hour 36, to Hour 72, and finally to Hour 96.

In summary, the cell and time trajectories inferred from the interpretable genes selected by scMGCA accurately tracked the stages of cell differentiation, demonstrating that these genes can effectively infer the cell and time trajectories, and further elaborating the effectiveness for selecting interpretable genes.

Supplementary Fig. 32 Trajectory inference and pseudotime analysis using the raw gene expression matrix and the gene expression matrix composed of those 200 interpretable genes selected by scMGCA on **a** 'Klein' dataset and **b** hESC dataset via Monocle.

4. About the authors' explanation of the incorrect boundaries, Fig. R1 and R2 only display extreme UMAP distance, $min_dist = 0.1$ or 1 ; it is difficult to estimate the performance. SCANPY v1.9.1 (default $min_dist = 0.5$) and Seurat v4.2.0 (default $min_dist = 0.3$).

Response: Thank you very much for the insights. We agree with you and have followed your comments to add visual comparisons in UMAP for other cases where min_dist is between 0.1 and 1 with a step size of 0.1 (including the default values as suggested), as well as redraw Fig. R1 (now **Fig. R7**) and Fig. R2 (now **Fig. R8**). As observed from those figures, the originally "lost" boundaries gradually reappear as min_dist increases. It can be demonstrated that scMGCA preserves the boundaries of various cell types, but those boundaries are "lost" due to the parameters of the visualization method adopted on top of scMGCA.

Fig. R7. Visualization of latent embedding representations of scMGCA with different parameters of UMAP.

Qx Limb Muscle

Fig. R8. Visualization of latent embedding representations of scMGCA with different parameters of UMAP on the 'Qx Limb Muscle' dataset.

References

1. F Alexander Wolf, Philipp Angerer, and Fabian J Theis. Scanpy: large-scale single-cell gene expression data analysis. *Genome biology*, 19(1):1–5, 2018.
2. Andrew Butler, Paul Hoffman, Peter Smibert, Efthymia Papalexi, and Rahul Satija. Integrating single-cell transcriptomic data across different conditions, technologies, and species. *Nature biotechnology*, 36(5):411–420, 2018.
3. Romain Lopez, Jeffrey Regier, Michael B Cole, Michael I Jordan, and Nir Yosef. Deep generative modeling for single-cell transcriptomics. *Nature methods*, 15(12):1053–1058, 2018.
4. Chun Wang, Shirui Pan, Ruiqi Hu, Guodong Long, Jing Jiang, and Chengqi Zhang. Attributed graph clustering: A deep attentional embedding approach. *arXiv preprint arXiv:1906.06532*, 2019.
5. Zixiang Luo, Chenyu Xu, Zhen Zhang, and Wenfei Jin. scgae: topology-preserving dimensionality reduction for single-cell rna-seq data using graph autoencoder. *bioRxiv*, 2021.
6. David H Wolpert and William G Macready. No free lunch theorems for optimization. *IEEE transactions on evolutionary computation*, 1(1):67–82, 1997.
7. Stavros P Adam, Stamatis-Aggelos N Alexandropoulos, Panos M Pardalos, and Michael N Vrahatis. No free lunch theorem: A review. *Approximation and optimization*, pages 57–82, 2019.
8. David H Wolpert, William G Macready, et al. No free lunch theorems for search. Technical report, Technical Report SFI-TR-95-02-010, Santa Fe Institute, 1995.
9. Dvir Aran, Agnieszka P Looney, Leqian Liu, Esther Wu, Valerie Fong, Austin Hsu, Suzanna Chak, Ram P Naikawadi, Paul J Wolters, Adam R Abate, et al. Reference-based analysis of lung single-cell sequencing reveals a transitional profibrotic macrophage. *Nature immunology*, 20(2):163–172, 2019.
10. Damian Szklarczyk, Annika L Gable, David Lyon, Alexander Junge, Stefan Wyder, Jaime Huerta-Cepas, Milan Simonovic, Nadezhda T Doncheva, John H Morris, Peer Bork, et al. String v11: protein–protein association networks with increased coverage, supporting functional discovery in genome-wide experimental datasets. *Nucleic acids research*, 47(D1):D607–D613, 2019.
11. Gary D Bader and Christopher WV Hogue. An automated method for finding molecular complexes in large protein interaction networks. *BMC bioinformatics*, 4(1):2, 2003.
12. Michael E Smoot, Keiichiro Ono, Johannes Ruschinski, Peng-Liang Wang, and Trey Ideker. Cytoscape 2.8: new features for data integration and network visualization. *Bioinformatics*, 27(3):431–432, 2011.
13. Chia-Hao Chin, Shu-Hwa Chen, Hsin-Hung Wu, Chin-Wen Ho, Ming-Tat Ko, and Chung-Yen Lin. cytohubba: identifying hub objects and sub-networks from complex interactome. *BMC systems biology*, 8(4):1–7, 2014.
14. Celine C Berthier, Matthias Kretzler, and Anne Davidson. A systems approach to renal inflammation in sle. *Clinical Immunology*, 185:109–118, 2017.
15. Orthodoxia Nicolaou, Andreas Kousios, Andreas Hadjisavvas, Bernard Lauwerys, Kleitos Sokratos, and Kyriacos Kyriacou. Biomarkers of systemic lupus erythematosus identified using mass spectrometry-based proteomics: a systematic review. *Journal of cellular and molecular medicine*, 21(5):993–1012, 2017.
16. Anupam Guleria, Avadhesh Pratap, Durgesh Dubey, Atul Rawat, Smriti Chaurasia, Edavalath Suresh, Sanat Phatak, Sajal Ajmani, Umesh Kumar, Chunnalal Khetrapal, et al. Nmr based serum metabolomics reveals a distinctive signature in patients with lupus nephritis. *Scientific reports*, 6(1):1–11, 2016.
17. Paulina Gomez-Rubio, Janet Piñero, Esther Molina-Montes, Alba Gutiérrez-Sacristán, Mirari Marquez, Marta Rava, Christoph W Michalski, Antoni Farré, Xavier Molero, Matthias Löhr, et al. Pancreatic cancer and autoimmune diseases: An association sustained by computational and epidemiological case–control approaches. *International journal of cancer*, 144(7):1540–1549, 2019.
18. Min Zhang, Yizhou Wang, Yutong Wang, Ye Bai, and Dongqing Gu. Association between systemic lupus erythematosus and cancer morbidity and mortality: Findings from cohort studies. *Frontiers in oncology*, page 1710, 2022.
19. Min-Seok Seo, Jina Yeo, In Cheol Hwang, and Jae-Yong Shim. Risk of pancreatic cancer in patients with systemic lupus erythematosus: a meta-analysis. *Clinical rheumatology*, 38(11):3109–3116, 2019.
20. Margaret A Tempero, Mokenge P Malafa, Mahmoud Al-Hawary, Stephen W Behrman, Al B Benson, Dana B Cardin, E Gabriela Chiorean, Vincent Chung, Brian Czito, Marco Del Chiaro, et al. Pancreatic adenocarcinoma, version 2.2021, nccn clinical practice guidelines in oncology. *Journal of the National Comprehensive Cancer Network*, 19(4):439–457, 2021.
21. Chen Xu and Zhengchang Su. Identification of cell types from single-cell transcriptomes using a novel clustering method. *Bioinformatics*, 31(12):1974–1980, 2015.
22. Ruizhi Xiang, Wencan Wang, Lei Yang, Shiyuan Wang, Chaohan Xu, and Xiaowen Chen. A comparison for dimensionality reduction methods of single-cell rna-seq data. *Frontiers in genetics*, 12:646936, 2021.
23. Sijie Mai, Songlong Xing, Jiakuan He, Ying Zeng, and Haifeng Hu. Multimodal graph for unaligned multimodal sequence analysis via graph convolution and graph pooling. *ACM Transactions on Multimedia Computing, Communications, and Applications (TOMM)*, 2022.
24. Allon M Klein, Linas Mazutis, Ilke Akartuna, Naren Tallapragada, Adrian Veres, Victor Li, Leonid Peshkin, David A Weitz, and Marc W Kirschner. Droplet barcoding for single-cell transcriptomics applied to embryonic stem cells. *Cell*, 161(5):1187–1201, 2015.
25. Li-Fang Chu, Ning Leng, Jue Zhang, Zhonggang Hou, Daniel Mamott, David T Vereide, Jee Choi, Christina Kendzierski, Ron Stewart, and James A Thomson. Single-cell rna-seq reveals novel regulators of human embryonic stem cell differentiation to definitive endoderm. *Genome biology*, 17(1):1–20, 2016.

REVIEWERS' COMMENTS

Reviewer #1 (Remarks to the Author):

I would like to thank the authors for their additional efforts and explanations. I am still not convinced that this method will see widespread use in the field because it offers very limited advantages compared to existing methods overall, but the revised paper is both interesting and technically sound.

Reviewer #2 (Remarks to the Author):

I have no further comments; thank you for addressing my comments. The paper and code are now much better than their initial state and ready for publication. Congrats to all authors for this beautiful work.

Mo Lotfollahi

Reviewer #3 (Remarks to the Author):

Thanks for the authors' revision, I do not have any major concern. However, based on my knowledge, some parts are not interpreted accurately in the manuscript.

Minor comments:

(1) In introduction part "According, clustering methods have been ... the hierarchical structure", actually, neither SCANPY nor Seurat itself does cluster approach, but only SNN and Leiden do, which are employed by Seurat and SCANPY toolkits (I mentioned this in my previous comments); therefore, the authors may include SNN and Leiden in your writing. And monocle3 does not develop its own cluster approach either.

(2) In the whole section of part C, there is a similar situation. If my understanding is correct, based on your demo examples and your code (function 'dopca', 'get_adj', 'PPMI_matrix', and 'SCMGCA'), as well as your tutorial,

<https://github.com/Philyzh8/scMGCA/blob/master/tutorial/tutorial.md>

<https://github.com/Philyzh8/scMGCA/blob/master/scMGCA/utils.py>

https://github.com/Philyzh8/scMGCA/blob/master/scMGCA/graph_function.py

https://github.com/Philyzh8/scMGCA/blob/master/scMGCA/scmgca_para.py

scMGCA mainly does dimension reduction correction (i.e. optimizing dimension reduction results) instead of generating low-dimensional representation.

If it is true, the interpretation of part C about scMGCA's functions may need to be modified, especially the subtitle. And in your Fig. 3b, you might clarify that the plots are from tSNE with different process/correction methods, scMGCA, SCANPY, and Seurat.

Suggestions but not comments:

<https://github.com/Philyzh8/scMGCA>

In your tool dependencies,

(1) scikit-learn and sklearn is the same package not two, you may check pypi which has explanation.

(2) scanpy and anndata may be listed in your dependencies not in specific setting, as scMGCA's preprocess and data structure are based on them, respectively.

Reviewer #1:

I would like to thank the authors for their additional efforts and explanations. I am still not convinced that this method will see widespread use in the field because it offers very limited advantages compared to existing methods overall, but the revised paper is both interesting and technically sound.

Response: Thank you very much for all your comments. They were highly appreciated and improved the manuscript tremendously. Thank you very much for your precious time and kind attention.

Reviewer #2:

I have no further comments; thank you for addressing my comments. The paper and code are now much better than their initial state and ready for publication. Congrats to all authors for this beautiful work.

Response: Thank you very much for the positive comments and thank you for the detailed suggestions. Thank you very much for your precious time and kind attention.

Reviewer #3:

Thanks for the authors' revision, I do not have any major concern. However, based on my knowledge, some parts are not interpreted accurately in the manuscript.

Response: Thank you very much for the positive and constructive comments. We have duly followed your comments and suggestions to address all the concerns as elaborated in the following responses. Thank you very much for your precious time and kind attention.

1. In introduction part "According, clustering methods have been ... the hierarchical structure", actually, neither SCANPY nor Seurat itself does cluster approach, but only SNN and Leiden do, which are employed by Seurat and SCANPY toolkits (I mentioned this in my previous comments); therefore, the authors may include SNN and Leiden in your writing. And monocle3 does not develop its own cluster approach either.

Response: Thank you very much for your insightful comments. We agree with you and have followed your comments to amend this part. It is copied below for your easy reference:

"...Seurat integrates scRNA-seq data with *in situ* RNA patterns to infer cell locations and clusters (1) while SCANPY is a scalable toolkit for analyzing single-cell gene expression data built jointly with anndata (2), both using shared nearest neighbor (SNN) modular optimization and Leiden for clustering..."

2. In the whole section of part C, there is a similar situation. If my understanding is correct, based on your demo examples and your code (function 'dopca', 'get_adj', 'PPMI_matrix', and 'scMGCA'), as well as your tutorial,

<https://github.com/Philyzh8/scMGCA/blob/master/tutorial/tutorial.md>

<https://github.com/Philyzh8/scMGCA/blob/master/scMGCA/utls.py>

https://github.com/Philyzh8/scMGCA/blob/master/scMGCA/graph_function.py

https://github.com/Philyzh8/scMGCA/blob/master/scMGCA/scmgca_para.py

scMGCA mainly does dimension reduction correction (i.e. optimizing dimension reduction results) instead of generating low-dimensional representation. If it is true, the interpretation of part C about scMGCA's functions may need to be modified, especially the subtitle. And in your Fig. 3b, you might clarify that the plots are from tSNE with different process/correction methods, scMGCA, SCANPY, and Seurat.

Response: Thank you very much for your insightful comments. As suggested by the reviewers, we have revised the subtitle of part C to "scMGCA effectively performs dimensionality reduction correction and improves visualization of scRNA-seq data". Moreover, we removed the content about "generating low-dimensional representation" and revised the interpretation of scMGCA's functions in part C of the revised manuscript. In addition, we have followed your comments to modified the caption of Fig. 3b (now Fig. 3c) to clarify it.

3. Suggestions but not comments: <https://github.com/Philyzh8/scMGCA>. In your tool dependencies, (1) scikit-learn and sklearn is the same package not two, you may check pypi which has explanation. (2) scanpy and anndata may be listed in your dependencies not in specific setting, as scMGCA's preprocess and data structure are based on them, respectively.

Response: Thank you very much for the suggestions. We have followed your comments to made corrections carefully.

References

1. Andrew Butler, Paul Hoffman, Peter Smibert, Efthymia Papalexi, and Rahul Satija. Integrating single-cell transcriptomic data across different conditions, technologies, and species. *Nature biotechnology*, 36(5):411–420, 2018.
2. F Alexander Wolf, Philipp Angerer, and Fabian J Theis. Scanpy: large-scale single-cell gene expression data analysis. *Genome biology*, 19(1):1–5, 2018.